# Macrophages foster anti-tumor immunity by ZEB1-dependent cytotoxic T cell chemoattraction
Kathrin Fuchs[1], Elisabetta D'Avanzo[1], Isabell Armstark[1], Ruthger van Roey[1], Ana Clavel Ezquerra[1], Nino Bindel[1], Katharina Siebenkäs [1], Yussuf Hajjaj[1], Renato Liguori[2,3], Fulvia Ferrazzi [2,3], Lukas Amon[4], Johanna Bulang[5], Julian Hübner [5], Marcel Edler[5], Ece Grace[1], Annemarie Schwab[1], Marwin Alfredo[1], Maria Faas[6], Jochen Ackermann[6], Elena Percivalle[7], Claudia Günther[7,8], Markus H. Hoffmann[9], Gerhard Krönke [6,8,10], Christoph Becker [8,11], Diana Dudziak [4,8,12,13,14,15,16], Philipp Arnold [17], Miriam Woehner[5], Falk Nimmerjahn [5,12,15], Simone Brabletz [1], Marc P. Stemmler [1], Thomas Brabletz [1,14,19] & Harald Schuhwerk [1,18,19] ✉

Tumor-associated macrophages (TAMs) dynamically influence anti-tumor immunity. Understanding TAM function is therefore critical to design immunotherapies. By combining syngeneic models of colorectal and pancreatic cancer with cell type-specific deletion of the epithelial-to-mesenchymal transition driver Zeb1, which is expressed in subsets of TAMs, we discovered that ZEB1 is an intrinsic regulator of TAM-controlled T cell trafficking and anti-tumor immune responses. ZEB1 supports secretion of a subset of chemokines via the constitutive pathway, including CXCL10, CCL2 and CCL22, by regulating their biosynthesis, vesicular transport and release. This elevates cytotoxic T cell (CTL) recruitment in vitro and fosters immunosurveillance by CTLs in tumors and metastases as well in an organotypic model for therapeutic CD8 + T cell addition. Our study identifies ZEB1 in TAMs as a facilitator of anti-tumor immunity, suggests a window of opportunity for cytokine-guided CTL tropism and reinforces the importance of onco-immunological context, particularly in the design of macrophage- and/or cytokine-depleting strategies.

Innate immunity comprises the first line of defense against cancer cells and initiates the adaptive immune response of tumor antigen-specific T cells, ideally eliciting the rejection of tumors[1-3]. Hence, cytotoxic T cells (CTLs) are fundamental executors of anti-tumor and anti-metastasis defense[4]. However, in cancer, this defense is notoriously inefficient due to immunosuppression and/or -evasion, consistently rendering CTL infiltration a prognostic factor for survival[5-8].

Immunomodulation is the consequence of the co-evolution between tumor cells and non-malignant cells[9], eventually constituting the 'tumor microenvironment' (TME) in solid cancers, a complex composition of extracellular matrix, fibroblasts, endothelial and immune cells[2,10]. Tumor-associated macrophages (TAMs) are one of the most abundant innate immune cell types in the TME[3,11,12]. Macrophages/TAMs polarize between "inflammatory" and "alternative" states – the two extreme poles of a broad spectrum of phenotypes which can either have anti- or pro-tumorigenic effects[13,14]. One major role of TAMs is the recruitment of immune cells such as monocytes, neutrophils and T cells via secreted chemokines[15]. Direct anti-

tumor effects include induction of tumor cell apoptosis and phagocytosis[16,17]. On the other hand, pro-tumor TAMs can promote survival, proliferation and epithelial-to-mesenchymal transition (EMT) of cancer cells by secretion of cytokines and growth factors[18,19]. Thus, macrophages/TAMs are plastic immune cells with a broad repertoire of functions that require delicate regulation in the TME.

The partial activation of the developmental process EMT in cancer cells is well-known to induce invasiveness and cellular plasticity, enabling adaptation to environmental challenges, such as posed by cancer therapies or faced during the metastatic cascade[20-23]. The core EMT-inducing transcription factor (EMT-TF) ZEB1 regulates the expression of a wide range of target genes influencing cell adhesion, differentiation, metabolism, therapy resistance, proliferation and the DNA damage response to promote plasticity and malignancy[20,22-24].

ZEB1 is also expressed in cells of the TME[25-33]. Specifically, in cancer-associated fibroblasts (CAFs), ZEB1 was described to be crucial for fibroblast plasticity regulating immune infiltration and supporting tumor

progression[27,33,34]. Moreover, heterozygous knockout of ZEB1 was reported to mitigate the pro-malignant influence of peritoneal macrophages on ovarian cancer cells directly, and homozygous loss of ZEB1 in myeloid cells impaired full activation and recovery from inflammation as well as the response to viral infection in mice[29,35,36]. However, the impact of ZEB1 in TAMs on the composition of the TME and the intercellular crosstalk within it, has not been addressed sufficiently. Hence, the onco-immunological consequences of ZEB1's role in TAM plasticity remain elusive.

Here we employed conditional knockout mouse models and uncovered an unexpected function of ZEB1 in TAMs in suppressing metastatic lung colonization of gastrointestinal cancers. Mechanistically, we show that ZEB1 does not act as a master executor of acute polarization, but is pertinent for the secretion of TAM-derived cytokines to augment immunosurveillance.

## Results

### ZEB1 is heterogeneously expressed in macrophages in CRC and PDAC

ZEB1 is heterogeneously expressed in tumor cells and in cells of the TME[27,28,30,32,33]. Similarly, we observed heterogenous expression of ZEB1 in macrophages in publicly available single cell transcriptomes of colorectal cancer (CRC) and pancreatic cancers (Fig. 1a–d). Supporting the transcriptional data, we detected ZEB1 positive (+) CD68+ macrophages by immunofluorescence (IF) stainings of human CRC (Fig. 1e) and mouse lung metastases (Fig. 1f) formed upon tail vein injection (tvi) of pancreatic ductal adenocarcinoma (PDAC) cells[28]. In order to clarify whether ZEB1 is upregulated in macrophages, we scored the number of ZEB1 + ;CD68+ cells in these samples. Interestingly, tumorous tissue contained more CD68+ cells and an increased share of ZEB1+ cells among those as compared to respective healthy tissue (Fig. 1g, h). Altogether, these data demonstrate that a subset of ZEB1+ macrophages enrich in PDAC and CRC primary tumors as well as in experimental metastases.

### ZEB1 in macrophages is dispensable for organ development and homeostasis

To investigate the role of ZEB1 in macrophages and TAMs, we generated myeloid-specific Zeb1-deleted mice by combining the conditional Zeb1 knockout[37] with LysM-Cre allele (knock-in of Cre into the Lyz2 locus)[38]. The resulting Zeb1$^{flox/flox}$;Lyz2$^{Cre/+}$ mice were designated as 'LysM$^{\Delta Zeb1}$' and their Lyz2 wildtype littermates as 'LysM$^{Ctrl}$' (Fig. S1a, b). LysM$^{\Delta Zeb1}$ mice do not show any obvious phenotypic abnormalities or gross alterations in tissue architecture, as deduced from histologic analyses (Fig. S1c, d), in line with a recent study[36]. Consistently, flow cytometry of cell suspensions from colon, lung, spleen and blood from LysM$^{\Delta Zeb1}$ mice revealed unchanged immune cell compositions, including macrophages (CD45 + ;CD11b + ;F4/80 + ), as compared to their LysM$^{Ctrl}$ littermates (Fig. 2a, S1e–h). To verify activity of LysM-Cre in LysM$^{\Delta Zeb1}$ mice in macrophages and neutrophils, the mT/mG Cre-reporter allele was employed, which switches from ubiquitous tdTomato to GFP expression upon Cre-mediated recombination[39]. As expected, lymphoid and myeloid compartments in pancreata, lungs and livers of mT/mG+ LysM$^{\Delta Zeb1}$ mice revealed preferential recombination in macrophages and neutrophils, as well as dendritic cells, albeit to a much lower extent in the latter (Fig. 2b–d, S2a–e). IF analysis suggested that organs of LysM$^{\Delta Zeb1}$ mice contained less CD68 + ZEB1+ macrophages as opposed to LysM$^{Ctrl}$ mice (Fig. 2e, S2f). However, only subsets of CD68+ were GFP+ (and vice versa) or ZEB1+ in situ (Fig. 2d, e), suggesting scarce expression of ZEB1 within heterogeneous macrophage-like myeloid populations in healthy tissues. This prompted us to investigate the extent of ZEB1 protein loss in naïve LysM$^{\Delta Zeb1}$ mice in more detail by intracellular flow cytometry, focusing on spleen, lung and liver. As compared to their LysM$^{Ctrl}$ littermates, LysM$^{\Delta Zeb1}$ mice contained significantly less ZEB1+ splenic (CD45 + ;Ly6C-;F4/80 + ;CD11b + ) as well as alveolar (SiglecF + ;CD11b-low) but not non-alveolar (CD11b-high) pulmonary macrophages (CD45 + ;Ly6G-;F4/80 + ;CD68 + ). For the hepatic Kupffer cells (CD45 + ;CD11b + ;F4/80 + ; Tim4 + ;CD206$^{high/low}$) and the very

scarce pancreatic macrophages (CD45 + ;CD68 + ;CD11b + ) a visible trend was observed (Fig. 2f, S2g, h). Collectively, these data show that ZEB1 is lost from subsets of macrophages in LysM$^{\Delta Zeb1}$ mice and that ZEB1 in these LysM-expressing cells is neither essential for organ, macrophage, neutrophil or lymphoid development nor for tissue and immune cell homeostasis.

### ZEB1 in macrophages subverts tumor growth and lung colonization

To determine the impact of ZEB1 loss in macrophages on tumor growth, we first employed a syngeneic subcutaneous (s.c.) model of CRC by injecting CMT-93 cells into LysM$^{Ctrl}$ and LysM$^{\Delta Zeb1}$ mice. As expected, we observed tumor growth for 2-3 weeks after injection of one million cells. Interestingly, this was followed by complete tumor regression in 40% of LysM$^{Ctrl}$, but not at all in LysM$^{\Delta Zeb1}$ mice (Fig. 3a, b) causing a significant reduction of tumor load at endpoints (Fig. 3c). Similar tumoricidal basal immunogenicity of this model has been reported before[40]. Thus, to rule out a cell line-specific effect, we also injected highly tumorigenic MC-38 CRC cells. While no difference was observed upon s.c. injection of a large number of cells, MC-38 tumor formation was significantly sustained in LysM$^{\Delta Zeb1}$ mice as compared to LysM$^{Ctrl}$ mice upon reducing tumor cell numbers for engraftment (Fig. 3d). The number of infiltrating macrophages was unchanged upon loss of ZEB1 (Fig. S3a, b). These data suggest that ZEB1 in macrophages impedes tumor outgrowth.

As CRC and PDAC frequently metastasize to the lungs, our data prompted us to clarify the potential impact of ZEB1 in macrophages on metastatic lung colonization. To achieve this, we applied an experimental metastasis model by intravenous injection of luciferase-expressing PDAC (KPC) and CRC (MC-38) cells into LysM$^{Ctrl}$ and LysM$^{\Delta Zeb1}$ mice. By longitudinal in vivo bioluminescence imaging (BLI) of luciferase activity, we analyzed metastatic seeding and outgrowth over time. In both models, both genotypes displayed equal BLI signals for up to 14 days post injection (dpi), suggesting similar initial metastatic seeding. Strikingly, while the signals plateaued further on in LysM$^{Ctrl}$ mice, LysM$^{\Delta Zeb1}$ mice displayed a sudden elevation of BLI signal, aggravating their metastatic burden (Fig. 3e–i), albeit only by trend upon MC-38 injection. Akin to the s.c. tumors, the number of macrophages and monocytes was unchanged in metastatic LysM$^{\Delta Zeb1}$ lungs (Fig. S3c–e).We validated the increased lung colonization of KPC cells in an additional model of mice lacking ZEB1 in monocytes and macrophages (Cx3cr1-Cre[41];Fig. S3f). As we observed recombination in LysM-Cre mice additionally in neutrophils, we employed a model lacking ZEB1 in neutrophils (Ly6g$^{Cre-Tom}$[42]) to decipher the responsible cell type, revealing a much weaker effect as compared to the LysM- and Cx3cr1-Cre strains (Fig. S3g).

These findings demonstrate impaired restriction of metastatic outgrowth upon loss of ZEB1 in macrophages.

### ZEB1 is dispensable for phagocytosis but selectively promotes cytokine secretion in macrophages

To gain insights into how ZEB1 in macrophages controls tumor and metastasis outgrowth, we utilized primary bone-marrow-derived macrophages (BMDMs) from LysM$^{Ctrl}$ and LysM$^{\Delta Zeb1}$ mice. As BMDMs increase Lyz2 (LysM) expression levels during maturation within the first days in vitro (div)[43], we observed increasing LysM-Cre mediated recombination of the mT/mG reporter from 3 div in LysM$^{\Delta Zeb1}$ BMDMs (Fig. 4a, S4a). Loss of ZEB1 in LysM$^{\Delta Zeb1}$ BMDMs was verified by qPCR and western blotting of LysM-Cre-recombined GFP+ BMDMs enriched by FACS (Fig. 4a–c, S4b). FACS-enriched LysM$^{\Delta Zeb1}$ BMDMs exhibited normal morphology, grew normally in cultures and did not show compensatory upregulation of other core EMT-TFs (Fig. S4b–f). We then explored an involvement of ZEB1 in archetype BMDM polarization into highly inflammatory and immuno-suppressive states by stimulation with lipopolysaccharide (LPS) and IL-4, respectively. Consistent with recent literature[36], Zeb1 expression was upregulated in BMDMs upon LPS, but not upon IL-4 treatment (Fig. 4c, S4g), indicating that ZEB1 likely does not play a major role in IL-4-polarized, but

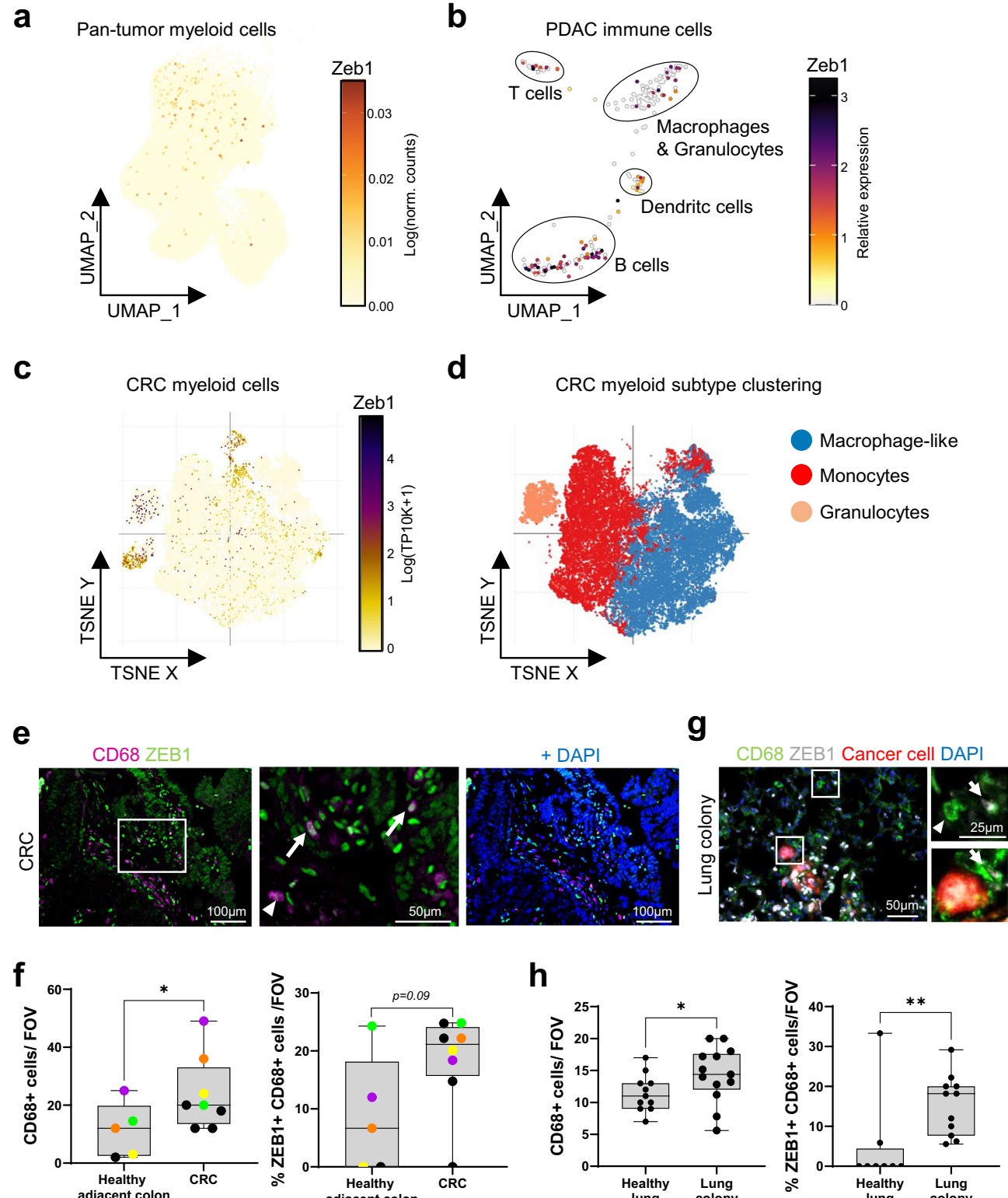

**Fig. 1 | ZEB1 is heterogeneously expressed in macrophages in CRC and PDAC.**
UMAP plot visualization of Zeb1 in single cell RNA sequencing data in myeloid cells derived from human pan tumors[108] (**a**), from subclustered immune cells from murine PDAC[109] (**b**), and of myeloid cells derived from human CRC (**c**). **d** Myeloid subtype clustering of cells from (**c**), as indicated. In (**c**) and (**d**), plots and cell type annotations were retrieved from the single cell portal from the Broad Institute using the dataset from[112]. Representative images (**e**) and quantification (**f**) of immuno-fluorescence staining (IF) for the number of CD68+ cells (left) per field of view and the fraction of ZEB1+ cells among CD68+ cells (right) of human colorectal cancer (CRC) with DAPI-stained nuclei. Insets show a higher magnification. Arrows indicate CD68 + ;ZEB1+ cells and arrowheads CD68 + ;ZEB1- cells (n (healthy adjacent colon/CRC) = 5/8). Color-code in (**f**) indicates tissue areas matched per case. Representative images (**g**) and quantification (**h**) of IF for CD68 and ZEB1 as in (**f**) of murine lungs and metastatic lung colonies with DAPI-stained nuclei. Insets show a higher magnification. Arrows indicate CD68 + ;ZEB1+ cells and arrowheads CD68 + ;ZEB1- cells (n ≥ 11; Mann-Whitney test (**f, h**)). *$p < 0.05$; **$p < 0.01$; ns: not significant.

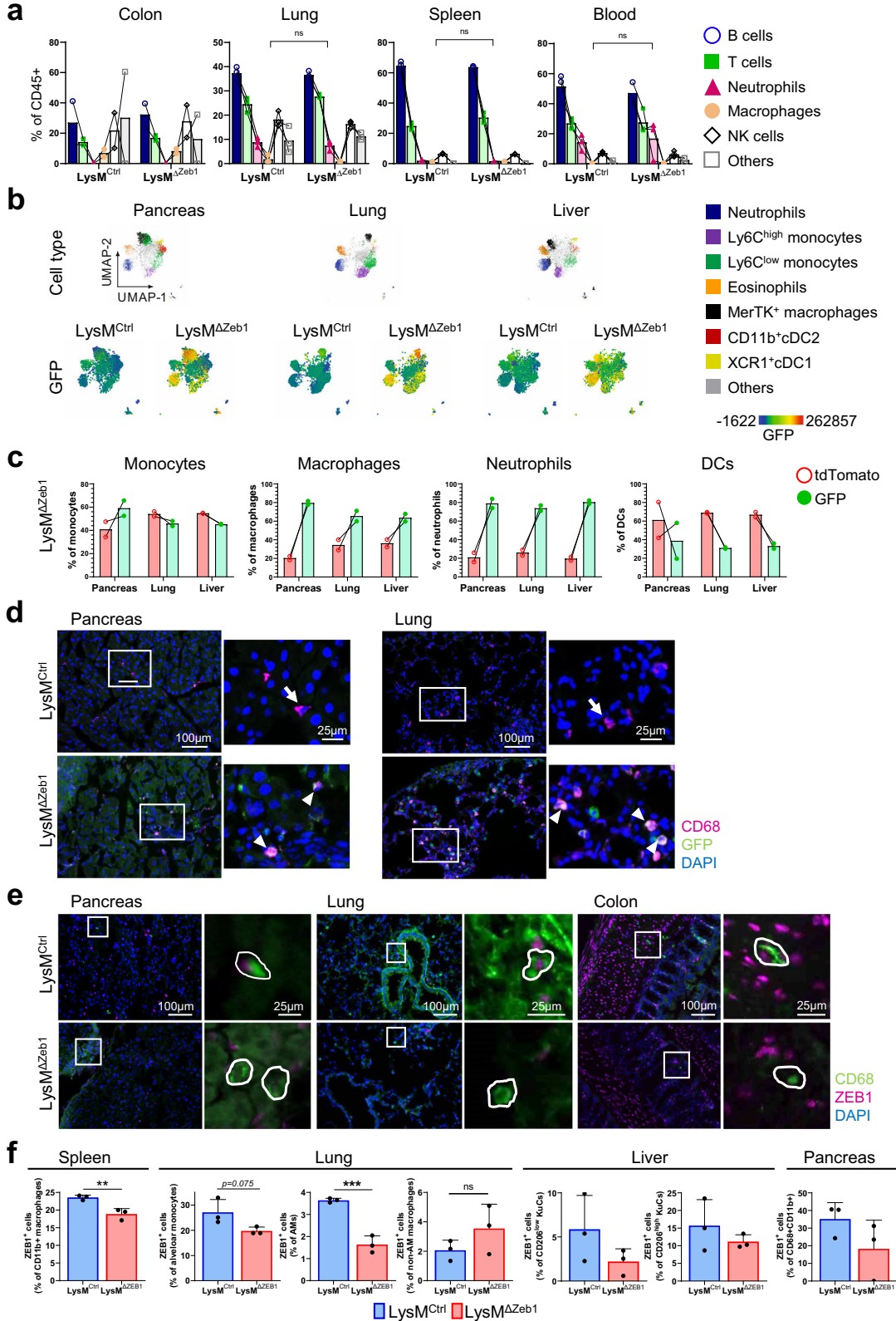

**Fig. 2 | ZEB1 in LysM-expressing cells is dispensable for organogenesis and homeostasis. a** Percentages of B cells, T cells, neutrophils, macrophages and NK cells of CD45+ cells in organs of LysM$^{Ctrl}$ and LysM$^{\Delta Zeb1}$ mice as determined by flow cytometry ($n = 3$; mean +SD; 2-way ANOVA). **b** Flow cytometry UMAP clustering of cells isolated from organs of mT/mG-positive(+) LysM$^{Ctrl}$ and LysM$^{\Delta Zeb1}$ mice. GFP expression is depicted as color gradient ($n = 2$). **c** Percentage of tdTomato+ or GFP+ cells in immune cell subtypes of mT/mG-positive LysM$^{\Delta Zeb1}$ mice. **d** Representative images of CD68 IF of organs of mT/mG+ LysM$^{Ctrl}$ and LysM$^{\Delta Zeb1}$

mice with DAPI-stained nuclei. Insets show a higher magnification. Arrows show CD68+ cells. Arrowheads show CD68 + ;GFP+ cells. Note that the tdTomato channel is excluded. **e** Representative images of CD68 and ZEB1 in organs of mT/mG-negative LysM$^{Ctrl}$ and LysM$^{\Delta Zeb1}$ mice with DAPI-stained nuclei (see S2f for scoring of CD68 + ;ZEB1+cells). **f** Percentages of ZEB1+ cells among the indicated CD45+ macrophage populations in organs of LysM$^{Ctrl}$ and LysM$^{\Delta Zeb1}$ mice, as determined by intracellular flow cytometry ($n = 3$; mean +SD; ***$p < 0.001$; **$p < 0.01$; two-tailed t-test).

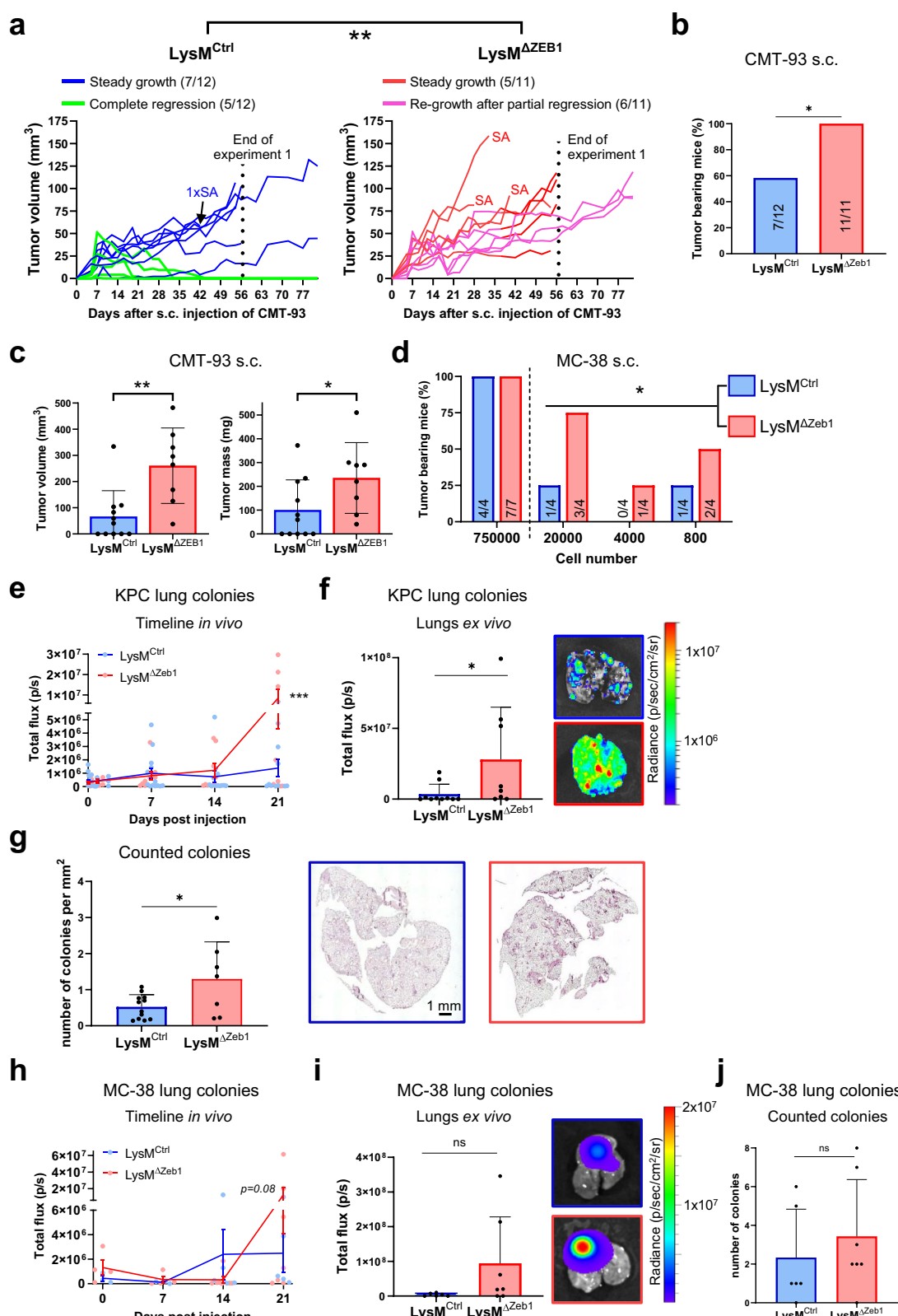

in LPS-polarized BMDMs. Macrophages in general, but in particular post LPS-dependent polarization, are efficient phagocytes and produce large amounts of inflammatory cytokines[13,44]. As phagocytosis of fluorescent bioparticles and tumor cell debris was insignificantly - withal minimally – altered in *Zeb1*-deficient compared to -proficient BMDMs (Fig. 4d, e), we screened for secreted cytokines. LysM^ΔZeb1 BMDMs secreted different amounts of various cytokines compared to LysM^Ctrl BMDMs in a partially stimulus-independent manner. Among these, CCL2 and CCL22 were consistently reduced and confirmed in a quantitative assay alongside CXCL10 upon LPS treatment (Fig. 4f, g). Notably, we observed similarly diminished secretion upon stimulation with IFNγ, which also induced CXCL9 secretion mainly in LysM^Ctrl but not LysM^ΔZeb1 BMDMs (Fig. 4h).

**Fig. 3 | ZEB1 in macrophages subverts tumor growth and lung colonization.**
**a** Caliper measurements of tumor growth and regression in LysM^Ctrl and LysM^ΔZeb1 mice after s.c. injection of $1 \times 10^6$ CMT-93. Sacrifices due to tumor ulceration (SA) are indicated. **b** Percentages of tumor-bearing LysM^Ctrl and LysM^ΔZeb1 mice from (**a**) at endpoints after s.c. injection of $1 \times 10^6$ CMT-93. The Number of mice (tumor-bearing/ total) is indicated. **c** Volumes and masses excluding SA of CMT-93 tumors from (**a, b**). **d** Percentages of tumor-bearing LysM^Ctrl and LysM^ΔZeb1 mice at endpoints after injection of the indicated numbers of MC-38 cells. The Number of mice (tumor-bearing/ total) is indicated in the respective bars. **e** In vivo BLI signal of tail vein injected KPC tumor cells over time in LysM^Ctrl ($n = 6$) and LysM^ΔZeb1 ($n = 7$) mice (means ± SEM). **f** Ex vivo BLI signal of lungs from mice in (**e**) at endpoint with representative images (means +SD). **g** Number of KPC colonies and representative images of H&E-stained lungs of LysM^Ctrl ($n = 12$) and LysM^ΔZeb1 ($n = 7$) mice (means +SD). **h** In vivo BLI signal of tail vein injected MC-38 tumor cells over time in LysM^Ctrl ($n = 6$) and LysM^ΔZeb1 ($n = 7$) mice (means ± SEM). **i** Ex vivo BLI signal of lungs from mice in (**h**) at endpoint with representative images (means +SD). **j** Number of counted MC-38 colonies in H&E staining of lungs of LysM^Ctrl ($n = 6$) and LysM^ΔZeb1 ($n = 7$) mice (means +SD). ***$p < 0.001$; *$p < 0.05$; 2-way ANOVA (**a, e, h**); Fisher´s exact test (**b**); two-tailed *t*-test (**c, g, j**); test for differences in stem cell frequencies using the limiting dilution assay online tool ELDA[113] (**d**). Mann-Whitney (**f, i**). In vivo experiments have been performed at least twice with the indicated total number of mice.

These data indicate an important role of ZEB1 in inflammatory macrophage polarization, particularly in the secretion of the chemokines CCL2, CCL22 and CXCL10.

## ZEB1 supports inflammatory macrophage polarization and cytokine secretion by regulating protein biosynthesis and trafficking

We next investigated how ZEB1 supports macrophage polarization and the secretion of cytokines including CCL2, CCL22 and CXCL10. As ZEB1 is a transcription factor, we characterized the transcriptional responses of BMDMs to LPS and IL-4 stimulation globally by bulk RNA sequencing. As expected, LPS and IL-4 treatments changed gene expression dramatically, albeit in both genotypes. LysM^ΔZeb1 showed more differentially expressed genes (DEGs) in response to both treatments than LysM^Ctrl. However, 62.3% (LPS) and 55.6% (IL-4) of the DEGs were shared between the genotypes and thus apparently not dependent on ZEB1 (Fig. 5a, b). Gene Ontology (GO) term analyses identified very similar enrichments in LysM^ΔZeb1 and LysM^Ctrl with respect to inflammation and the expected LPS and IL-4 responses, akin to the analysis of those DEGs shared between the genotypes (Supplementary Data 1). Consistently, we did not observe robust alterations in expression of polarization genes, cytokines and established M1/M2-like markers comparing LysM^Ctrl to LysM^ΔZeb1 BMDMs - neither in this dataset, nor in a customized qRT-PCR array (Fig. S5a–c). *Ccl2* and *Ccl22* mRNA levels were moderately but insignificantly reduced in LysM^ΔZeb1 compared with LysM^Ctrl BMDMs, partially matching the secretome data (Fig. S5a, b).

Intrigued by the considerable number of DEGs exclusively in LysM^Ctrl or LysM^ΔZeb1 BMDMs, we performed GO term analyses on these genes (Supplementary Data 2). IL-4-induced exclusive DEGs did not retrieve significant GO terms in LysM^Ctrl BMDMs but indicated moderately altered cell cycling in LysM^ΔZeb1 BMDMs, consistent with the multifaceted role of ZEB1 in cell cycle control, which we addressed systematically recently[23,32]. In the LPS condition, LysM^Ctrl BMDMs showed altered cellular respiration and ribonucleotide biosynthesis, while, interestingly, protein translation, ubiquitination and transport were deregulated in LysM^ΔZeb1 BMDMs on the mRNA level, with major players in protein trafficking, such as members of the Rab GTPases, Vamp SNAREs and syntaxin (Stx) families (Fig. 5c, d). These data indicate general polarization competence of LysM^ΔZeb1 BMDMs concomitant to ample alterations in LysM^ΔZeb1 BMDMs on the post-transcriptional level upon LPS treatment.

To validate these post-transcriptional changes, we first analyzed global translation by measuring the incorporation of O-Propargyl-Puromycin (OPP), a 'clickable' Puromycin analog, into nascent proteins in LysM^Ctrl and LysM^ΔZeb1 BMDMs. As expected, protein synthesis increased upon LPS treatment but the full translational capacity was only slightly reduced in *Zeb1*-deficient BMDMs (Fig. 5e). To evaluate alterations in intracellular vesicular trafficking in *Zeb1*-deficient macrophages on the ultrastructural level, we performed transmission electron microscopy (TEM) of LPS-treated BMDMs. Interestingly, while the total number of intracellular vesicles was independent of ZEB1, LysM^ΔZeb1 BMDMs contained more multivesicular bodies (MVBs) by trend and, particularly, an enrichment of electron-dense vesicles as compared to LysM^Ctrl BMDMs, corroborating our hypothesis that loss of ZEB1 affects vesicle trafficking (Fig. 5f). Since macrophages mostly lack secretory granules, unlike *e.g.*, mast cells, these

electron-dense vesicles might either represent accumulated, cargo-laden endosomes within constitutive cytokine secretion that failed to be released or autophagosomes and/or lysosomes (as well as fusion thereof) for cargo degradation[45–47]. However, intracellular cytokine concentrations were significantly lower in LysM^ΔZeb1 than in LysM^Ctrl BMDMs (Figs. 5g and S5d), rejecting the first hypothesis. To gain molecular insights into this phenotype, we performed western blotting for trafficking factors that were altered in the transcriptional LPS response of *Zeb1*-deficient BMDMs (Fig. 5h, i). Interestingly, the levels of the GTPase RAB6 regulating retrograde endosomes-to-trans-golgi trafficking and of RAB35, a regulator of endocytic recycling and autophagosome maturation[48], were higher in LysM^ΔZeb1 BMDMs upon LPS treatment, together indicating altered endosome sorting and abundance in *Zeb1*-deficient BMDMs. Strikingly, LysM^ΔZeb1 BMDMs contained remarkably small amounts of the SNARE VAMP3, a known regulator of exocytosis that is crucial for constitutive cytokine secretion via recycling endosomes in myeloid cells[45,49–51] (Fig. 5i). Notably, it appeared to be slightly more upregulated upon LPS stimulation in LysM^ΔZeb1 than in LysM^Ctrl BMDMs, consistent with the RNAseq data. The quite opposite was observed for VAMP8, a late endosomal/lysosomal SNARE and driver of autophagosome-lysosome fusions that is important for degranulation of secretory granules in mast cells with limited influence on constitutive cytokine secretion utilized by macrophages, *e.g.* for CCL2[51–55] (Fig. 5i). Collectively, these data show that ZEB1 plays diverse roles in regulating the entire biosynthetic, endosomal trafficking and constitutive secretion process of certain cytokines, such as CCL2, CCL22 and CXCL10, during inflammatory activation of BMDMs (Fig. 5j).

## ZEB1 in macrophages has no major direct effect on tumor cells but is important for CCL2- and CCL22-driven recruitment of CD8 + T cells in vitro

Inflammatory cytokines can launch oncogenic signaling in tumor cells[22]. Consistently, co-culture of BMDMs with tumor cells increased tumor cell proliferation and invasion (Fig. 6a, b), yet to the same extent in ZEB1-proficient and -deficient BMDMs. Transfer of conditioned medium from BMDMs to tumor cells had no pro-proliferative influence on them (Fig. 6c). These data suggest that macrophages do not directly influence tumor cells in a ZEB1-dependent manner.

CXCL10 is a well-known T cell attractant and the pleiotropic cytokines CCL2 and CCL22 are potent monocyte attractants but also recruit T cells[56–58]. As macrophage and monocyte influx was similar in s.c. tumors as well as metastatic lung colonies of LysM^Ctrl and LysM^ΔZeb1 mice (Fig. S3a–d), we investigated myeloid ZEB1-dependent T cell recruitment by performing T cell migration assays through narrow-pored transwells towards BMDMs. Strikingly, *Zeb1*-deficient BMDMs, unlike the *Zeb1*-proficient ones, were incapable of recruiting CD8 + T cells, which was rescued by addition of recombinant CCL2 and CCL22, but not by CXCL10, in amounts levelling out the concentration differences between both BMDM genotypes that were quantified before (Fig. 4g, Fig. 6d, left panel). Notably, absence of an additive influence of CCL2 and CCL22 co-supplementation on CD8 + T cell migration was probably due to coupled surface expression of the cognate receptors for CCL2 (CCR2/CD192) and CCL22 (CCR4/CD194) but not of CXCL10 (CXCR3/CD183) on the splenic CD8 + T cells used in this assay (Fig. S6a). These data imply that the low residual amount of secreted

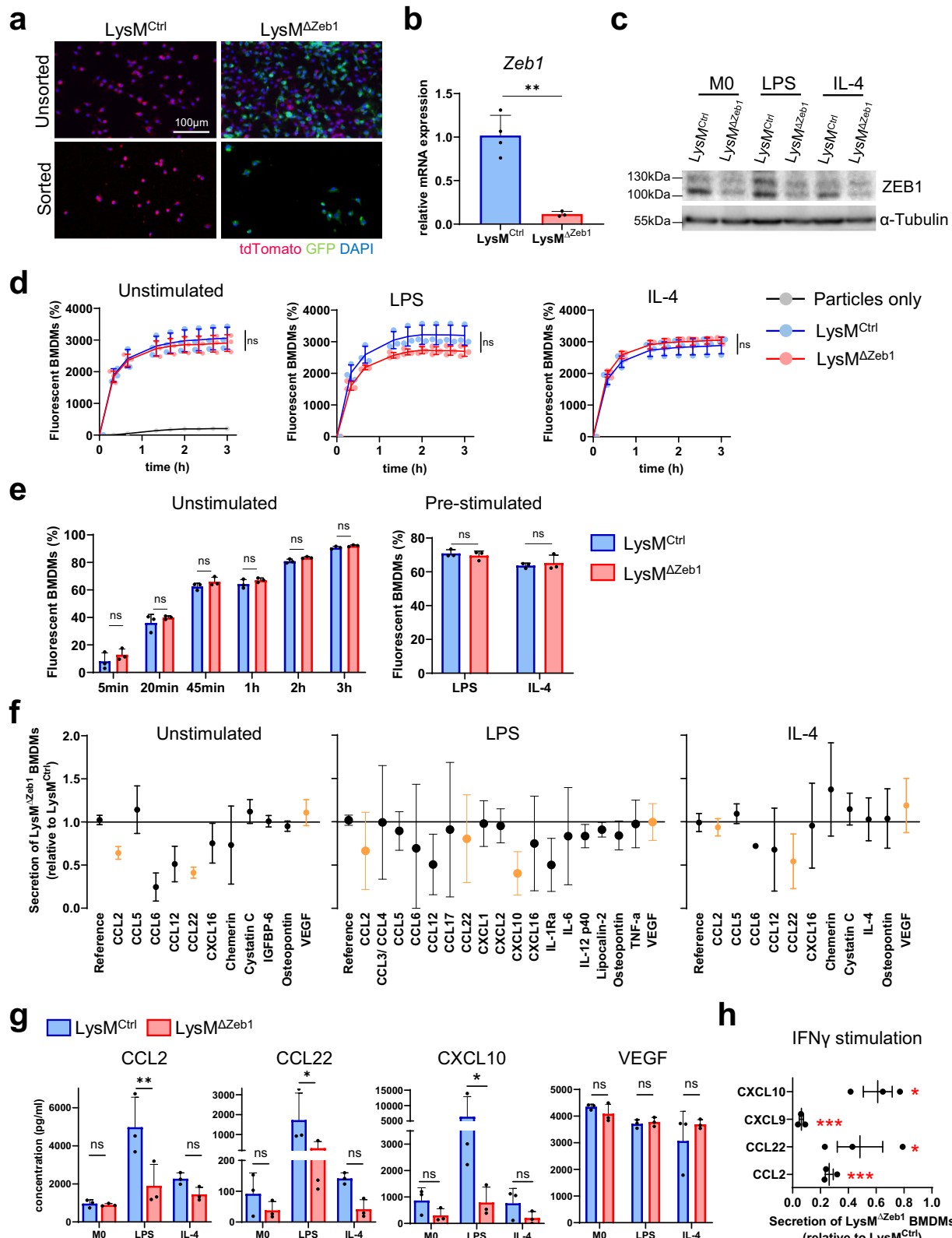

CXCL10 in LysM$^{\Delta Zeb1}$ BMDMs suffices to recruit CXCR3 + CD8 + T cells in vitro, whereas the partially redundant CCL2 and CCL22 become functionally limiting. Substantiating the cytokine supplementation effect, CCL2- and CCL22-depleting antibodies lowered CD8 + T cell recruitment in LysM$^{Ctrl}$ BMDM co-cultures to LysM$^{\Delta Zeb1}$ BMDM co-culture levels (Fig. 6d, right panel). Notably, no major differences in proliferation of CD8 + T cells

upon co-culture with LysM$^{Ctrl}$ or LysM$^{\Delta Zeb1}$ BMDMs pre-stimulated with LPS or IL-4 or unstimulated were observed (Fig. S6b). These data show that CD8 + T cells are recruited by BMDMs via CCL2 and CCL22 in a ZEB1-dependent manner in vitro.

To corroborate the relevance of these findings in cancer, we performed in silico analyses. In human CRC single cell transcriptomes, *CCL2* and

**Fig. 4 | ZEB1 is dispensable for phagocytosis but selectively promotes cytokine secretion in macrophages. a** Representative fluorescence images of unsorted and sorted mT/mG+ LysM$^{Ctrl}$ and LysM$^{ΔZeb1}$ BMDMs at 8 days in vitro (div) with DAPI-stained nuclei. **b** Zeb1 mRNA expression levels of sorted LysM$^{Ctrl}$ ($n = 4$) and LysM$^{ΔZeb1}$ ($n = 3$) BMDMs. **c** Representative western blot of ZEB1 and α-Tubulin as loading control of unstimulated, LPS or IL-4 stimulated LysM$^{Ctrl}$ and LysM$^{ΔZeb1}$ BMDMs. **d** Phagocytosis of phRodo E. coli particles by unstimulated, LPS or IL-4 pre-stimulated LysM$^{Ctrl}$ and LysM$^{ΔZeb1}$ BMDMs, as measured by percentage of GFP+ BMDMs ($n = 3$ for BMDM conditions, $n = 1$ for the particles-only condition).

**e** Time-course of phagocytosis of KPC debris by unstimulated LysM$^{Ctrl}$ and LysM$^{ΔZeb1}$ BMDMs or at t = 20 min after pre-stimulation with LPS or IL-4, as measured by percentage of Claret+ BMDMs ($n = 3$). **f** Secretome assays showing relative secreted molecules in supernatants of unstimulated, LPS or IL-4 pre-stimulated LysM$^{ΔZeb1}$ BMDMs normalized to the respective LysM$^{Ctrl}$ values ($n = 3$). **g** Quantification of selected cytokines using a bead-based immunoassay in LysM$^{Ctrl}$ and LysM$^{ΔZeb1}$ BMDM supernatants ($n = 3$; means ± SD). **h** Secretome assay as in (**f**) after IFNγ stimulation of BMDMs ($n = 3$). *$p < 0.05$; **$p < 0.01$; ***$p < 0.001$; ns: not significant; two-tailed *t*-test (**b**, **e** (right), **h**); 2-way ANOVA (**e** (left), **g**).

*CCL22* are expressed in macrophages and monocytes, albeit to a lower extent than *CXCL10* (Fig. S7a). Furthermore, subsets of *CD8 + T* cells displayed expression of *CCR2* and *CCR4*, albeit to a lower extent than *CXCR3*, in human CRC (Fig. S7b), pancreatic neuroendocrine and breast cancers. This was also reflected in a dataset derived from T cells in tumor-draining lymph nodes from mice bearing s.c. MC-38 (CRC) and B16 (melanoma) tumors (Fig. S8a–c)[59]. In line with this, CD8 + T cell abundance correlated significantly with the expression levels of *CCR2*, *CCR4* and *CXCR3* (Fig. S8d), as deconvoluted from bulk RNA sequencing datasets using *Timer2.0*[60]. These data altogether indicate a functional crosstalk of macrophage-derived *CCL2*, *CC22* and *CXCL10* with CD8 + T cell subsets in cancers, including human pancreatic, colon and rectal malignancies.

## ZEB1 in macrophages promotes CD8+ cell abundance in s.c. tumors as well as during metastatic lung colonization and concomitant tumor cell killing

To clarify whether the defect in CCL2-/CCL22-dependent CD8 + T cell recruitment affects CD8 + T cell infiltration into tumors and metastatic lung colonies, we performed IHC analyses. Strikingly, we found substantially diminished CD8+ cells in MC-38 and CMT-93 s.c. tumors, accompanied by reduced tumor cell apoptosis as marked by cleaved Caspase 3 (Fig. 7a). Underscoring an influence of adaptive immunity on impeding outgrowth of CMT-93 tumor cells in LysM$^{Ctrl}$ mice, all allografted immune-incompetent NOD-SCID gamma (NSG) mice formed tumors and were thus almost indistinguishable from LysM$^{ΔZeb1}$ mice, whereas tumor formation failed in almost 50% of LysM$^{Ctrl}$ mice (Fig. 7b). These data show that ZEB1 in macrophages is crucial for CD8+ cell infiltration into s.c. tumors. Similarly, in metastatic lung colonies of MC-38 cells of LysM$^{ΔZeb1}$ mice, we observed reduced tumor cell death and influx of CD8+ cells (Fig. 7c). Reconciling this observation with the in vivo BLI dynamics of metastatic outgrowth indicating net stagnation of growth in LysM$^{Ctrl}$ but not in LysM$^{ΔZeb1}$ lungs, we hypothesized that CD8+ cell infiltration may precede tumor cell elimination in the lungs. To investigate this possibility and to validate the findings of the MC-38 model, we analyzed lungs by IHC at different time points during metastatic lung colonization of KPC cells (Fig. 7d). In line with our hypothesis, the number of CD8+ cells began rising at 10 dpi in colonies of LysM$^{Ctrl}$, but not of LysM$^{ΔZeb1}$ mice, which displayed a defect in CD8+ cell infiltration. Consequently, tumor cell death transiently peaked at 14 dpi, but to a much higher extent in LysM$^{Ctrl}$ than in LysM$^{ΔZeb1}$ mice. This relative difference remained until 21 dpi, despite a general reduction of tumor cell apoptosis (Fig. 7d). Consistent with the CD8+ influx, CCL2 levels in situ peaked at 10 dpi and failed to be induced fully throughout the metastatic colonization of LysM$^{ΔZeb1}$ lungs. As CCL2 and CCL22 can also attract T helper cells[61], we scored CD4+ cells in these lung colonies. Their number increased similarly in LysM$^{Ctrl}$ and LysM$^{ΔZeb1}$ mice over time (Fig. S9). Notably, tumor cell proliferation was increased in KPC colonies upon ZEB1 loss in macrophages but remained unaffected in MC-38 ones (Fig. 7e). This indicates a cell line-dependent contextual impact of ZEB1 in macrophages on tumor cell proliferation but not on tumoricidal CD8+ cell influx.

Our findings suggested that CCL2 and CCL22 from ZEB1-expressing alveolar macrophages increase CTL abundance to limit metastatic lung colonization of KPC tumor cells. To confirm this, we combined syngeneic precision cut lung slice (PCLS) cultures[62] with therapeutic addition of

unprimed, Claret-labelled CD8 + T cells (CD8 + TCA) after seeding of tdTomato-expressing KPC cells (Fig. 7f, g). Akin to the in vivo observations, CD8 + TCA transiently controlled KPC growth on LysM$^{Ctrl}$ lungs but not on LysM$^{ΔZeb1}$ lungs (Fig. 7g). Strikingly, CCR2 and CCR4 antagonists abolished CD8 + TCA-driven control of KPC growth on LysM$^{Ctrl}$ lungs whilst their growth on LysM$^{ΔZeb1}$ lungs remained unaffected (Fig. 7h). Notably, early KPC growth control in LysM$^{Ctrl}$ PCLS correlated with lower CTL abundance, as approximated by total Claret intensity (Fig. S10a), which was reduced upon treatment with the antagonists (Fig. S10b). These data collectively reveal the importance of macrophage-initiated CTL engagement via the CCL2-CCR2 and CCL22-CCR4 axes for controlling metastatic lung colonization as well as therapeutic CD8 + TCA efficiency.

In summary, our data show that ZEB1 in macrophages promotes CD8+ cell infiltration into s.c. tumors and metastatic colonies in the lung, preceding tumor cell death to counteract tumor outgrowth.

## Discussion

ZEB1 in macrophages appears largely dispensable for organogenesis and hematopoiesis, but modulates the anti-tumor immune response. TAMs are mostly associated with tumor promotion, but also exhibit tumor-suppressive functions[14,63]. In this regard, we discovered an important function of ZEB1 in macrophages to limit tumor and lung metastatic growth by cytokine-mediated CTL recruitment in syngeneic models of two gastrointestinal tumor entities, without direct noticeable effects on tumor cells.

Previously, an ovarian carcinomatosis model revealed haploinsufficiency of *Zeb1* in peritoneal macrophages in inducing a tumor-promoting phenotype in tumor cells directly[29]. Specifically, using heterozygous zygotic knockout of Zeb1 (*i.e.* retaining one functional allele), it was shown that ZEB1 levels increase in peritoneal macrophages upon i.p. injection of ID8 cells, alongside myeloid-related genes including *Ccr2*. In ID8-linked TAMs, lowering ZEB1 levels proportionally decreased expression of these genes, of which MMP9 induced de-differentiation, proliferation and chemoresistance in tumor cells. The tumor cells in turn upregulated *Ccl2* expression to reinforce monocyte recruitment. However, cell line specificity and contributions of other immune cells have not been addressed, rendering it an oncogenic circuit in a single cell line model that remains immunologically insufficiently understood. ZEB1 induction in ID8-linked TAMs matches increased abundance of ZEB1 + ;CD68+ cells in our CRC data. However, in our study, homozygous deletion of *Zeb1* in macrophages neither weakened the pro-malignant effect of TAMs on tumor cells, nor did it reduce *Mmp9* expression. The opposing net effect in tumor growth may also be caused by a cell line-specific effect on tumor cells, immunological context and/or heterozygous zygotic *Zeb1* loss, which may cloud ZEB1's immunological functions in TAMs.

Importantly, we confirmed leveraged tumor suppression in different tumor models and entities. Using additional Cre lines we provide evidence that myeloid ZEB1 exerts its anti-tumorigenic effect mostly via TAMs and not via cancer-relevant neutrophils, in which *Zeb1* is expressed but has not been functionally investigated yet[64–66]. Notably, it cannot be excluded that myeloid-derived suppressor cells are targeted as well and potentially influence these in vivo effects. In light of the ID8-centered aggravation of carcinomatosis by ZEB1 in TAMs[29], its requirement for efficient anti-tumor immunity in gastrointestinal cancer models uncovered by us shows that relevant immuno-oncological contextures are manifested by ZEB1 in TAMs.

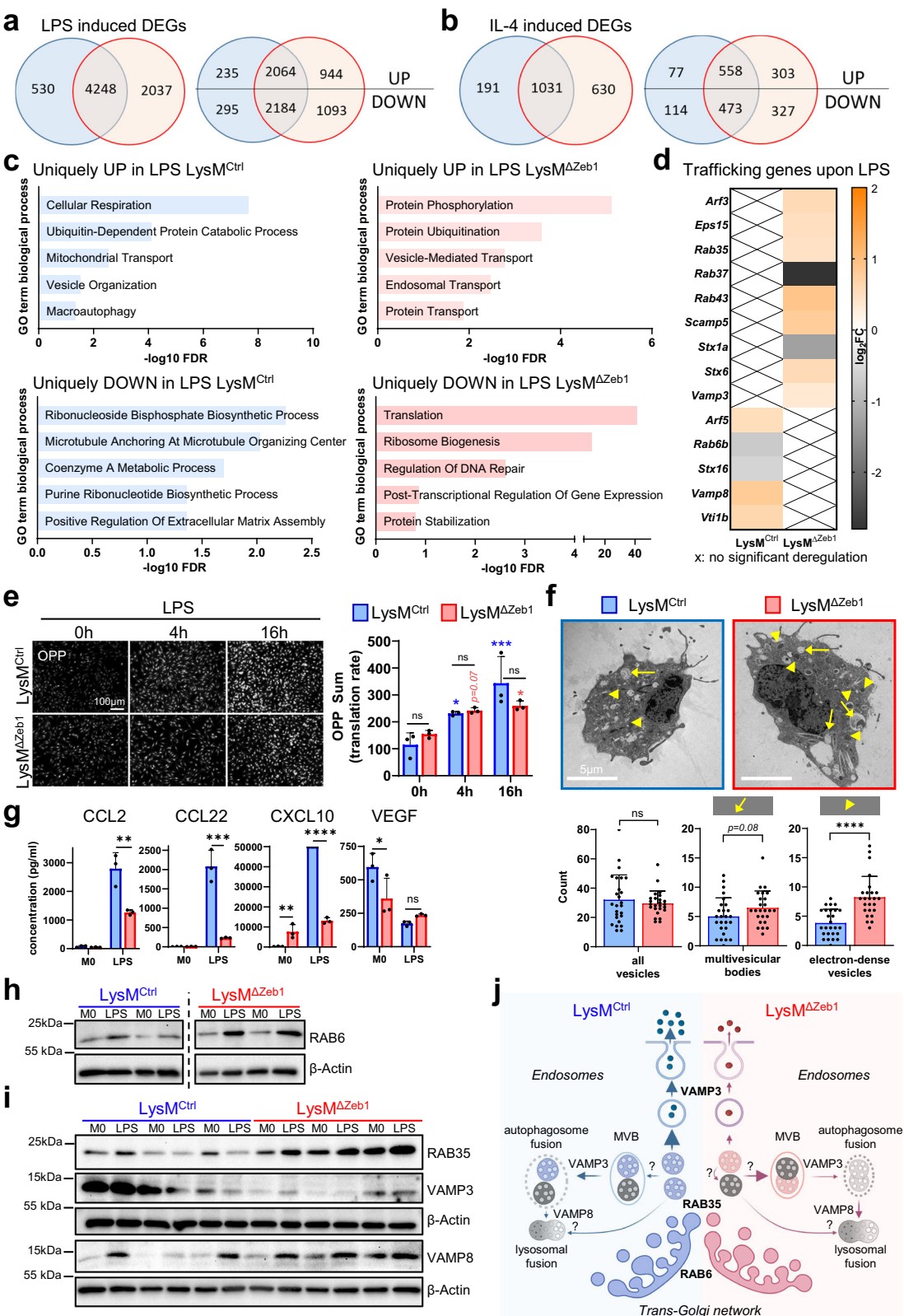

**a** LPS induced DEGs

**b** IL-4 induced DEGs

**c** Uniquely UP in LPS LysM^Ctrl

Uniquely UP in LPS LysM^ΔZeb1

**d** Trafficking genes upon LPS

Uniquely DOWN in LPS LysM^Ctrl

Uniquely DOWN in LPS LysM^ΔZeb1

**e** LPS

**f**

**g** CCL2 CCL22 CXCL10 VEGF

**h**

**i**

**j**

We propose ZEB1 as a regulator of T cell trafficking. From transcriptomics, secretomics, molecular and ultrastructural analyses, we conclude that ZEB1 assists full inflammatory activation and CD8 + T cell recruitment by supporting cytokine expression, translation, trafficking and their secretion. This is in line with the recently reported dual role of ZEB1 in inducing inflammation in macrophages and transitioning into an immunosuppressive state[36]. Intriguingly, the latter was not mainly elicited by regulation of effector genes and cytokines directly. Instead, ZEB1 controlled mitochondrial translation and autophagy, implying the involvement of intracellular membrane fusions and cargo trafficking[36].

In macrophages, defects in synthesis and endosomal processing of cytokines diminish abundance of extracellular cytokines[45,67], as we observed

**Fig. 5 | ZEB1 supports inflammatory macrophage polarization and cytokine secretion by regulating protein biosynthesis, trafficking and cytokine release.** Venn diagrams of differentially expressed genes (DEGs) of LysM$^{Ctrl}$ (blue) and LysM$^{\Delta Zeb1}$ BMDMs (red) after stimulation with LPS (**a**) or IL-4 (**b**) compared to unstimulated in total (left) and divided in up-/downregulated DEGs (right). **c** GO term enrichment analysis for DEGs (FDR < 0.05) uniquely up- or downregulated by LysM$^{\Delta Zeb1}$ BMDMs after LPS stimulation. **d** Log$_2$ fold change of expression of selected trafficking genes after LPS stimulation. X marks no significant deregulation. **e** Representative images and quantification of OPP incorporation of LysM$^{Ctrl}$ and LysM$^{\Delta Zeb1}$ BMDMs with 0 h, 4 h and 16 h LPS pre-stimulation (*n* = 3; means ± SD). **f** Transmission electron microscopy of LPS-stimulated LysM$^{Ctrl}$ and LysM$^{\Delta Zeb1}$ BMDMs with representative images (top) and scoring of indicated vesicle types (bottom) from 26 (LysM$^{Ctrl}$) and 25 (LysM$^{\Delta Zeb1}$) cells derived from 3 independent BMDM lines per genotype. (means ± SD;). **g** Quantification of intracellular cytokine

levels in pg/ml per 10 µg of protein lysates obtained from untreated (M0) or LPS-treated LysM$^{Ctrl}$ and LysM$^{\Delta Zeb1}$ BMDMs using a bead-based immunoassay (*n* = 3; means ± SD). **h–i** Western blots of the indicated trafficking factors and β-Actin as loading control of protein lysates from independent LysM$^{Ctrl}$ and LysM$^{\Delta Zeb1}$ BMDM lines either unstimulated (M0) or stimulated with LPS (\*$p < 0.05$; \*\*$p < 0.01$; \*\*\*$p < 0.001$; \*\*\*\*$p < 0.0001$ ns: not significant; 2-way ANOVA (**e, g**), Mann-Whitney (**f**)). Note that in (**e**), color-coded indicators (blue/red = LysM$^{Ctrl}$/LysM$^{\Delta ZEB1}$) on top of the bars indicate comparisons of a given time point type to the respective time point "0 h" within one genotype and comparisons between LysM$^{Ctrl}$ and LysM$^{\Delta ZEB1}$ at specific time points are marked by the connectors. **j** Hypothetical model for ZEB1-related cargo trafficking in BMDMs leading to reduced cytokine release. (MVBs: Multivesicular bodies). Thickness of the arrows indicates the anticipated trafficking strength. Created in BioRender. Schuhwerk, H. (2025) https://BioRender.com/eymhtmg.

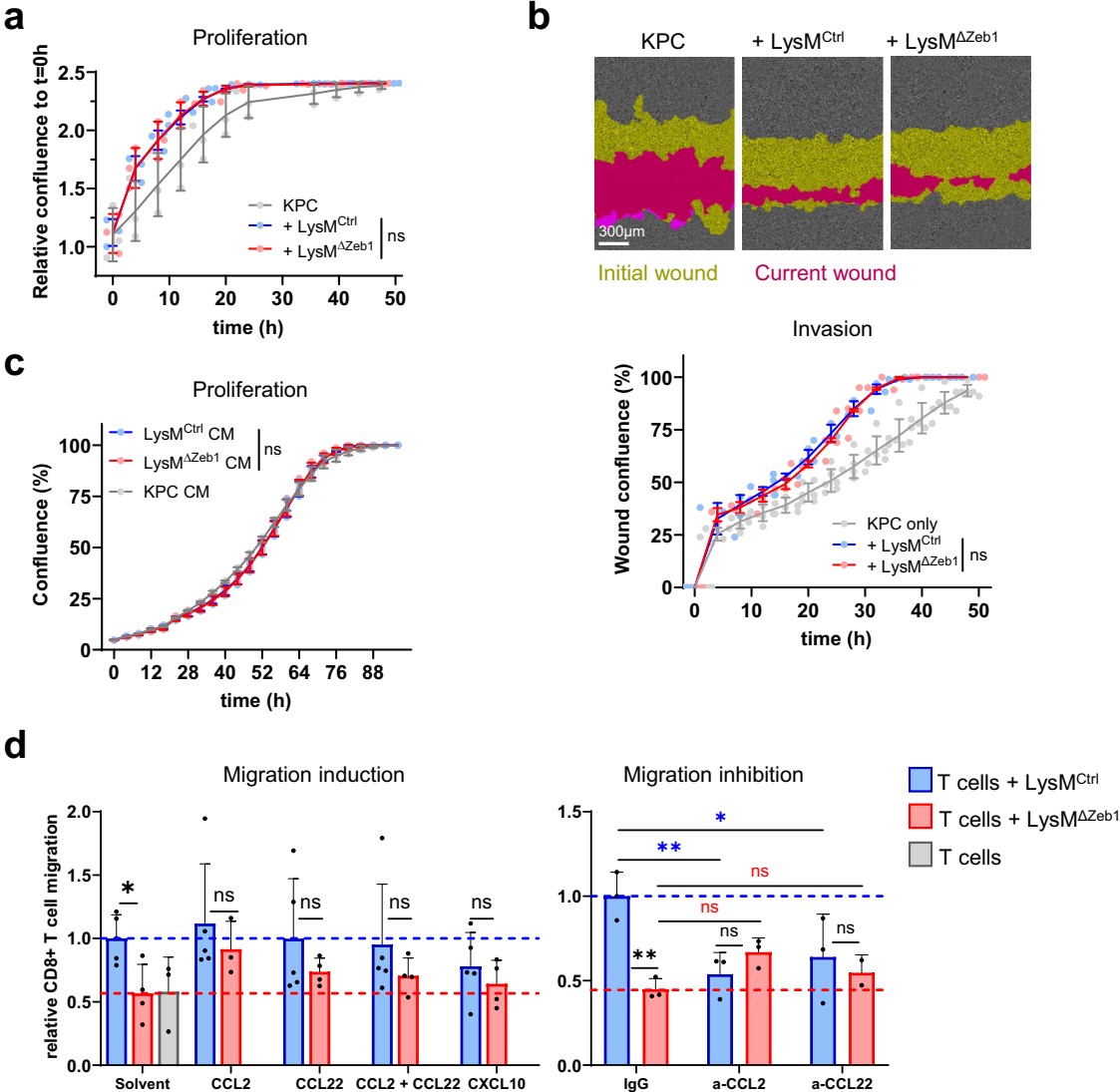

**Fig. 6 | ZEB1 in macrophages has no major direct effect on tumor cells but is important for CCL2- and CCL22-driven recruitment of CD8 + T cells.**
**a** Confluence of KPC cells alone or co-cultured with LysM$^{Ctrl}$ or LysM$^{\Delta Zeb1}$ BMDMs (*n* = 3). **b** Representative images at t = 28 h and quantification over time of KPC cell invasion into a scratch wound without or with co-culture of LysM$^{Ctrl}$ or LysM$^{\Delta Zeb1}$ BMDMs (*n* = 3). **c** Confluence of KPC cells alone or with LysM$^{Ctrl}$ or LysM$^{\Delta Zeb1}$

BMDM conditioned medium (CM) (*n* = 2 KPC CM, *n* = 3 LysM$^{Ctrl}$ and LysM$^{\Delta Zeb1}$ CM). **d** Transwell migration assay of CD8 + T cells alone or towards LysM$^{Ctrl}$ or LysM$^{\Delta Zeb1}$ BMDMs in absence or presence of recombinant CCL2 and CCL22 (left panel, *n* > 3) or absence or presence of anti-CCL2 and anti-CCL22 antibodies (right panel, *n* = 3). Means ± SD; \*$p < 0.05$; \*\*$p < 0.01$; ns: not significant; 2-way ANOVA.

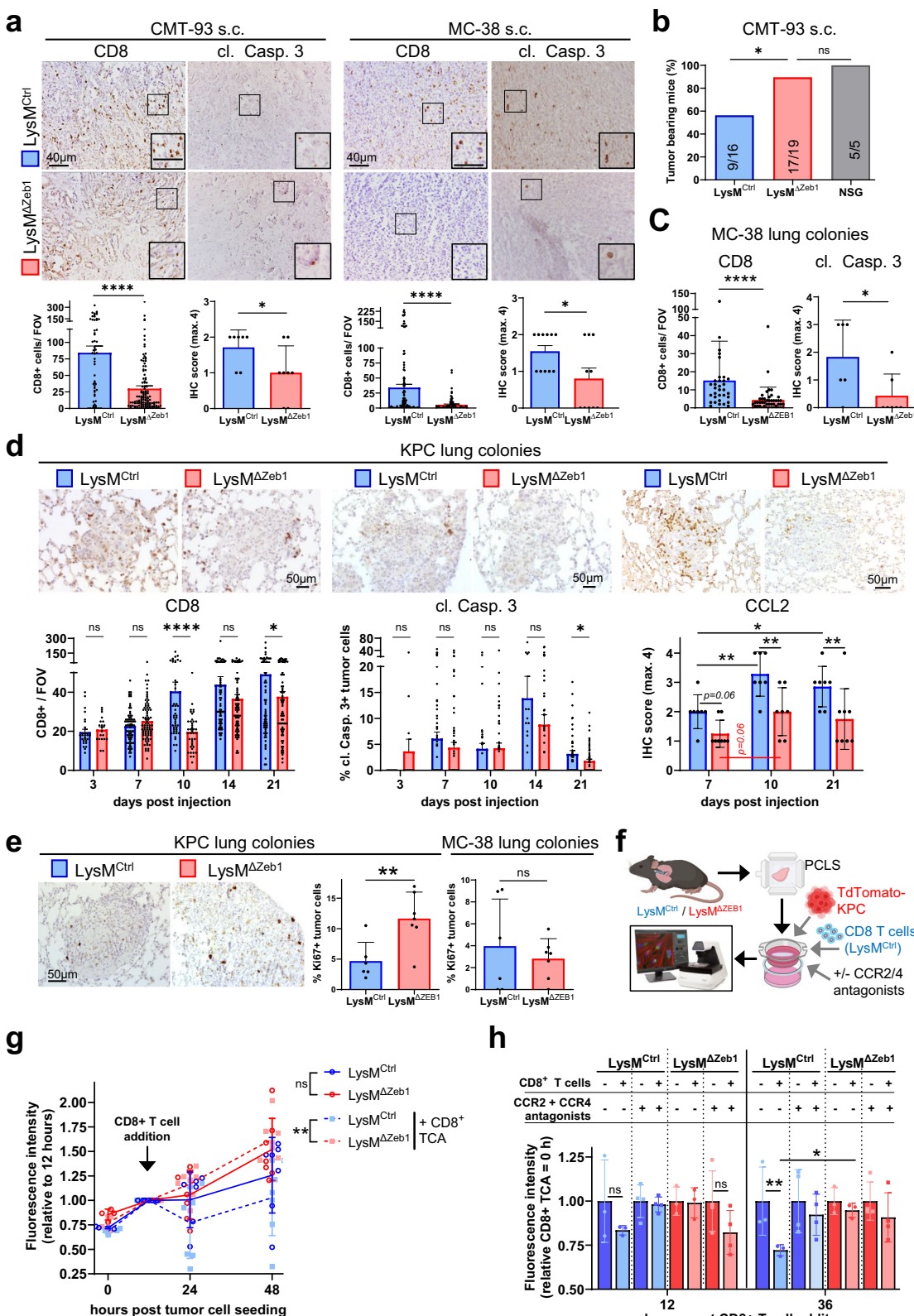

in *Zeb1*-deficient macrophages which accumulated electron-dense and fused intracellular vesicles. This suggests that ZEB1 regulates endosomal cytokine trafficking in macrophages, expanding the described role of ZEB1 in vesicular trafficking in cancer cells[68,69] and membrane biology, as we described recently[62].

Mechanistically, ZEB1 loss in BMDMs caused deregulation of factors crucial for membrane fusions during constitutive secretion[45]. Among these, the protein levels of VAMP3, proposed as pacemaker of constitutive secretion (of TNFα) in macrophages by guiding the release of cargo at the plasma membrane[49], was diminished in the absence of ZEB1. This should

**Fig. 7 | ZEB1 in macrophages promotes abundance of CD8+ cells and concomitant tumor cell killing. a** Representative images and quantification of IHC for CD8 positive (+) ($n > 57$ fields of view (FOVs) of 7 mice) and cleaved Caspase 3+ cells (cl. Casp. 3 IHC scores from $n > 7$ mice) in s.c. CMT-93 (50-58 days-post injection (dpi), see Fig. 3a, b, c) and MC-38 tumors (13-19 dpi, also see Fig. 3d) in LysM^Ctrl and LysM^ΔZeb1 mice. Insets show higher magnification. **b** Percentage of s.c. CMT-93 tumor-bearing LysM^Ctrl, LysM^ΔZeb1 and NSG mice. Number of mice are indicated (tumor-bearing/ total). **c** Quantification of IHCs in MC-38 lung colonies in LysM^Ctrl and LysM^ΔZeb1 mice for CD8+ cells ($n > 33$ tumorous FOVs images of $n > 6$ mice) and cl. Casp. 3+ cells (IHC scores from $n > 6$ mice). **d** IHCs in KPC lung colonies in LysM^Ctrl and LysM^ΔZeb1 mice with representative images and quantification over time of CD8+ cells ($n > 20$ tumorous FOVs per condition; means +SEM; image at 10 dpi), of cl. Casp3+ cells ($n > 10$ tumorous FOVs per condition, means +SEM images at 14 dpi) and of CCL2 (IHC scores of colonies from $n > 7$ mice per condition; means ± SD; images at 10 dpi). **e** Ki67 IHC in indicated lung colonies in LysM^Ctrl and LysM^ΔZeb1 mice with representative image of KPC colonies at 21dpi and quantification (IHC score of colonies from n > 6 mice per condition). **f–h** Precision cut lung slice cultures and therapeutic CD8 + T cell addition (TCA) of splenic CD8 + T cells with experimental setup created with Biorender (**f**), monitoring of KPC-mCherry total fluorescence (per lung area) as surrogate for cell growth on LysM^Ctrl and LysM^ΔZeb1 lung slices over time in the presence / absence of ACT (**g**) as well as the combination of CCR2 and CCR4 antagonists or DMSO as vehicle controls relative to ACT start (**h**). Means ± SD. ****$p < 0.0001$; ***$p < 0.001$ *$p < 0.05$; ns: not significant; Mann-Whitney (**a, c**; for CD8); Qui-square test (**b**); two-tailed $t$-test (**a, c**; for cl. Casp. 3; **e**); 2-way ANOVA (**d, g, h**).

cause a reduction in actual secretion by *Zeb1*-deficient BMDMs. However, intracellular cytokine concentrations were also reduced, which cannot be explained solely by the minor, partially insignificant reduction of mRNA levels and translation in *Zeb1*-deficient BMDMs. From our ultrastructural and molecular analyses, we reason that a strong ZEB1-dependent contributor is vesicle fusions leading to cargo degradation. In fact, the protein levels of RAB35, promoting autophagosome maturation, and of VAMP8, driving autophagosome-lysosome fusions[52,53], were both increased in the absence of ZEB1, suggesting reinforced degradative endosomal trafficking. This interpretation is consistent with unchanged total number of intracellular vesicles. Likewise, as VAMP3 can also mediate fusions between MVBs and autophagosomes[51], its limited abundance in *Zeb1*-deficient BMDMs may impede MVBs breakdown and thus explain their relatively mild yet insignificant accumulation. We noted ambiguity in the case of VEGF, whereby intracellular VEGF levels were lower in *Zeb1*-deficient M0 BMDMs and generally reduced upon LPS stimulation. However, these changes did not correlate with secreted VEGF. Generally, we detected much less intracellular VEGF, but similar amounts of secreted VEGF as compared to the other tested cytokines, implying that trafficking and release of VEGF might be more efficient than of the other tested cytokines, particularly in response to LPS. It should be emphasized that cytokines are processed and released via different, intricate pathways which depend on the cargo, the (stimulatory) context and the cell type[45,49,70,71]. For VEGF trafficking, a complex crosstalk exists, in which autophagy and angiogenesis, including VEGF production, secretion and activity, showed corresponding regulation in various models[72–74]. This may together indicate preferential usage of a distinct secretion pathway for VEGF release in BMDMs, such as 'secretory autophagy', as specifically proposed earlier in the context of age-related macular degeneration[70]. Notably, the trafficking of IL-1β, which utilizes more secretory autophagy[71], was largely unaffected by ZEB1 in BMDMs. Specific effects on the trafficking of VEGF, particularly in the M0 BMDMs, or of IL-1β, might therefore underlie precise alterations in secretory versus degradative fluxes. These nuances, however, remain to be dissected in future mechanistic studies. Although the detailed molecular mechanisms remain elusive, it is intriguing to speculate that ZEB1 restricts retrograde transport of endosomes, their maturation to autophagosomes and lysosomal fusions, thereby mostly maximizing constitutive secretion upon inflammatory stimulation of macrophages. Consistently, ZEB1 was reported to limit phago-lysosomal fusions to enhance antigen export, thereby regulating cytokine production in dendritic cells[31] and antigen cross-presentation, to elicit CD8 + T cell responses[75]. Together with the aforementioned studies in DCs and our recent findings on ZEB1-controlled inflammatory activation in fibroblasts, the present work on macrophages portrays a delicate, cell-type and context-dependent, immunologically relevant facet of ZEB1-linked plasticity[22,23,28,33,36,76,77]. These studies warrant future detailed dissection of ZEB1/EMT-related cellular plasticity in specific immune cell subtypes within different (onco-)immunological niches. Particularly in light of the intracellular heterogeneity in ZEB1 expression even within the same cell subtype, it will be relevant to explore differential functions of the potentially selectively targetable ZEB1-high and ZEB1-low cells therein, similar to our recent study in tumor cells[32].

The role of ZEB1 in cytokine biosynthesis in macrophages was so far mostly limited to measuring mRNA abundance of selected cytokines, without evidence for direct transcriptional regulation[29,36,78]. Going beyond this by conducting intra- and extracellular cytokine profiling, we identify ZEB1 as an important regulator of post-transcriptional processing and secretion of cytokines as (onco-)immunological effectors. Notably, an influence on the release of relevant extracellular vesicles remains elusive. ZEB1 in macrophages augments the secretion of CXCL10, CCL2 and CCL22 in a stimulus-independent manner (*i.e.*, upon LPS and IFNγ treatment), the latter two of which appeared to be important for ZEB1-dependent, BMDM-guided migration of splenic CD8 + T cells in vitro. Consistently, ZEB1-dependent CCL2 abundance correlated with CTL influx and tumor cell killing in metastatic lung colonies.

Given the importance of CXCR3 pathway for CD8 + T cell chemoattraction into solid tumors[58] and the reduced secretion of CXCL10 in particularly LPS-treated LysM^ΔZeb1 BMDMs, it may seem surprising that the migration of CD8 + T cells towards BMDMs could not be elevated by adding CXCL10, but by CCL2 and CCL22, in concentrations levelling out the respective mean differences in secretion by *Zeb1*-proficient versus *Zeb1*-deficient BMDMs. We reason that, in contrast to CCL2 and CCL22, the levels of CXCL10 in LysM^Ctrl and LysM^ΔZeb1 BMDMs both suffice to saturate CXCR3 binding and thereby also the migration of the subset of CXCX3 + CD8 + T cells. While this does not challenge the importance of the CXCR3 system in tumors[58], it may indicate that ZEB1 in BMDMs might be dispensable for the attraction of CXCR3 + CD8 + T cells, but is important for the additional CCL2-CCR2 and CCL22-CCR4 axes, at least in our in vitro system.

In addition to the well-established CXCR3 pathway[58], several studies have shown the chemotactic effect of the CCL2-CCR2 and CCL22-CCR4 axes on CTLs in vitro[79–83] and in vivo, partially with an overall anti-tumor effect[56,57,84–86]. In contrast, also pro-tumorigenic effects of CCL2 and CCL22 in the TME are well established, comprising *e.g.*, monocyte attraction[87,88] as well as recruitment of T helper cells[61,89–91]. This was, however, unaffected in our models and we corroborate chemotaxis of splenic CD8 + T cells by CCL2 and CCL22 in physiologically relevant concentrations in vitro. As metastatic control by infiltrating unprimed CD8 + T cells in PCLS cultures depended on ZEB1 in macrophages as well as the CCL2-CCR2 and/or CCL22-CCR4 axis, we altogether reason that CD8 + T cell specific tumoricidal effects of CCL2 and CCL22 may outweigh pro-tumorigenic and immune-suppressive effects via other cell types in our models. The provided important evidence that at least the efficacy of CD8 + TCA may require the CCL2-CCR2 and/or CCL22-CCR4 axes, is in accordance with earlier observations on CCL2-dependent tumor tropism of adoptively transferred T cells[92]. Importantly, though, with our PCLS and in situ data, we cannot distinguish whether the ZEB1-dependent CTL abundance (in situ) and the CCR2/CCR4-dependent tumor control (PCLS) are regulated by a direct recruitment of CD8 + T cells by CCL2 and/or CCL22 or other alternative mechanisms. These include altered CTL expansion (proliferation) or depletion (exhaustion, cell death) that depend on other macrophage/ (inflammatory) monocyte-linked functions like antigen cross-presentation /-dressing[93], and/or complex cellular communications within the TME.

However, we did not observe a ZEB1-dependent effect on proliferation of CD8 + T cells in vitro when co-cultured with BMDMs, but a ZEB1-dependent modulation of CD8 + T cell migration by CCL2 and CCL22, as discussed above, as well as *CCR2* and *CCR4* expression in *CD8* + T cell subsets in murine and human cancers. Thus, we altogether favor the interpretation that macrophage-derived CCL2 and CCL22 also contribute to the intratumor influx of CCR2+ and/or CCR4 + CD8 + T cell subsets. This said, the individual contribution of the different receptor-ligand axes in distinct immunological niches on tumor-infiltrating lymphocyte responses, particularly upon therapeutic interventions still remains insufficiently understood and warrants further detailed studies.

In clinical trials, inhibitory therapeutic targeting of the CCL2 and CCL22 axes were often unfavorable for patients[61,85,89], reinforcing an overall anti-tumorigenic effect of CCL2 and CCL22 at least in certain tumor settings[94]. Our data on CD8 + T cell abundances in tumorous tissues and CCR2/4-dependent tumor control via CD8 + TCA in PCLS cultures as well as other studies concordantly describing homing of CCR4+ CTLs to diseased skin or lung[95–97], jointly provide an encouraging therapeutic prospect to improve T cell influx to metastasized lungs by local CCL2 or CCL22 administration.

Collectively, our study identified a new tumor-suppressive function of ZEB1 in macrophages, exposing an important immunological context provided by ZEB1 in TAMs. Hence, our work re-appeals for caution in designing anti-TAM therapies and points to a potential therapeutic window of opportunity for topical CCL2 or CCL22 supplementation in patients bearing micro-metastatic lungs.

## Methods

### Animal experiments

We have complied with all relevant ethical regulations for animal use. Animal husbandry and the experiments were approved by the committee of ethics of animal experiments of the state of Bavaria (Regierung von Unterfranken, Würzburg; TS-18/14, TS-30-2021, 55.2-DMS-2532-2-270, -2-1234) and performed according to the European Animal Welfare laws and guidelines. Animals were kept on a 12:12 h light-dark cycle and provided with food and water ad libitum in the animal facilities of the Friedrich-Alexander University of Erlangen-Nürnberg. *Zeb1*$^{flox/flox}$ conditional Zeb1 knockout mice, *LysM*$^{Cre}$ (*Lyz2*$^{Cre}$), *Rosa26*$^{mT/mG}$, *Cx3cr1-Cre* and *Ly6g*$^{Cre-Tom}$ mice have been described previously[36–40]. All mouse strains were kept on C57BL/6 background. *Zeb1*$^{flox/flox}$ mice were crossed with *Lyz2*$^{Cre/+}$ mice to obtain *Zeb1*$^{flox/+}$;*Lyz2*$^{Cre/+}$ which were crossed *inter se* to obtain *Zeb1*$^{flox/flox}$;*Lyz2*$^{+/+}$ ('LysM$^{Ctrl}$') and *Zeb1*$^{flox/flox}$;*Lyz2*$^{Ki/+}$ ('LysM$^{ΔZeb1}$') mice. *Zeb1*$^{flox/flox}$ mice harboring either one *Cx3cr1-Cre* or one *Ly6g*$^{Cre-Tom}$ allele instead of *LysM*$^{Cre}$ were generated by breeding as described above for 'LysM$^{Ctrl}$' and 'LysM$^{ΔZeb1}$' mice. The *Rosa26*$^{mT/mG}$ allele was crossed in as necessary and all mice were PCR genotyped for all alleles (see Supplementary Table 1 for oligonucleotides). NOD.Cg-Prkdc$^{scid}$ Il2rg$^{tm1Wjl/SzJ}$ (Nod-Scid-gamma, NSG) were bred in-house. In general, age-matched littermates of both sexes were used for experiments and the order of treatments, *i.e.*, injections of cells and luciferin for bioluminescence imaging, as well as of in vivo measurements were randomized. Where possible, the investigators were blinded for collecting data on tumor and colony growth as well as for manual scoring of immunohistochemistry stainings. No blinding was applied for other experiments since information about the groups was required for correct execution and data analysis.

### Tumor cell culture and retroviral transduction

Tumor cell lines and their genetically modified derivatives (tdTomato, Luciferase, as indicated) were cultured in DMEM containing 10% FBS and 1% P/S in a humidified incubator (5% $CO_2$) at 37 °C. All cell lines used in this study were routinely tested for mycoplasma contamination using the Mycoplasma detection kit (Jena Bioscience, PP-401). None of the cell lines is listed in the database of commonly misidentified cell lines, ICLAC.

Tumor cell transduction for constitutive luciferase and tdTomato expression was carried out by ecotropic retroviral infection. The pLib_EF1A_nlstdTomato-2A-puro plasmid was generated by HiFi DNA assembly (Cell Signaling, E2621) according to the manufacturer's instructions. In brief, PCR fragments containing pLib vector backbone (Clontech) and EF1A promoter, a cassette coding for tdTomato with an N-terminal nuclear localization sequence and lack of a stop codon and a puromycin cassette with an N-terminal self-cleaving T2A peptide were amplified by KOD polymerase (Merck, 71085) and subjected to Gibson assembly.

For retroviral transduction, $2 × 10^6$ PlatE cells were plated into 10 cm plates in DMEM/10% FBS, transfected with 8 µg of pLib_E-F1A_LuciNeo (kindly provided by Ralf Graeser, ProQinase, Freiburg) or pLib_EF1A_nlstdTomato-2A-puro using FugeneHD (Promega). Medium was exchanged after 4–8 h. Virus-containing supernatant of PlatE cells (virus-SN) was harvested and filtered through 0.2 µm membranes after 2 days, and directly used for viral transduction of cancer cell lines. $1 × 10^5$ KPC and MC-38 cells were plated one day before transduction into 6-wells and transduced by replacing the medium with 2 ml of virus-SN supplemented with 8 µg/ml polybrene. After 3-4 h of incubation, the transduction medium was replaced by standard medium. Starting from the day after, cells were selected by G418 (KPC661-Luc, MC-38-Luc; 175-300 µg/ml for 7 days) or puromycin (KPC661-nTom; 1.5–1.75 µg/ml for 4 days), where appropriate. tdTomato-positive cells were enriched by FACS and expanded.

### Subcutaneous tumor allografts

Cancer cells in 100 µl PBS (containing $1 × 10^6$ CMT-93, $1 × 10^5$ KPC661 or MC-38 as indicated) were subcutaneously injected into flanks of mice. Mouse weight and tumor size were measured by calipering. The maximal permitted tumor volume was 1 cm$^3$. This limit was exceeded in none of the experiments. Mice were sacrificed when their tumor approached a critical size or ulcerated.

### Lung colonization and bioluminescence imaging

100 µl PBS containing $2 × 10^5$ MC-38, $2 × 10^5$ KPC661-Luc or $2 × 10^5$ KPC661-nTom was injected into the tail vein of mice. Mice were monitored at least twice per week and sacrificed at indicated time points. Growth of tumors with Luciferase reporter was monitored by bioluminescence imaging. 10 min after injection of 100 µl D-Luciferin-Na-salt (25 mg/ml, PJK, 102133) in PBS s.c. in anesthetized mice, background-corrected bioluminescence signal was measured in the IVIS Spectrum In Vivo Imaging System (Perkin Elmer). Lung ex vivo bioluminescence was measured in the IVIS system directly after sacrifice. BLI images were analyzed using the Living Image software.

### Histology and immunohistochemistry (IHC)

Organs or tumors were collected and either fixed in 4% paraformaldehyde, and embedded in paraffin (FFPE) or cryopreserved by freshly freezing the tissue in Tissue-Tek O.C.T. Compound (Sakura Finetek, 4583). Lungs were inflated with PBS or via intratracheal injection of 60% Tissue-Tek O.C.T. Compound in PBS prior to FFPE or cryopreservation, respectively.

For hematoxylin and eosin (H&E) staining, FFPE slides were deparaffinized and rehydrated in Roti-histol ($4 × 15$ min), isopropanol ($2 × 5$ min) and ethanol ($2 × 5$ min 96% ethanol, 5 min 80% ethanol, 5 min 70% ethanol) and washed 5 min in deionized $H_2O$. Slides were stained in 1:10 hematoxylin solution in Milipore $H_2O$ and counterstained with 0.2% eosin in 100% ethanol. Slides were dehydrated in an ethanol row (80% ethanol, 2×100% ethanol), followed by 2x isopropanol and xylol. For mounting, Roti-Histokitt was used.

IHC was performed as previously described whereby IHC scores were estimated from the cellularity of (DAB-)staining positive cells, with none (0), rare (1), few (2), several (3) and many/abundant (4)[33]. The antibodies used for IHC are provided in Supplementary Table 1.

For immunofluorescence (IF), cryopreserved tissue on glass slides were fixed using 4% PFA for 15 min at RT (all except Ly6C) followed by 2 washes in PBS at RT, then permeabilized and quenched for 10 min in PBS

containing 0.1 M glycine and 0.1% Triton X-100. Following 3x washing in PBS, tissues or cells were blocked in blocking buffer (3% BSA in PBS) for 30 min. Slides were incubated with primary antibodies, diluted in blocking buffer, overnight at 4 °C. The following day, slides were washed 3x in PBS and incubated with fluorophore-coupled secondary antibodies for 1 h. For Ly6C-IF, specimens were fixed using ice-cold 100% acetone at −20 °C for 8 min, followed by 2 washes in PBS, blocking for 30 min (including mouse Fc block) and incubation with Alexa647-coupled anti-mouse Ly6C antibody in PBS for 30 min at RT. Details about all antibodies are provided in Supplementary Table 1. Following washing 3x in PBS, slides were incubated 15 min in 1 µg/ml DAPI working solution. Slides were washed in 3x PBS and mounted with CitiFluor™. Images were acquired using a Leica DM5500B microscope. For analysis, fields of view (FOV) containing tumor cells, micro-, moderately sized and/or macro-colonies, were acquired using a Leica DM5500B microscope and automated quantification of these was performed using ImageJ. Briefly, images were formatted to RGB, deconvoluted for RGB and background subtracted for thresholding of positive cells. For analysis of Ly6C, CellProfiler[98] was used to threshold the Ly6C and DAPI channel images.

## Flow cytometry

After opening the abdomen and thorax, about 300 µl blood was slowly extracted from the right ventricle of the heart with an EDTA flushed syringe and added to 14 ml Hank´s solution. Samples were centrifuged for 10 min at 1400 rpm and all but 3 ml supernatant was removed. After blood harvesting, mice were perfused with 0.9% NaCl and organs collected in ice-cold PBS. Organs were minced with scissors and incubated in 10 ml digestion mix (DMEM/ F-12 containing 0.05% Collagenase D (Sigma-Aldrich, 11088858001), 0.3% Dispase II (Sigma-Aldrich, D4693) and 0.05% DNase I (Sigma-Aldrich, 10104159001) for 25 min at 37 °C with gentle agitation. Digestion was stopped by adding 30 ml ice-cold PBS, the samples were filtered through a 70 µm cell strainer and centrifuged at 1200 rpm at 4 °C for 5 min. The cell pellet was resuspended in 3 ml ACK Lysis buffer. After 3 min, erythrocyte lysis was stopped with 27 ml PBS and the samples centrifuged for 5 min at 1200 rpm at 4 °C. $1 \times 10^6$ cells were blocked in 50 µl TruStain FcX PLUS in PBS for 15 min a RT. 50 µl of 2x FACS antibody mix was added and incubated for 20 min at 4 °C in the dark. Details about all antibodies are provided in Supplementary Table 1. Following centrifugation for 5 min at 1200 rpm at 4 °C, cell pellets were resuspended in 500 µl FACS buffer. Samples were analyzed in the CytoFlex Analyzer (Beckman Coulter). Data was analyzed using the CytExpert or Kaluza software (Beckman Coulter).

For flow cytometry cross-tissue UMAP clustering, samples were stained in 50 µL of FACS buffer (PBS + 2% FCS) containing purified FC blocking antibodies (clone 2.4G2 and clone 9E9) as well as the monoclonal biotin coupled antibodies for a total of 20 min at 4 °C. Next, samples were washed three times with FACS buffer. Subsequently, cells were stained with monoclonal fluorescently-labeled antibodies and streptavidin-BUV496. Details about all antibodies are provided in Supplementary Table 1. After three washing steps, samples were supplemented with FACS buffer containing DAPI in a final concentration of 50 ng/mL and analyzed using a LSR Fortessa SORP (BD). Data was analyzed using FlowJo (BD). 5000 (myeloid panel) or 2150 (lymphoid panel) cells per tissue were randomly selected from the pool of CD45+ cells, combined to form an analysis sample and UMAP was performed. The individual samples were again separated by tissue and the populations identified by gating.

Surface staining of CD192, CD194 and CD183 on splenic CD8 + T cells (enriched via magnetic bead separation as described below) was performed as described above but for 30 min at 37 °C. For details on antibodies see Supplementary Table 1. Samples were analyzed by full spectrum flow cytometry using a Northern Lights™ cytometer (Cytek Biosciences). Following spectral unmixing using the built-in acquisition software (Cytek Bioscience), the data was analyzed using FlowJo (BD). For flow cytometry of intracellular ZEB1, organs were collected as described above and isolated using the Tumor Dissociation Kit (130-096-730,

Miltenyi Biotec) according to the manufacturer's instructions. Following live/dead discrimination using Zombie NIR (423105, Biolegend), surface and intracellular staining after Fc block was performed using the FoxP3 Staining Buffer Set (130-093-142, Miltenyi Biotec) according to the manufacturer's instructions, using the antibodies listed in Supplementary Table 1. Data was acquired and analyzed by full spectrum flow cytometry as described above.

## Bone marrow-derived macrophage (BMDM) isolation and culture

BMDMs were isolated as progenitors, differentiated and maintained in culture as described previously in ref. 99. On day 0, bone marrow of mouse femur and tibia bones was flushed with 10 ml PBS using a 26 G cannula. Following centrifugation for 5 min at 1200 rpm, the cell pellet was resuspended in 3 ml ACK Lysis buffer. After 3 min, erythrocyte lysis was stopped with 27 ml PBS and cell suspension was filtered using a 70 µm cell strainer. Following centrifugation for 5 min at 1200 rpm, the cell pellet was resuspended in 10 ml BMDM medium (DMEM containing 10% FBS, 10% L-929 supernatant and 1% P/S), plated in a 10 cm petri dish and cultured in a humidified incubator (5% $CO_2$) at 37 °C. On day 1, the supernatant containing BMDMs was collected and centrifuged for 5 min at 1200 rpm and BMDMs were seeded in BMDM medium ($5$-$6 \times 10^6$, $0.3 \times 10^6$, $0.1 \times 10^6$ and $0.015 \times 10^6$ cells/ well or dish in 10 cm dishes, 6-well, 24-well and 96-well plates, respectively). On day 3, additional BMDM medium was added. When required, cells were sorted according to mT/mG fluorophores on day 4 by FACS. To this end, BMDMs were detached by incubation with Cell-Stripper (Corning, 25-056-CI) for 15 min at 37 °C, followed by 5 min at 4 °C. After sorting (MoFlo XDP, Beckman Coulter), cells were plated as described above. When required, BMDMs were stimulated with 10 ng/ml LPS (Sigma-Aldrich, L2630), 20 ng/ ml IL-4 (BioLegend, 574304) or 20 ng/ ml IFNγ (R&D Systems, 485-MI) for 24 h, unless indicated otherwise.

For BMDM confluence assays, BMDMs were cultured in the IncuCyte ZOOM incubator (Sartorius) in a 24-well plate from day 1 on. Images of each well were acquired automatically every 4 h by the IncuCyte ZOOM on-board camera and then analyzed using the IncuCyte ZOOM 2018A software.

For BMDM cell death assay, BMDMs were cultured with 1:1000 SYTOX Green Ready Flow Reagent (Invitrogen, R37168) in the IncuCyte ZOOM incubator in a 96-well plate from day 6 on. Cells were cultured and monitored using IncuCyte. Image acquisition and analysis was performed as described above.

## Phagocytosis assays

For bioparticle assays, 100 ng/ml pHrodo Green E. coli BioParticles (Thermo Fisher Scientific, P35366) were added to BMDMs, on day 7. Cells were cultured in the IncuCyte ZOOM incubator. Image acquisition and analysis was performed as described above.

For phagocytosis of necrotic cell debris, KPC661-Luc cells were labeled with CellVue Claret (Sigma-Aldrich, MIDICLARET) following manufacturer´s instructions. $4 \times 10^6$ cells/ ml were incubated in a water bath at 60 °C for 30 min to create necrotic bait cells. BMDMs (5 div) were stimulated with 10 ng/ml LPS or 20 ng/ ml IL-4. On day 6, stimuli were removed and 100 µl of necrotic cells were added to each well for indicated time points. BMDMs were washed 4x with ice-cold PBS and incubated 15 min at 37 °C with CellStripper. Detached cells were collected in FACS tubes, centrifuged at 1200 rpm for 5 min and resuspended in 500 µl FACS buffer. Data was acquired in the CytoFlex analyzer (Beckman Coulter) and analyzed using the CytExpert or Kaluza software (Beckman Coulter).

## Immunofluorescence staining and protein translation labeling via OPP

For mT/mG reporter visualization in cultured BMDMs, BMDMs were washed, fixed with 4% PFA for 15 min and permeabilized and quenched for 10 min in PBS containing 0.1 M glycine and 0.1% Triton X-100. Following three washes in PBS, slides were incubated 15 min in 1 µg/ml DAPI in PBS.

After three washes in PBS, slides were mounted on glass slides using CitiFluor[TM].

Protein translation was detected with the "Click-iT Plus OPP Alexa Fluor 488 Proteinsynthese-Assay-Kit" (Thermo Fisher Scientific, C10456) according to manufacturer´s instructions. Briefly, BMDMs in 96-well plates were stimulated with LPS for the indicated timepoints followed by 2 h of 10 µM OPP incubation before fixation and quenching as described above. Click reaction was carried out for 1 h at room temperature. Subsequently, IF staining for ZEB1 was performed by incubating primary antibody over night at 4 °C and secondary antibody for 1 h at room temperature (details about antibodies are provided in Supplementary Table 1). DAPI was used as counterstain (1-2 µg/ml) and cells were kept in PBS until image acquisition. 20 images per well were taken using the EVOS M7000 microscope (Invitrogen). Images were analyzed using CellProfiler[98], employing previously optimized pipelines[32]. Remaining strongly Zeb1-positive cells in LysM[ΔZeb1] BMDM cultures were excluded from the analysis and a minimum of 5000 cells were analyzed per condition in 3 independently isolated LysM[Ctrl] and LysM[ΔZeb1] BMDM lines.

### Transmission electron microscopy (TEM)

TEM on BMBMs, differentiated and treated as indicated until day 6, was performed as described previously[100]. After embedding, sectioning and staining, samples were transferred into a JEOL1400Plus transmission electron microscope (JEOL, Garching, Germany) and imaged at a magnification of 2500x to obtain pictures showing one entire cell. Intracellular vesicles were counted from the indicated number of cells derived from the indicated number of independent BMDM lines.

### Analysis of gene expression

For RNA isolation, cDNA synthesis and quantitative reverse transcriptase PCR, cells were washed with PBS and lysed with 350 µl RLT Plus from the RNeasy Plus Mini Kit (QIAGEN, 74136) before RNA isolation following manufacturer´s instructions. cDNA was synthesized using the RevertAid First Strand cDNA synthesis Kit (Thermo Fisher Scientific, K1622) according to manufacturer´s instructions.

qRT-PCR was performed in triplicates in 384-well plates using primers and Roche universal probe library (Roche, 04869877001) with TaqMan Universal MasterMix II (Thermo Fisher, 4440044) and LightCycler 480 II (Roche). Alternatively, mRNA expression levels were measured with the RT[2] Profiler PCR Array with a custom designed panel (QIAGEN, 3445261) using the Power SYBR Green PCR Master Mix (QIAGEN, 4367659) according to manufacturer´s instructions. Details about primers for qPCR are provided in Supplementary Table 1.

### Bulk RNA sequencing analysis

Bulk RNA sequencing of BMDM RNA ($n = 3$ per condition) was performed by Novogene using poly(A) enrichment library preparation protocol and paired-end sequencing (PE150). The preprocessing of raw RNA-seq data (FASTQ files) was performed using the nf-core RNA-seq pipeline v.3.8.11[101]. In particular, reads were adapter- and quality-trimmed using Trim Galore v.0.6.7 (https://github.com/FelixKrueger/TrimGalore). The reads were then mapped to the Ensembl mouse genome assembly GRCm39 (release 107) using STAR v.2.7.10a[102]. For transcript-level read counting, Salmon v.1.5.2[103] was employed, relying on the Ensembl gene annotation file release 107. To generate gene-level counts, Salmon's transcript-level quantification files were processed using the R-package tximport v.1.22[104] within R v.4.0.3. Differential expression analysis for the RNA-seq data was performed using the DESeq2 package v.1.34.0[105]. As DESeq2 design formula ~*litter* + *group* was utilized, where the *group* factor was created by merging together genotype (LysM[Ctrl]/LysM[ΔZeb1]) and treatment (LPS/IL-4). Fold change shrinkage was performed relying on the "ashr" method[106], made available within the DESeq2 package. GO term analysis was performed using Enrichr[107]. For this, for all significantly differentially expressed genes (DEGs) from the indicated comparisons (adjusted $p$ value: FDR < 0.05) were used and the -log10 (FDR) values plotted with GraphPad Prism.

The numbers of identified DEGs for the indicated comparisons that were used for GO term analyses are indicated in the Venn-like diagram in Fig. 5a, b.

### Analysis of single-cell RNA sequencing data

For pan-cancer single cell sequencing analysis of myeloid cells, the online tool of[108] was utilized (http://panmyeloid.cancer-pku.cn). The dataset 'Pan_cancer scanorama_corrected' was selected and filtered for 'Macro', 'Mono', 'Monolike' and 'Myleoid' clusters. The Zeb1 feature plot (on UMAP clusters) was directly retrieved from the portal.

For murine pancreatic cancers, the FASTQ files of[109] were aligned to the reference transcriptome CellRanger 7.1.0[110] and subjected to further processing using Seurat 5.0., involving quality control (number of genes detected in each cell > 300; total number of molecules detected per cell >500; mitochondrial read count ratio cutoff <25%; Haemoglobin ratio <10%), normalization of counts using Seurat´s 'SCTransform', integration of the three datasets using Seurat´s 'RPCAIntegration', and clustering. Next, DECORDER annotation was used[111] to annotate the individual cells, of which immune cells were UMAP sub-clustered. 'FindAllMarkers' (top 50 DEGs of a given cluster versus all other cell clusters) allowed annotation of individual immune cell types/clusters (B cells: CD2 + ;Bkl + ;Fcmr + , T cells: Gata2 + ;Itk + ; dendritic cells: Dcstamp + ;Slamf9 + , macrophages/granulocytes: Tlr4 + ;Ccrl2 + ). The feature plot in Fig. 1b was created using 'SCPubr' (https://enblacar.github.io/SCpubr-book/closing_remarks/Citation.html).

For single cell RNA sequencing analysis of myeloid cells, monocytes, macrophages and tumor cells as well as CD8 + T cells in human cancer, the built-in online tool from the Broad Institute´s 'single cell portal' (https://singlecell.broadinstitute.org/single_cell) was used to explore the dataset from[112] as well as *SCP1891*, *SCP2393* and *SCP1039*. Plots and cell type annotations were directly retrieved from the portal, using the dataset-specific annotations and filtering options. For myeloid subtype clustering, cell types were filtered for 'myeloid' cells (in Fig. 1c, Zeb1 feature plot), and, among those, for 'macrophage-like', 'monocytes' and 'granulocytes' (in Fig. 1d for comparison), using the respective available filters. Filtering for 'CD8 + T cells' and the 'macrophages' and 'monocytes' compartment was performed analogously, using the dataset-specific annotations and filtering options.

For analysis of murine tumor-infiltrating T cell subsets from tumor-draining lymph nodes, the '*Swiss Portal for Immune Cell Analysis*' was used (https://spica.unil.ch)[59] and plots as well as cell type annotations were directly retrieved from the portal.

For assessing the correlation of *CD8 +* T cell infiltration with cytokine receptor expression in human cancer patients by deconvoluting from bulk RNA sequencing data using *TIMER2.0*, the online built-in tool was used (http://timer.cistrome.org/)[60]. Correlation coefficients in specific cancer entities were thereby directly retrieved from the portal and plotted as a heatmap in GraphPadPrism.

### Protein isolation and western blot analysis

Protein was isolated as previously described[32]. Briefly, cells were washed twice with PBS. Triple lysis buffer containing 150 mM NaCl, 50 mM Tris-HCl pH 8.0, 0.5% Na-Desoxycholate (w/v), 0.1% SDS (v/v), 1% NP40 (v/v), 1 mM PMSF, 1x complete protease inhibitor cocktail (Roche, 04693132001) and 1x PhosStop (Roche, 4906837001) was added, cells scraped off and transferred to a 1.5 ml tube for cell lysis before clearance by centrifugation and protein concentration measurement using the Pierce BCA Protein Assay Kit (Thermo Fisher Scientific, 23225) following manufacturer´s instructions and the FLUOstar Omega reader. After SDS-PAGE, proteins were transferred onto nitrocellulose (Roth, 4685.1) or methanol-activated PVDF membranes (1620177, Bio-Rad) by wet blot transfer. Membranes were subjected to immunoblotting using the antibodies defined in Supplementary Table 1 and ECL-based signal revelation according to the manufacturer´s instructions (Western Lightning Plus-ECL, NEL103001EA, Perkin Elmer).

## Cytokine protein arrays

The Proteome Profiler Mouse XL Cytokine Array (R&D systems; ARY028) kit was used according to manufacturer´s instructions in order to detect secreted proteins in 500 µl cell culture supernatant. Briefly, after overnight incubation with the supernatants on the kit´s membranes, washing, incubation with the kit´s reaction cocktail, washing and final signal revelation, pixel density was measured and background-subtracted using ImageLab (BioRad). Subsequently, the means of 2 duplicate spots (A) for each detected cytokine on a membrane was normalized against the mean of positive control/ reference spots (B) on the same membrane. The obtained values for each cytokine from supernatants from ZEB1-deficient BMDMs were then divided by the values of the respective cytokines from ZEB1-proficient BMDM littermates.

Quantitative analysis of secreted proteins was performed using a bead-based LEGENDplex Custom Mouse 14-plex Panel (BioLegend) according to the manufacturers´ instructions. 10 µg of protein lysate, extracted as described above, was used for intracellular LEGENDplex.

## Co-culture assays and conditioned medium transfer

For proliferation assays, BMDMs (7 div) and KPC661-Luc cells were seeded in a 1:3 ratio to a total number of $8 \times 10^4$ cells/ well into a 24 well plate. For the invasion assay, BMDMs (7 div) and KPC661-Luc cells were seeded in a 1:3 ratio to a total number of $1.6 \times 10^4$ cells/ 96-well. After 4 h, monolayers were scratched with the IncuCyte 96-Well Scratch Woundmaker. Cells were washed twice with pre-warmed culture medium and 50 µl 8 mg/ ml matrigel was added to each well. Following incubation in a humidified incubator (5% $CO_2$) at 37 °C for 30 min, BMDM medium was replenished. Cells were cultured and monitored using IncuCyte. Image acquisition and analysis was performed as described above.

For assays involving conditioned medium transfer, 200 Luciferase-expressing KPC661 cells in 200 µl/ well were plated in a 96 well plate. Conditioned medium from BMDMs or KPC661-Luc cells was added every other day by removal of 100 µl medium and addition of 100 µl conditioned medium. Cells were cultured and monitored using IncuCyte. Image acquisition and analysis was performed as described above.

## T cell isolation, enrichment of CD8 + T cells and T cell-involving assays

Mouse spleens were smashed through a 70 µm cell strainer. Following centrifugation at 1200 rpm for 5 min at 4 °C, the splenocyte pellet was resuspended in 3 ml ACK lysis buffer for 3 min. To stop erythrocyte lysis, 27 ml ice-cold PBS was added and the samples centrifuged at 1200 rpm for 5 min at 4 °C. Splenocytes were resuspended in 3 ml MojoSort buffer and enriched for CD8 + T cells using the MojoSort Mouse CD8 T Cell Isolation Kit (BioLegend, 480035), following manufacturer´s instructions.

For the T cell attraction assays, 500 µl of double concentrated cytokines or antibodies in BMDM medium were added to BMDMs (5 div) in 500µl BMDM medium. Final concentrations of cytokines were 1.93 ng/ ml CCL2 (R&D Systems, 479-JE-050), 0.18 ng/ ml CCL22 (R&D Systems, 439-MD-025) and 1.74 ng/ml CXCL10 (R&D Systems, 466-CR-50-CF). IgG (Diagenode, C15410206), anti-CCL2 (Novus Biologicals, NBP1-07035SS) and anti-CCL22 (abcam, ab124768) were used at 5 µg/ ml. $5 \times 10^5$ isolated T cells in BMDM medium with 0.1% FBS were seeded in the upper chamber of transwells with 3 µm pore size (Greiner, 662630) placed in each well and incubated for 24 h at 37 °C in a humidified incubator (5% $CO_2$). The semi-adherent T cells were harvested from the lower chamber by harsh pipetting and centrifuged at 1200 rpm for 10 min. T cells were resuspended in 50 µl ice-cold PBS and counted.

For the T cell activation assay, isolated T cells were labeled with CellVue Claret Labeling following manufacturer´s instructions. 3 µg/ml anti-CD3 (Thermo Fisher Scientific, 16-0032-82) and 5 µg/ ml anti-CD28 (Thermo Fisher Scientific, 16-0281-82) antibodies were added to $2 \times 10^6$ T cells/ ml in RPMI medium. $2 \times 10^5$ T cells/ well were added to BMDMs (6 div) and incubated for 3 days at 37 °C in a humidified incubator (5% $CO_2$). After the co-cultures with BMDMs, the CD8 T cells were harvested, pelleted by

centrifugation at 1500 rpm at 4 °C for 10 min, resuspended in 200 µl ice-cold FACS buffer and transferred to a fresh 96-well plate. T cells were washed with ice-cold FACS buffer and fluorescence data was acquired in the CytoFlex analyzer (Beckman Coulter) and analyzed using the CytExpert or Kaluza software (Beckman Coulter). The fraction of CD8 + T cells that have undergone cell division(s), as determined from the CellVue claret intensities, has been plotted.

## Precision-cut lung slice (PCLS) cultures and therapeutic CD8 + T cell addition (CD8 + TCA)

PCLS were generated from lungs obtained from naïve LysM^Ctrl and LysM^ΔZeb1 mice and cultured as described previously in ref. 62. Briefly, PCLS were sectioned using a vibratome VT1200S (Leica) and after 12 h culturing in DMEM-F12 (Thermo Fisher Scientific, 11320033), single lung slices were incubated with $2 \times 10^5$ KPC-tdTomato cells in low-attachment 48-well plates (Nunc, Thermo Fisher Scientific, 150787) for 4 h, then transferred into fresh plates. 12 h after tumor cell seeding, CD8 + TCA was performed by adding $1 \times 10^5$ CD8 + T cells per PCLS (*i.e.*, per well) which were enriched from freshly isolated LysM^Ctrl splenocytes and labelled with CellVue Claret (Sigma-Aldrich, MIDICLARET) as described above. For treatments with antagonists, slices were pretreated with inhibitors of CCR2 (Sigma-Aldrich, 227016) and CCR4 (MedChemExpress, HY-157453) at final concentrations of 0,05 µM and 0,2 µM, respectively, or DMSO as control, for 12 h before seeding of tumor cells in media containing the same concentration of the drugs or DMSO. After 12 h, CD8 + TCA was performed as described above but using media containing the drugs or DMSO. Imaging was performed using the EVOS system (M7000 Thermo Fisher) after tumor cell seeding (*i.e.*, after transfer into fresh plates) and at the indicated time points. Fluorescence signals were quantified and normalized to their respective PCLS areas using ImageJ v.1.53a. The resulting values were divided by those from the time point of CD8 + TCA for plotting, as indicated. In case of antagonist treatments, the values were divided by the respective condition without CD8 + TCA. CD8 + T cell abundance was approximated by measuring total Claret intensities, normalized by lung area and presented as relative intensities, as indicated.

## Statistics and reproducibility

Statistical analysis was performed using GraphPad Prism software (version 9.3.1) and within the provided R packages or cited online tools. Data is depicted as indicated. The n-numbers represent the number of biological replicates in each group. Statistical significance was assessed as indicated, depending on assay-specific sampling as well as type and distribution of the obtained data. The source data for all graphs are provided as Supplementary Data.

## Data availability

RNA-seq data has been deposited at GEO and is publicly available under the accession GSE286348. The source data for all graphs are provided in the Supplementary Data 3. Uncropped and unedited gel images as well as bots are provided in Fig. S11 within the Supplementary Information file, where all Supplementary Figs. are included. Additional information and all other data reported in this paper are available from the corresponding author upon reasonable request. No original code has been generated in this study.

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

## Acknowledgements
We are grateful for Eva Bauer, Britta Schlund, Friederike Gräbner and Danica Stich for excellent technical assistance. We thank Uwe Appelt and Markus Mroz for excellent FACS support as well as Andrea Eichhorn and Elke Kretzschmar for technical assistance for the TEM measurements. We also thank Elisabeth Naschberger, Timothy Chege Kuria and Dirk Mielenz for infrastructural support. We are grateful for all colleagues in the Franz-Penzoldt-Center and the Preclinical Imaging Platform Erlangen for excellent animal work support. This work was supported by grants to S.B. and H.S. from the Wilhelm-Sander Foundation (2020.039.1), to T.B., S.B., F.N., F.F., D.D. and MPS from the German Research Foundation (FOR2438/P04; TRR305/A03, A04, B01, B02, B05, B07, Z01 and the Priority Programme SPP 2084, EN 453/13-1) to MPS from the European Union's Horizon 2020 research and innovation program under the Marie Skłodowska-Curie grant agreement N° 861196 (PRECODE), to D.D. by the Agence Nationale de la Recherche and Deutsche Forschungsgemeinschaft as well as to H.S. from the Interdisciplinary Center for Clinical Research Erlangen (P34, P133).

## Author contributions
Conceptualization, K.F., T.B., H.S.; Methodology and Validation, K.F., M.P.S., E.D., C.G., M.W., P.A., M.H.H., G.K., H.S.; Software K.F., Rv.R., Y.H., R.L., F.F., L.A., H.S.; Formal Analysis, K.F., Rv.R., E.D., A.C., M.W., J.B., A.S., P.A., Y.H., H.S.; Investigation, K.F., I.A., E.D., A.C., N.B., A.C., K.S., M.W., J.B., M.E., M.A., E.P., A.S., P.A., E.G., M.F., J.A., J.H., L.A., H.S.; Resources, S.B., R.L., F.F., Y.H., Rv.R., F.N., D.D., C.B., M.H.H., C.G., M.P.S., T.B., H.S.; Writing – Original Draft, K.F., H.S.; Writing – Review & Editing, K.F., M.P.S., T.B., H.S.; Supervision, T.B., H.S.; Funding Acquisition, T.B., H.S.

## FundingInformation

## Competing interests
The authors declare no competing interests.

## Ethics
Animal husbandry and the experiments were approved by the committee of ethics of animal experiments of the state of Bavaria (Regierung von Unterfranken, Würzburg; TS-18/14, TS-30-2021, 55.2-DMS-2532-2-270, -2-1234) and performed according to the European Animal Welfare laws and guidelines. The responsibilities and roles were defined and agreed upon all involved colleagues, contributors and collaborators. The resulting contributions of all authors were important for designing and conducting the study and thus fulfilled the required criteria for authorship.

## Additional information

[1]Department of Experimental Medicine 1, Nikolaus-Fiebiger Center for Molecular Medicine, Friedrich-Alexander-Universität Erlangen-Nürnberg, Erlangen, Germany. [2]Department of Nephropathology, Institute of Pathology, Friedrich-Alexander-Universität Erlangen-Nürnberg, Erlangen, Germany. [3]Institute of Pathology, Friedrich-Alexander-Universität Erlangen-Nürnberg, Erlangen, Germany. [4]Department of Dermatology, Laboratory of Dendritic Cell Biology, Friedrich-Alexander-Universität Erlangen-Nürnberg, Universitätsklinikum Erlangen, Erlangen, Germany. [5]Division of Genetics, Department of Biology, Friedrich-Alexander-Universität Erlangen-Nürnberg, Erlangen, Germany. [6]Department of Internal Medicine 3 - Rheumatology and Immunology, Friedrich-Alexander-Universität Erlangen-Nürnberg and Universitätsklinikum Erlangen, Erlangen, Germany. [7]Department of Medicine 1, Universitätsklinikum Erlangen, Friedrich-Alexander-Universität Erlangen-Nürnberg, Erlangen, Germany. [8]Deutsches Zentrum Immuntherapie (DZI), Erlangen, Germany. [9]Institute for Systemic Inflammation Research, University of Lübeck, Lübeck, Germany. [10]Department of Rheumatology and Clinical Immunology, Charite Universitätsmedizin Berlin, Berlin, Germany. [11]Department of Medicine 1, Gastroenterology, Endocrinology and Pneumology, University Hospital Erlangen, Friedrich-Alexander University Erlangen-Nürnberg, Erlangen, Germany. [12]Medical Immunology Campus Erlangen, Friedrich-Alexander-Universität Erlangen-Nürnberg, Erlangen, Germany. [13]Institute of Immunology, Jena University Hospital, Friedrich-Schiller-University Jena, Jena, Germany. [14]Comprehensive Cancer Center Erlangen - European Metropolitan Area of Nuremberg (CCC-EMN), Erlangen, Germany. [15]FAU Profile Center Immunomedicine (FAU I-MED), Friedrich-Alexander-Universität Erlangen-Nürnberg, Erlangen, Germany. [16]Comprehensive Cancer Center Central Germany (CCCG) Jena, Jena, Germany. [17]Institute of Functional and Clinical Anatomy, Friedrich-Alexander-Universität Erlangen-Nürnberg, Erlangen, Germany. [18]Universitätsklinikum Erlangen, Department of Experimental Medicine 1, Nikolaus-Fiebiger Center for Molecular Medicine, Erlangen, Germany. [19]These authors contributed equally: Thomas Brabletz, Harald Schuhwerk. ✉e-mail: harald.schuhwerk@fau.de

