## [Transparent Peer Review file · Communications Biology]

Macrophages foster anti-tumor immunity by ZEB1-dependent cytotoxic T cell chemoattraction

Corresponding Author: Dr Harald Schuhwerk

Version 0:

Reviewer comments:

Reviewer #1

(Remarks to the Author)

In this manuscript, the authors identified the transcription factor ZEB1 as an intrinsic regulator of TAM function in adaptive anti-tumor immunity. Using a LysM-Cre transgenic mice with syngeneic models of colorectal and pancreatic cancer, they observed that ZEB1 supports secretion of CCL2 and CCL22 by promoting their biosynthesis and intracellular transport, regulating CTL recruitment and immunosurveillance during tumor outgrowth and lung metastasis.

Specific comments/questions:

Overall, the manuscript is very interesting. The experimental plan is very well organized and the author's conclusions are supported by the results.

1) In Fig 1 the authors found that ZEB1 is heterogeneously expressed in macrophages in CRC and PDAC. What is the role of ZEB1 in myeloid-derived suppressor cells (MDSC) in the spleen, bone marrow and tumor samples?

2) Fig 4 shows that ZEB1 is dispensable for phagocytosis but selectively promotes cytokine secretion in M1- or M2-macrophages. However, the authors should test the role of ZEB1 in tumor-conditioned macrophages (by culturing BMDM with tumor supernatant).

Reviewer #2

(Remarks to the Author)

In this manuscript, the Authors study the role of Zeb1 in macrophages, in the context of cancer. By using tissue-specific genetic depletion models, they convincingly demonstrate that Zeb1 expression in macrophages has an impact, albeit modest, on primary and metastatic disease. To identify the mechanistic bases of macrophage Zeb1 anti-tumor activity, the Authors assess phagocytic capacity, gene expression and cytokine release. Interestingly, protein and vesicular trafficking was decreased by the lack of Zeb1 in macrophages, which was reflected in a corresponding decrease in T cell recruitment to the tumor microenvironment.

The work is interesting and worth reporting, once minor issues are addressed. If the Authors want to further increase the impact of their findings, additional suggestions are provided.

Minor issues:

1. Statistical analyses: results that are central to the message of the paper lack statistical analyses or the statistics are unclear. For example, Fig. 4f and 5g lack stats but the Authors claim significance in the manuscript; Fig. 5e, stars and "ns" coexist on the same comparisons; Fig. 6d lacks stats within the Zeb1-deficient group if the Authors want to claim Ccl2/Ccl22 rescue T cell recruitment in vitro;
2. Ly6C should be used to assess whether monocyte influx was similar in Fig. S3. Currently, the Authors used F4/80 and CD68, which are expressed mainly on macrophages.
3. Analysis procedure for cytokine protein array is missing from methods.
4. Fig. S6a and related methods are unclear: is it proliferation or activation? From the methods, the Authors provided polyclonal activation via anti-CD3/CD28 before co-culturing with macrophages. This is counter-intuitive since anti-

CD3/CD28 already boost to max T cell proliferation, if IL2 or IL15 are present. The readout seems odd too, why % of CD8 instead of cell counts?

Impact boosters:

1. Given the finding that several Rab proteins are deregulated and the current interpretation by the Authors is that Zeb1 safeguards secretion processes, are extracellular vesicles also impacted by Zeb1 deficiency in macrophages?
2. Since both MC38 and CMT-93 respond very well to checkpoint inhibition, and given that the Authors found that Zeb1-deficient macrophages recruit less CD8 T cells into the tumor microenvironment, does the anti-tumor effect of checkpoint blockade also decrease in Zeb1-deficient condition?

Reviewer #3

(Remarks to the Author)

The study by Fuchs et al explores the role of the transcription factor ZEB1 in regulating Tumor-Associated Macrophages (TAMs) function and the resulting influence on anti-tumor immunity. Utilizing myeloid cell-specific deletion of Zeb1 and in vitro BMDM assays and in vivo murine tumor models, the authors conclude that ZEB1 is upstream of a transcriptional program in macrophages that regulates secretion of chemokines CCL2 and CCL22 and thereby controls T cell trafficking to tumors. While the authors performed an extensive set of studies and mechanisms regulating TAM functions is an area of high interest to the field, the differences observed between Zeb1 KO and WT conditions are relatively small and the mechanistic connections between Zeb1 loss and effects on macrophage-mediated control of the anti-tumor response are unclear. As a result, the dataset in its current form does not support the authors conclusions.

Specific comments:

1. In Figure 1 F-H, the y-axis units are not clear. For "CD68 Macrophages", is this a percentage or an absolute number per unit area? For "% ZEB1+ Macrophages", is this as a percentage of total CD68+ macrophages or something else?
2. In Figures 2 and S2, the data showing extent of Zeb1 protein loss in LysM Zeb1 is not convincing. The IF images show a large amount of background staining for CD68, making it difficult to determine the true macrophage population. Given that in the reporter animals only 70-80% of macrophage show evidence of Cre recombination, it seems unlikely that all tissue macrophages are knocked-out for Zeb1 as shown in Figure S2F. Understanding the extent of Zeb1 KO in macrophages and other cell types at the protein level is important for interpreting the results of these studies and the authors should work to strengthen these data.
3. In Figures 3A-B, the authors show effects of myeloid-specific Zeb1 loss on tumor engraftment rate following s.c. implantation of CMT-93 and MC38 cells. Of the tumors that successfully engrafted, were there differences in tumor growth over time between LysM Ctrl and LysM Zeb1 animals? Tumor growth rates should be shown for these studies. If the effect is only on engraftment, but not growth, this should be considered and discussed in the context of the authors claims around the importance of Zeb1 for chemokine expression that drives T cell recruitment to tumors. In general, tumor engraftment early after implantation may happen before an antigen-specific T cell response has formed, and is thus not thought to be as dependent on adaptive CD8+ T cell responses as subsequent tumor growth kinetics.
4. In reference to Figure 3C and F, the authors state that "In both models, both genotypes displayed equal BLI signals for up to 14 days post injection (dpi), suggesting similar initial metastatic seeding." The BLI signal from days 0-14 is quite low. What is the background BLI signal and limit of detection in these studies? If the signal is at or below detection limits, the authors cannot be sure that initial tumor seeding is not affected by Zeb1 loss. This could affect interpretation of the results in the context of whether Zeb1-mediated regulation of macrophages is affecting initial tumor seeding or it is playing a role in later stages of adaptive anti-tumor immunity.
5. In Figures 3F-H, it appears that none of the differences between LysM Ctrl and LysM Zeb1 are statistically significant. While the trends may be consistent with findings in the KPC model, the differences are small and not convincing, in particular the ex vivo BLI and colony count measurements. The authors should discuss these findings, rather than stating that data from both KPC and MC38 models are supportive over their conclusions.
6. In Figure 4F, it is unclear why the authors chose to focus on CCL2 and CCL22, when other cytokines were also seemingly affected by Zeb1 loss. In particular, given the critical role of CxCR3 chemokine ligands in regulating T cell trafficking to tumors, did the authors evaluate CXCL9/10 expression in more detail (notably it appears that CxCL10 expression is reduced to the same or greater extent that CCL2 or CCL22 in the LPS treatment conditions). On a similar note, did the authors evaluate the effect of IFN γ treatment on LysM Ctrl and LysM Zeb1 macrophage phenotype? There is a large body of literature demonstrating an important role for IFN γ signaling in macrophages, and downstream IFN γ -induced chemokines (CxCL10) and cytokines, in regulating adaptive anti-tumor immune response. This would be an important comparison and perhaps more relevant than LPS treatment.
7. In Figure 5D, the differences in LysM Ctrl and LysM Zeb1 expression of select intracellular protein trafficking genes appears to be quite small, with the majority less than a Log₂FC of 1. Was differential expression of any of these genes evaluated at the protein level? Demonstrating a clear difference in expression of trafficking protein would help to support the authors claims that these relatively small mRNA expression differences mechanistically responsible for defects in secretion of select protein in LysM Zeb1 macrophages. In addition, the authors should discuss why these broad mechanisms regulating intracellular protein trafficking/secretion would selectively affect a subset of secreted proteins in macrophages and

only under certain stimulation conditions?

8. In Figure 5F, it appears that almost all proteins on the array are increased in the LysM Zeb1 lysates relative to LysM Ctrl lysates. This raises concerns about artifacts related to small differences in protein loading between groups and non-linearity associated with this method of protein detection. The authors should consider an orthogonal method for intracellular protein detection (e.g. intracellular FACS staining) to confirm these results.

9. In Figure 6D, did the authors evaluate the combination of both anti-CCL2 and anti-CCL22? If both chemokines are responsible for driving CD8 T cell migration, an additive effect should be observed. For the current data, it is difficult to understand why anti-CCL2 and anti-CCL22 can both bring the level of T cell migration driven by LysM Ctrl macrophages down to the same level as observed with LysM Zeb1 macrophages. Wouldn't one expect that blockade of either chemokines alone would only partially affect migration and that the combination of the two blocking antibodies would be required for full suppression of migration?

10. In Figure 7A, the authors should state what timepoint post-tumor implantation this analysis was performed. Given these differences in CD8 T cell infiltration and cleaved caspase, one might expect a difference in tumor growth over time between LysM Ctrl and LysM Zeb1 groups. Was this observed (see comment 3 above)? If not, the authors should address this discrepancy.

11. In Figure 7C, the authors report CD8 T cells/tumor colony in the lung and show a non-significant result. In Figure 7D, the authors report CD8 T cells/field of view and show a significant result. Why were different quantification approaches used? Given that differences in tumor colony formation in the lung were observed in the KPC model (Figure 3), quantification by field of view doesn't seem appropriate. Do the authors see differences in tumor colony-associated CD8 T cell density between LysM Ctrl and LysM Zeb1 groups when normalized to tumor colony number and tumor area?

Reviewer #4

(Remarks to the Author)

In the present manuscript, the authors investigated the role of ZEB1 in macrophages by using conditional macrophage-specific ZEB1 KO (cKO) mice. The deletion of ZEB1 in macrophages did not affect organ development or homeostasis. However, tumor formation and colony formation in the lung were increased in cKO mice, suggesting the role for macrophage ZEB1 in anti-tumor effects. The secretion of CCL2 and CCL22 by cKO BM-derived macrophages (BMDM) was decreased compared to control macrophages. When mRNA levels of two chemokines were examined, there were only small differences between cKO and control macrophages, but the expression of genes related to protein trafficking, such as Rab family members and Vamp3 was affected, suggesting its role at a posttranscriptional level. As expected, protein synthesis increased upon LPS treatment but the full translational capacity was only slightly reduced in cKO BMDMs. Intriguingly, cKO BMDMs contained more intracellular cytokines upon LPS treatment than control BMDMs. Particularly, decreased CCL2 and CCL22 secretion levels were more evident than lower mRNA expression levels, and increased intracellular accumulation was remarkable. Finally, the authors found that CCL2 and CCL22 play a role in the recruitment of CD8+ T into tumors. From these results, the authors conclude that ZEB1 in macrophages promotes CD8+ cell infiltration into tumors and metastatic colonies in the lung via secretion of CCL2 and CCL22, preceding tumor cell death to counteract tumor outgrowth.

General comments:

Although some studies demonstrated a direct role of CCL2 in T cell recruitment, CCL2 is a chemoattractant for blood monocytes and its chemotactic activity for T cells is controversial. cKO macrophages secreted a lower level of CCL2, but they still secreted a significant amount of CCL2. Therefore, it is surprising that cKO BMDMs were incapable of recruiting T cells. There is not enough evidence to conclude the role for ZEB1 in the intracellular trafficking of CCL2 and CCL22 in macrophages. There are several sentences which are difficult to understand and editing is needed.

Specific comments:

1. Figure 1f and h: CD68+ macrophages on the Y axis. Are they numbers or percentages per field? Is there any functional differences between ZEB1 high and low macrophages?
2. Figure 4f: CXCR3 ligands are well accepted chemokines for CD8+ T cell recruitment. After LPS stimulation, the expression of CXCL10 was lower in cKO cells but there was no follow up study. In addition, the expression/secretion of other CXCR3 ligands, such as CXCL9, was not studied. It should be noted that significant concentrations of CCL2 and CCL22 were still detected in the supernatants of cKO cells.
3. Figure 5g: Secreted CCL2 by cKO cells was -1, whereas intracellular CCL2 was more than 10. Does this mean cKO cells produce more CCL2 in total? It would be more informative if real values by ELISA are presented. Arrays are good for a screening purpose but not to quantitate the concentration.
4. Figure 6d: Actual numbers of migrated T cells should be presented. CCL2 and CCL22 use different receptors. It is unclear why addition of either chemokine or antibody almost completely abolished the differences.
5. The level of intracellular cytokine was presented as the amount/100 ug protein. What were the absolute amounts per cell or per a same number of cells?
6. The biological relevance of in vitro T cell migration assay is questionable. The results were obtained after 24 hour culture. Why was this time point chosen? T cell migration can be evaluated with much shorter culture time. Random migration could

be increased after a long culture.

7. Line 75: "by safeguarding intracellular protein transport". What does "safeguarding" actually mean? The authors did not test whether ZEB1 actually regulates intracellular protein transport.

8. Figure 7a: What is the histological score on the Y-axis?

9. Figure 7d: Why were KPC cells used in this experiment? Despite the infiltration of CD8+ T cells was less in cKO mice, the size of metastatic foci appears smaller than those in control mice.

10. Figure 7d: It would be helpful to examine the kinetics of CCL2 and CCL22 expression in tumors to support the authors' conclusion.

11. Figure 7e: The difference was statistically significant on only Day 21 and this result does not appear to support the conclusion.

12. Throughout the study, primary tumors were not examined. What about tumor size? When were tumors harvested? What about the expression of CCL2 and CCL22 in tumors? What about the infiltration of CD8+ T cells?

13. BMDCs culture: BM cells were seeded into a petri dish in BMDC medium. On day 1, nonadherent cells were cultured to prepared BMDCs. This is different from the method often used by others. It is unclear why nonadherent cells on day 1 were used to prepare BMDCs.

Version 1:

Reviewer comments:

Reviewer #1

(Remarks to the Author)

Most of the raised questions have been addressed. The new version of the manuscript is sufficiently improved.

Reviewer #3

(Remarks to the Author)

The authors have addressed the majority of the questions I had raised in my original review. The quality and clarity of the revised manuscript have been strengthened and the overall conclusions are better supported by the experimental data. I have a few remaining questions and suggestions:

- In Fig 5, the authors conclude that reduced secretion of CCL2, CCL22, and CxCL10 are the result of increased trafficking of secretory endosomes to a degradative pathway in LysM-Zeb1 KO cell.

1) There is a discrepancy between Fig 5G and Fig 4G with respect to the VEGF control, where reduced intracellular levels of VEGF are observed in LysM-Zeb1 M0 macrophages, but this has no apparent effect on VEGF secretion. Similarly, M0 and LPS macrophages have differing levels of intracellular VEGF that doesn't appear to correlate to secreted levels. My assumption is that VEGF is meant as a control to show some degree of specificity of the Zeb1 effect to select chemokines. As such, that authors should address these discrepancies.

2) Given the importance of Fig 5 to deciphering the mechanism of Zeb1-mediated effects on macrophages, have the authors attempted to pharmacologically block lysosomal degradation or autophagy pathways in macrophages? Showing a restoration of intracellular levels of chemokines would provide important supporting evidence for the model show in Fig 5J.

- In Fig 7F-H, the authors utilize PCLS to further investigate mechanisms of Zeb1 regulation of macrophage function and anti-tumor T cell responses. Adoptive cell therapy is generally used to describe transfer of cells in vivo. In this case, my understanding is the CD8+ T cells are added to the PCLS culture. Is this correct? If so, the authors may want to avoid using this terminology to avoid confusion (in vivo transfer before harvesting lung tissue or adding T cells in vitro to the PCLS culture).

Reviewer #4

(Remarks to the Author)

1. As noted in my previous comments, the major chemokine/chemokine receptor system regulating the trafficking of CD8+ T cells into solid tumors is CXCR3 and its ligands, such as CXCL9, CXCL10 and CXCL11 (reviewed by Kohli et al., Cancer Gene Therapy, 29:10-21, 2022). After a revision, it is still unclear whether CCL2 and CCL22 directly regulate the recruitment of CD8-positive T cells in the models used in this study. This important point is not even discussed.

2. Raw data of T cell migration assay are not presented, making it difficult to evaluate the results. The experiment should be conducted with positive controls. It would be more straightforward if the authors used cell culture supernatants instead of BMDMs (Fig. 6d).

3. In tumor microenvironments, not only macrophages but also other cell types, including cancer cells, produce and release CCL2. In some mouse cancer models, cancer cells rather than macrophages are the main source of this chemokine. It is important to examine the levels of CCL2 and CCL22 in tumors growing in control and cKO mice.

4. Several new sentences/paragraphs are added but they are not clearly written and difficult to understand.

Version 2:

Reviewer comments:

Reviewer #3

(Remarks to the Author)

The authors have adequately addressed my comments. The manuscript is suitable for publication.

Reviewer #4

(Remarks to the Author)

January 20th 2025

Subject: Revised manuscript : "Macrophages foster anti-tumor immunity by ZEB1-dependent cytotoxic T cell chemoattraction"

Manuscript Number: COMMSBIO-24-3324-T

To the Reviewers:

We thank the reviewers for their critical and overall positive assessment of our manuscript. We highly appreciate their well-grounded suggestions, which helped us to significantly improve our study.

We have performed a number of additional experiments to address their concerns with new data and edited the manuscript as described in detail in a point-by-point response below. We have:

1. Analyzed subcutaneous growth of CMT-93 tumor cells, discovering a high rate of tumor regression after initial growth in *Zeb1*-proficient mice ($LysM^{Ctrl}$) but not in those lacking *Zeb1* in macrophages ($LysM^{\Delta Zeb1}$), substantiating a role of ZEB1 in macrophages in adaptive anti-tumor immunity in this model.
2. Provided mechanistic evidence from accurate intracellular cytokine quantifications, transmission electron microscopy and western blotting for key players in endosomal vesicular trafficking that loss of *Zeb1* restricts cytokine release and unleashes degradative vesicle fusions likely involving VAMP3, VAMP8, RAB6 and RAB35, together manifesting an impairment of the constitutive endosomal secretion pathway in macrophages.
3. Employed a model for adoptive transfer of CD8⁺ T cells (ACT) to organotypic precision-cut lung slice cultures, recapitulating the growth advantage of KPC cells on $LysM^{\Delta Zeb1}$ lungs and showing that ACT efficiency requires ZEB1 in macrophages and the CCL2-CCR2 and CCL22-CCR4 axes in this model.
4. Confirmed reduced cytokine secretion in response to the cancer-relevant Interferon- γ in $LysM^{\Delta Zeb1}$ macrophages, accurately quantified ZEB1-dependent CXCL10 release from macrophages upon LPS stimulation, revealed increased sensitivity of CD8⁺ T cell migration *in vitro* to the availability of CCL2 and CCL22 than to CXCL10 and explain the observed redundancy of CCL2 and CCL22 therein with coupled CCR2 and CCR4 but not CXCR3 surface expression on CD8⁺ T cells.
5. Determined the kinetics of CCL2 abundance in KPC-colonized lungs by IHC, revealing reduced signals in colonized $LysM^{\Delta Zeb1}$ lungs which matches the influx of CD8⁺ cells and tumor cell death.

6. Substantiated the loss of ZEB1+ cells within macrophage sub-populations in organs of $LysM^{\Delta Zeb1}$ mice by intracellular flow cytometry, for example in pulmonary alveolar macrophages.
7. Updated abstract and main text, improved figure labeling and presentation as well as statistical and methodological descriptions, where requested. The RNA sequencing data is now deposited for public access upon publication of this study (GEO accession: GSE286348; reviewer token: izivggqabpifvyx).

Overall, we are confident that scientific rigor and translational relevance of our study increased considerably. In addition to the ultrastructural and molecular analyses corroborating the impact of ZEB1 on endosomal vesicular trafficking during cytokine secretion, the importance of the CCL2-CCR2 / CCL22-CCR4 axes for ACT in a therapeutically relevant organotypic model has been shown, alongside with its dependence on ZEB1 in macrophages. Hence, we are convinced that publication in *Communications Biology* would raise the awareness of critical yet tangible onco-immunological context within the biomedical research community.

Point-by-point response to the reviewers' comments:

Reviewer #1:

In this manuscript, the authors identified the transcription factor ZEB1 as an intrinsic regulator of TAM function in adaptive anti-tumor immunity. Using a LysM-Cre transgenic mice with syngeneic models of colorectal and pancreatic cancer, they observed that ZEB1 supports secretion of CCL2 and CCL22 by promoting their biosynthesis and intracellular transport, regulating CTL recruitment and immunosurveillance during tumor outgrowth and lung metastasis.

Specific comments/questions:

Overall, the manuscript is very interesting. The experimental plan is very well organized and the author's conclusions are supported by the results.

Answer:

We thank the reviewer for the concise summary of our study and for the positive feedback. We are very happy to read that our study is convincing.

1) In Fig 1 the authors found that ZEB1 is heterogeneously expressed in macrophages in CRC and PDAC. What is the role of ZEB1 in myeloid-derived suppressor cells (MDSC) in the spleen, bone marrow and tumor samples?

Answer:

We thank the reviewer for pointing this out and agree that MDSCs are a very important, specific myeloid cells in the tumor microenvironment (TME). The myeloid TME is very complex and dynamic and is by far not understood properly. Therein, MDSCs are characterized by the expression of multiple markers, including for example CD11b, Ly6G and Ly6C but this may well vary in a context- and polarization-dependent manner. Studying the role of *Zeb1* specifically in MDSCs would thus require their selective targeting which is not possible with available and established genetic strategies. Approaching it by systematic (co-)depletions of other myeloid cells is technically challenging and is too ambitious for one study in our opinion. In this regard, it may be noteworthy to mention that knowledge about *Zeb1* in TAMs is scarce, and we are among the first to approach its potential relevance *in vivo* using the well-studied LysMCre, known to target predominantly macrophage subsets and neutrophil granulocytes and thereby discovered a novel tumor-suppressive role of *Zeb1* in these cell types. Since we confirmed this major effect in a metastatic lung colonization model *in vivo* using an additional macrophage-deleter (CX3CR1-Cre) while a much weaker effect was observed using a neutrophil-specific Cre (Ly6G-Cre), we are confident that *Zeb1* exerts this effect when expressed in certain myeloid cells, foremost in macrophages and subtypes therein, rather than neutrophil granulocytes. However, with regards to MDSCs, we cannot exclude or confirm with confidence that we target them as well in the tumor context. This is now also discussed in the discussion section of the manuscript, stating: "(...) Notably, it cannot be excluded that myeloid-derived suppressor cells are targeted as well and potentially influence these *in vivo* effects. (...)” Importantly, though, we would not be able to definitely attribute any of their potential contributions in the tumor context specifically to them due to the aforementioned technical limitations.

2) Fig 4 shows that ZEB1 is dispensable for phagocytosis but selectively promotes cytokine secretion in M1- or M2-macrophages. However, the authors should test the role of ZEB1 in tumor-conditioned macrophages (by culturing BMDM with tumor supernatant).

Answer:

We thank the reviewer for addressing this interesting point. Macrophages are very heterogenous and plastic cells that can adopt a variety of interchangeable polarization states depending on the precise stimulation conditions *in vitro* and immunological context *in vivo*. Among the artificially induced archetypes within the spectrum of macrophage polarization states, we first used LPS and IL4 as

established polarizing agents to induce particularly distinguishable phenotype changes *in vitro* to study the role of *Zeb1* in phenotypically and functionally adopting these states. In this way, we pinpointed a cancer-relevant role of *Zeb1* in cytokine secretion upon inflammatory activation (using LPS) that affects CD8+ T cell recruitment. In principle, we agree on the importance of modeling the actual tumor context. However, we first focused on confirming the role of *Zeb1* in inflammatory activation by exposing BMDMs to the cancer-relevant Interferon- γ that is crucial for adaptive anti-tumor immune responses, in order to verify impaired cytokine secretion when *Zeb1* is deleted in macrophages. Here, secreted CCL2 and CCL22 were decreased as well upon ZEB1 loss (rev. Fig. 4h), reinforcing the stimulus-independent importance of ZEB1 in macrophages for cytokine release upon inflammatory activation. Moreover, we show that *Zeb1* in macrophages as well as the CCL2-CCR2 and CCL22-CCR4 axes are instrumental for tumor control upon adoptive transfer of CD8+ T cells in organotypic KPC-colonized precision-cut lung slice cultures (rev. Fig. 7f-h). As pulmonary macrophages therein are conditioned by the KPC cells seeded to these lungs, these data confirm the importance of ZEB1 in macrophages and the pinpointed cytokines (CCL2 and CCL22) in the tumor context. Collectively, we conclude that *Zeb1* promotes cytokine secretion in macrophages and T cell recruitment in a largely stimulus-independent manner and, importantly, in a context that is relevant for cancer biology and immunotherapy.

Reviewer #2:

In this manuscript, the Authors study the role of Zeb1 in macrophages, in the context of cancer. By using tissue-specific genetic depletion models, they convincingly demonstrate that Zeb1 expression in macrophages has an impact, albeit modest, on primary and metastatic disease. To identify the mechanistic bases of macrophage Zeb1 anti-tumor activity, the Authors assess phagocytic capacity, gene expression and cytokine release. Interestingly, protein and vesicular trafficking was decreased by the lack of Zeb1 in macrophages, which was reflected in a corresponding decrease in T cell recruitment to the tumor microenvironment.

The work is interesting and worth reporting, once minor issues are addressed. If the Authors want to further increase the impact of their findings, additional suggestions are provided.

Answer:

We thank the reviewer for the positive and constructive feedback. We are delighted to read that our study is overall worth reporting and only needs addressing of minor issues

Minor issues:

1. Statistical analyses: results that are central to the message of the paper lack statistical analyses or the statistics are unclear. For example, Fig. 4f and 5g lack stats but the Authors claim significance in the manuscript; Fig. 5e, stars and “ns” coexist on the same comparisons ; Fig. 6d lacks stats within the Zeb1-deficient group if the Authors want to claim Ccl2/Ccl22 rescue T cell recruitment in vitro;

Answer:

We thank the reviewer for exposing a lack of clarity in reporting and presenting of statistical comparisons in the figures, their descriptions in the main text and figure legends. We corrected these passages in the manuscript and would like to specifically address the examples given by the reviewer.

In the original Fig. 4f (rev. Fig. 4f), which shows a cytokine spot array, a claim for significance was neither intended, nor stated in the original manuscript. We validated interesting cytokine hits identified from this array using a bead-based, quantitative assay (Legendplex, Biolegend), as presented in the original Fig. 4g (now rev. Fig. 4g), which is as sensitive as ELISA. In the rev. Fig. 4g, statistics are properly described, we think. Importantly, to strengthen our central conclusion that ZEB1 in macrophages is crucial for cytokine secretion upon inflammatory stimulation, we now confirmed reduced secretion by Zeb1-deficient BMDMs upon IFN γ treatment (rev. Fig. 4h) and included quantification of the well-known and cancer-relevant cytokine CXCL10 in the bead-based assay upon LPS treatment in the rev. Fig. 4g. Notably, a functional follow-up of CXCL10 has been conducted as well and is presented in rev. Fig. 6d, showing absence of an effect of CXCL10 supplementation on migration of CD8 $^+$ T cells *in vitro*, corroborating importance and specificity of CCL2 and CCL22 in our models (please also see for example the response to reviewer #3, point 6).

With regards to original Fig. 6d, this comparison indeed does not reach significance. However, one astonishing observation from our point of view is that the LysM $^{\Delta Zeb1}$ BMDMs seemingly cannot elevate CD8 $^+$ T cell migration as such when compared to the T cell-only condition. Moreover, the significant difference in the control conditions (PBS, IgG) when comparing LysM $^{\Delta Zeb1}$ to LysM Ctrl is gone upon addition of CCL2 and CCL22, seemingly because of a clearly visible, yet statistically insignificant increase upon co-culture with LysM $^{\Delta Zeb1}$ BMDMs, which is described as “rescue” in our manuscript. To improve visualization of the mean differences observed in the experimental conditions, two colored lines indicating the mean of the control conditions upon LysM Ctrl (blue) and LysM $^{\Delta Zeb1}$ (red) co-cultures have been added to the graphs. Supporting our hypothesis developed partly on the basis of these *in vitro* assays, we show reduced CD8 $^+$ cell abundance in situ (original Fig. 7 and now rev. Fig. 7a, c, d) in tumors and metastases in LysM $^{\Delta Zeb1}$ mice, correlating with CCL2 abundance in KPC-colonized lungs, which we analyzed by IHC during the revision (rev Fig. 7d; see also point 10). Moreover, efficient control of tumor growth (KPC) by adoptively transferred splenic CD8 $^+$ T cells (ACT) in an organotypic precision-cut lung slice culture model required Zeb1 in macrophages and was sensitive to CCR2/CCR4 antagonists (rev. Fig. 7f-h). Collectively, these data suggest that CCL2 and CCL22 indeed support

recruitment of CD8⁺ T cells also in tumors and/or metastases, at least partially via the CCL2-CCR2 and CCL12-CCR4 axes.

Please note that the original Fig. 5 has been substantially modified and improved, whereby orig. Fig. 5f and orig. Fig. 5g has been replaced by the quantitative bead-based assay presented in rev. Fig. 5g.

We are sorry for the insufficient explanation of the statistical indicators in the orig. Fig. 5e (rev. Fig. 5e.) Here, color-coded indicators (blue=LysM^{Ctrl}, red= LysM^{ΔZeb1}) on top of the bars are provided for the comparison of a given time point in a given genotype to the respective time point “0 h” in the same genotype to evaluate the actual response kinetic to LPS in the two genotypes separately. In addition, comparisons of the genotype (LysM^{Ctrl} versus LysM^{ΔZeb1}) at specific time points are marked by the connectors. An explanation has been added to the revised figure legend.

2. Ly6C should be used to assess whether monocyte influx was similar in Fig. S3. Currently, the Authors used F4/80 and CD68, which are expressed mainly on macrophages.

Answer:

We agree that monocyte influx is important to be assessed, in addition to macrophage markers. We now scored Ly6C⁺ cells on fresh frozen tissue slides. This was again similar in colonized lungs of LysM^{Ctrl} and LysM^{ΔZeb1} mice. The data is now presented in rev. Fig. S3e and mentioned in the rev. manuscript on page 8.

3. Analysis procedure for cytokine protein array is missing from methods.

Answer:

We thank the reviewer for spotting the missing procedure and have now included a description of the method.

4. Fig. S6a and related methods are unclear: is it proliferation or activation? From the methods, the Authors provided polyclonal activation via anti-CD3/CD28 before co-culturing with macrophages. This is counter-intuitive since anti-CD3/CD28 already boost to max T cell proliferation, if IL2 or IL15 are present. The readout seems odd too, why % of CD8 instead of cell counts?

Answer:

We are sorry for imprecise usage of these terminologies. It is in fact “proliferation”, as stated in the figure title, which is an established marker for “activation” of T cells that was assayed here. We added this description to the respective figure legend (see below). Methodologically, the freshly isolated CD8⁺ T cells were pulse-labelled with CellVue Claret (a membrane stain similar to CFSE), then incubated with anti-CD3/CD28, and added to BMDMs for 3 days at 37 °C in a humidified incubator (5% CO₂). Notably, during cell division, the CellVue Claret stained membrane of the mother cell is distributed approximately 1:1 among two daughter cells (akin to CFSE). It is therefore routinely used to assess proliferation of T cells as a reliable yet general marker for T cell activation. After the co-cultures with BMDMs, the CD8 T cells were harvested and CellVue Claret intensity was measured by flow cytometry, as described in the methods. In Fig. S6 (now rev. Fig. S6b), we then plotted the fraction of CD8⁺ T cells that have undergone cell division, as determined from the decreasing CellVue claret intensities. This is one reason to represent the data as fraction of all CD8 T cells that have been proliferating. Another reason is inter-experimental variance that should be normalized for. The legend to rev. Fig. S6b now states: “ **b**. Percentage of proliferated (i.e., activated) CD8⁺ T cells as among total CD8⁺ T cells in absence (n=1) or presence of M0, LPS or IL-4 pre-stimulated LysM^{Ctrl} (n=2) or LysM^{ΔZeb1} (n=3) BMDMs (means +SD; 2-way ANOVA).” Moreover, we have now amended the method description on page 58 of the revised manuscript to “(...) After the co-cultures with BMDMs, the CD8 T cells were harvested, pelleted by centrifugation at 1500 rpm at 4 °C for 10 min, resuspended in 200 μl ice-cold FACS buffer and transferred to a fresh 96-well plate. T cells were washed with ice-cold FACS buffer and fluorescence data was acquired in the CytoFlex analyzer (Beckman Coulter) and analyzed using the CytExpert or Kaluza

software (Beckman Coulter). The fraction of CD8⁺ T cells that have undergone cell division(s), as determined from the CellVue claret intensities, has been plotted”.

Regarding the treatments of the CD8 T cells, anti-CD3/CD28 stimulation was present but neither was additional IL-2, nor IL-15. In this way, the effect of ZEB1 in BMDM on T cell activation, here proliferation, was analyzed without saturating it beforehand. This assay revealed a slight increase in the fraction of proliferated CD8 T cells in the presence of BMDM, particularly LPS-prestimulated ones (as expected). However, as no difference was observed between the LysM^{Ctrl} and LysM^{ΔZeb1} BMDMs, we concluded that the extent of CD8 T cell activation is independent of ZEB1 in BMDMs.

Impact boosters:

1. Given the finding that several Rab proteins are deregulated and the current interpretation by the Authors is that Zeb1 safeguards secretion processes, are extracellular vesicles also impacted by Zeb1 deficiency in macrophages?

2. Since both MC38 and CMT-93 respond very well to checkpoint inhibition, and given that the Authors found that Zeb1-deficient macrophages recruit less CD8 T cells into the tumor microenvironment, does the anti-tumor effect of checkpoint blockade also decrease in Zeb1-deficient condition?

Answer:

We thank the reviewer for these additional suggestions. Although we truly appreciate these optional impact boosters, we feel like particularly the second one appears somewhat peripheral to the main thread, despite its high translational relevance.

Nevertheless, while focusing on corroborating our major findings within this manuscript, we seized the idea of demonstrating immunotherapeutic implications of our findings. We now demonstrate that efficient control of tumor growth (KPC) by adoptively transferred splenic CD8⁺ T cells (ACT) in an organotypic precision-cut lung slice culture model requires *Zeb1* in macrophages and is sensitive to CCR2/CCR4 antagonists (rev. Fig. 7f-h).

Indeed, EVs play a significant role in macrophage biology. To address this, we attempted to quantify EVs from available supernatant samples using Nanoparticle Tracking Analysis (NTA) in collaboration with Prof. Claudia Günther (Erlangen), a recognized expert in EV research. Notably, the isolation of high-quality EVs ideally requires specific protocols (MISEV) for cell culture and sample preparation, including thorough filtering of all medium supplements (0.22 μm), particularly FCS, to avoid masking cell-derived EVs with medium components. Unfortunately, these specialized procedures were not applied in our original experiments, nor did we significantly alter our standard BMDM culture conditions for the revision experiments, as this would have compromised the comparability of the new data with our initial findings. While we observed a potential trend toward a reduced release of EVs in LysM^{ΔZeb1} BMDMs compared to LysM^{Ctrl} BMDMs without affecting the size distribution, cell-derived EVs were significantly masked by medium-derived EVs. Importantly, the EVs analyzed displayed the expected size distribution, with a

main peak between 80–100 nm, and showed a spherical shape. However, this dataset remains preliminary and inconclusive at this point. We have therefore decided not to include these findings in the current manuscript but regard them as an encouraging basis for future studies. In the discussion we now state on page 17: “(...) Notably, an influence on the release of relevant extracellular vesicles remains elusive. (...)”. We are committed to continuing this research using optimized EV isolation protocols to better elucidate this relevant aspect in follow-up studies.

Reviewer #3

The study by Fuchs et al explores the role of the transcription factor ZEB1 in regulating Tumor-Associated Macrophages (TAMs) function and the resulting influence on anti-tumor immunity. Utilizing myeloid cell-specific deletion of Zeb1 and in vitro BMDM assays and in vivo murine tumor models, the authors conclude that ZEB1 is upstream of a transcriptional program in macrophages that regulates secretion of chemokines CCL2 and CCL22 and thereby controls T cell trafficking to tumors. While the authors performed an extensive set of studies and mechanisms regulating TAM functions is an area of high interest to the field, the differences observed between Zeb1 KO and WT conditions are relatively small and the mechanistic connections between Zeb1 loss and effects on macrophage-mediated control of the anti-tumor response are unclear. As a result, the dataset in its current form does not support the authors conclusions.

Answer:

We thank the reviewer for the critical, yet well-grounded and constructive feedback. By gathering a significant amount of new experimental evidence, we are confident to now change the reviewer's initial mind that our conclusions were not supported by the presented data in the original manuscript. Particularly with regards to macrophage-mediated control of the anti-tumor response, we now demonstrate that efficient control of tumor growth (KPC) by adoptively transferred splenic CD8+ T cells (ACT) in an organotypic precision-cut lung slice culture model requires *Zeb1* in macrophages and is sensitive to CCR2/CCR4 antagonists (rev. Fig. 7f-h). Our point-by-point response is detailed below.

Specific comments:

1. In Figure 1 F-H, the y-axis units are not clear. For "CD68 Macrophages", is this a percentage or an absolute number per unit area? For "% ZEB1+ Macrophages", is this as a percentage of total CD68+ macrophages or something else?

Answer:

We thank the reviewer for pointing out this inaccurate labeling, which also caught the attention of reviewer #4, (point 1). It is number of CD68+ cells per field of view on the left and percentage of ZEB1+ cells among the CD68+ cells per field of view on the right. The figure legend and the axis labels have been corrected accordingly in the rev. Fig. 1.

2. In Figures 2 and S2, the data showing extent of Zeb1 protein loss in LysMΔZeb1 is not convincing. The IF images show a large amount of background staining for CD68, making it difficult to determine the true macrophage population. Given that in the reporter animals only 70-80% of macrophage show evidence of Cre recombination, it seems unlikely that all tissue macrophages are knocked-out for Zeb1 as shown in Figure S2F. Understanding the extent of Zeb1 KO in macrophages and other cell types at the protein level is important for interpreting the results of these studies and the authors should work to strengthen these data.

Answer:

We appreciate this comment and agree that a more thorough analysis of ZEB1 loss may be informative. However, we want to mention that we have provided a thorough multi-organ analysis of reporter allele recombination driven by LysMCre, a well-established and frequently employed myeloid-/ macrophage deleter - also for conditional loss of ZEB1 (Cortes et al., 2023; doi: 10.1038/s41467-023-42277-4). In addition, we have confirmed the loss of ZEB1 in BMDMs as well as the major pro-tumorigenic effect *in vivo* (i.e., increased lung colonization in Cre+ Zeb1-deleted mice) using additional macrophage-specific (CX3CR1-Cre) and neutrophil-specific (Ly6G-Cre) Cre lines. The original analysis was not intended to show absolute elimination of ZEB1 protein with high precision but to verify that a considerable amount of target cells lost ZEB1 protein. In light of the considerable heterogeneity in myeloid cell populations and their partial scarcity, we now performed multi-parametric, full spectrum intracellular flow cytometry

of ZEB1 with various myeloid markers from organs from naïve $LysM^{Ctrl}$ and $LysM^{\Delta Zeb1}$ mice. To our knowledge, such an approach has not been undertaken before. We focused on spleen, lung, liver and pancreas for this analysis. The main text of the rev. manuscript now says "(...) As compared to their $LysM^{Ctrl}$ littermates, $LysM^{\Delta Zeb1}$ mice contained significantly less ZEB1+ splenic (CD45+;Ly6C-;F4/80+;CD11b+) as well as alveolar (SiglecF+;CD11b-low) but not non-alveolar (CD11b-high) pulmonary macrophages (CD45+;Ly6G-;F4/80+;CD68+). For the hepatic Kupffer cells (CD45+;CD11b+;F4/80+;Tim4+;CD206^{high/low}) and the very scarce pancreatic macrophages (CD45+;CD68+;CD11b+) a visible trend was observed (Fig. 2f, S2g, h)." Hence, we now revealed a reduction of ZEB1+ cells in certain subsets of macrophages, particularly in pulmonary alveolar macrophages, consistent with the *in vivo* effect of impaired anti-tumor immune responses during metastatic lung colonization. More detailed characterization of ZEB1 expression and its loss in different immunological niches exceeds the scope of the present manuscript. However, this certainly entails important follow-up research on EMT-related plasticity in specific immune subpopulations. We now addressed this on page 17 of the revised discussion, stating: "(...) These studies warrant future detailed dissection of ZEB1/EMT-related cellular plasticity in specific immune cell subtypes within different (onco-)immunological niches. (...)".

3. In Figures 3A-B, the authors show effects of myeloid-specific Zeb1 loss on tumor engraftment rate following s.c. implantation of CMT-93 and MC38 cells. Of the tumors that successfully engrafted, were there differences in tumor growth over time between $LysM^{Ctrl}$ and $LysM^{\Delta Zeb1}$ animals? Tumor growth rates should be shown for these studies. If the effect is only on engraftment, but not growth, this should be considered and discussed in the context of the authors claims around the importance of Zeb1 for chemokine expression that drives T cell recruitment to tumors. In general, tumor engraftment early after implantation may happen before an antigen-specific T cell response has formed, and is thus not thought to be as dependent on adaptive CD8+ T cell responses as subsequent tumor growth kinetics.

Answer:

We are very grateful for this important comment. We have now included the s.c. tumor growth curves of the CMT-93 model in rev. Fig. 3a as single-mouse /-tumor spaghetti plots. Interestingly, we did not observe failed tumor engraftment but complete regression of the tumors after initial tumor growth for 1-4 weeks, exclusively in $LysM^{Ctrl}$ mice. In this model, which has been shown to be remarkably immunogenic depending on inflammatory cytokines (Crittenden et al., Cancer Res. 2003 PMID: 14500387), there was not a single case, in which tumors were not formed at least initially. However, there was a similar fraction of cases in $LysM^{\Delta Zeb1}$ mice that re-started growing upon partial regression. In conjunction with the longitudinal lung colonization BLI data (see also point 4. below) and the expected timeline for forming adaptive immunity (up to a few weeks), this argues for a contribution of adaptive immune responses. Since absence of tumors was a result of regression rather than failed engraftment, the corresponding endpoint volumes and masses of all mice except the ones which needed to be sacrificed prematurely due to rare tumor ulceration, are now presented in rev Fig. 3C, showing a significant difference between the genotypes.

Unlike epithelial CMT-93 cells, which grew slowly yet constantly subcutaneously and showed ulcerations rarely and mildly, mesenchymal MC-38 cells formed tumors with highly aggressive growth kinetics. We experienced that, whereas small and still flexible tumors could not be measured accurately, established tumors grew very fast, partially invasive, ulcerated frequently, rapidly and deeply. Following ethical guidelines, this resulted in premature dropout of a substantial fraction of mice and decreased the statistical power. This data was therefore not included in the manuscript. Please see below the growth curves upon injection of 0.75×10^6 cells (left: spaghetti plot; right: means per genotype).

While engraftment was not affected in $LysM^{\Delta Zeb1}$ mice, as stated in the main text, it appeared that a fraction of the tumors in $LysM^{\Delta Zeb1}$ mice grew bigger before the mice needed to be sacrificed (SA). Nevertheless, significance is reached 19 days after injection, albeit only for the survivors showing the slowest tumor growth in general. This prompted us to conduct the limiting dilution assay (Fig. 3b and rev. Fig. 3d), in which for ethical reasons the animals were sacrificed early, *i.e.*, once a tumorous mass was visible and palpable over a few days. This confirmed a tumor-suppressive effect of ZEB1 in macrophages.

Our thorough and comprehensive previous as well as new analyses indicate an impaired adaptive immune response rather than a reduced engraftment or seeding capability. This is evident from the timeline of regression in the s.c. CMT-93 model (rev. Fig. 3a), the longitudinal BLI measurements during lung colonization (rev. Fig. 3e, h; see also point 4), the cell death and the CD8+ cell influxes at endpoints in s.c. tumors as well as measured over time during lung colonization (rev. Fig. 7 a, c, d). Furthermore, our new finding that ZEB1 in macrophages controls KPC growth via the CCL2-CCR2 as well as CCL22-CCR4 axes that became apparent upon adoptive transfer of CD8+ T cells in an organotypic precision-cut lung slice culture model (rev. Fig. 7, f-h), strongly supports this idea. However, as we cannot rule out other tumoricidal influences, we omitted the term “adaptive” from definite conclusions in the text, abstract and title.

4. In reference to Figure 3C and F, the authors state that “In both models, both genotypes displayed equal BLI signals for up to 14 days post injection (dpi), suggesting similar initial metastatic seeding.” The BLI signal from days 0-14 is quite low. What is the background BLI signal and limit of detection in these studies? If the signal is at or below detection limits, the authors cannot be sure that initial tumor seeding is not affected by *Zeb1* loss. This could affect interpretation of the results in the context of whether *Zeb1*-mediated regulation of macrophages is affecting initial tumor seeding or it is playing a role in later stages of adaptive anti-tumor immunity.

Answer:

We appreciate this sensible technical remark. We have set up BLI measurements properly beforehand to stay in a dynamic signal range across mice, genotypes and experiments, despite keeping identical exposure times and durations of luciferase bio-distribution after subcutaneous injection of luciferin. In this way, it was ensured that saturated pixels were prevented and thus did not occur. Hence, the limit of detection was not reached. With regards to the detection minimum, the background signal in the *in vivo* measurements typically ranges around $2-3 \times 10^3$, independent of whether it is measured on the stage or on the abdomen of a mouse. This is much lower (at least 1 decimal) than the signals obtained from the mouse lungs bearing luciferase-expressing tumor cells. Please see a version of rev. Fig. 3e below, where the range of the y-axis was modified to discriminate signals from the background values (dashed line). Hence, early-stage and low-level tumorous burden, *i.e.*, initial seeding, can be measured accurately by BLI *in vivo*. Therefore, the growth kinetics measured via BLI *in vivo* reliably indicates exactly what we have stated and the reviewer has quoted.

Please also note the small drop in BLI signal in the $LysM^{Ctrl}$ mice between day 7 and day 14, which is precisely the time where CCL2, CD8+ cell influx and tumor cell death is peaking *in situ*, mostly in this genotype. This data in concert with the regressing subcutaneous tumors after initial growth and the IHC analyses suggests that ZEB1 in macrophages supports later stages of anti-tumor immune responses rather than initial seeding.

5. In Figures 3F-H, it appears that none of the differences between $LysM^{Ctrl}$ and $LysM^{\Delta Zeb1}$ are statistically significant. While the trends may be consistent with findings in the KPC model, the differences are small and not convincing, in particular the *ex vivo* BLI and colony count measurements. The authors should discuss these findings, rather than stating that data from both KPC and MC38 models are supportive over their conclusions.

Answer:

We agree that the differences are small and not significant in the MC-38 tail vein injection (TVI) model and we amended the respective text to clarify this: "(...) Strikingly, while the signals plateaued further on in $LysM^{Ctrl}$ mice, $LysM^{\Delta Zeb1}$ mice displayed a sudden elevation of BLI signal, aggravating their metastatic burden (Fig. e-i), albeit only by trend upon MC-38 injection (...)". We do see a clear trend in these assays, particularly in the longitudinal BLI measurements where the p-value at the last time point is $p=0.08$ (now also indicated in the rev. Fig. 7). Along that line, to harmonize the IHC analyses on CD8+ cell infiltration in the MC-38 model with the analysis in the KPC model (please also see point 11), we re-analyzed these data. It now shows a clearer (and significant) reduction of CD8+ cells in $LysM^{\Delta Zeb1}$ lungs as compared to $LysM^{Ctrl}$ ones, matching the reduced tumor cell death (rev. Fig. 7c).

Moreover, to better understand the differential effect strengths between the cell lines KPC and MC-38 we have now performed Ki67 IHC staining to assess proliferation in tumor cells at endpoints upon TVI of either of them. Interestingly, while KPC tumor cells showed much higher fraction of Ki67+ cells in $LysM^{\Delta Zeb1}$ lungs as compared to $LysM^{Ctrl}$, no difference was compared in MC-38 cells (rev. Fig. 7e). In concert with the CD8+ cell influx and tumor cell death, we altogether reason that an anti-tumor immune response in the TVI may be cell line-independent in principle, whereas a cell-line dependent effect on proliferation further augments lung colonization and thereby the difference between $LysM^{\Delta Zeb1}$ and $LysM^{Ctrl}$ mice. This has now been placed in the main text (page 13): "(...) This indicates a cell line-dependent contextual impact of ZEB1 in macrophages on tumor cell proliferation but not on tumoricidal CD8+ cell influx (...)."

6. In Figure 4F, it is unclear why the authors chose to focus on CCL2 and CCL22, when other cytokines were also seemingly affected by Zeb1 loss. In particular, given the critical role of CXCR3 chemokine ligands in regulating T cell trafficking to tumors, did the authors evaluate CXCL9/10 expression in more detail (notably it appears that CXCL10 expression is reduced to the same or greater extent that CCL2 or CCL22 in the LPS treatment conditions). On a similar note, did the authors evaluate the effect of IFN γ treatment on LysM Δ Zeb1 macrophage phenotype? There is a large body of literature demonstrating an important role for IFN γ signaling in macrophages, and downstream IFN γ -induced chemokines (CXCL10) and cytokines, in regulating adaptive anti-tumor immune response. This would be an important comparison and perhaps more relevant than LPS treatment.

Answer:

We are grateful for this relevant suggestion and fully agree that IFN γ signaling and IFN γ -induced CXCR3 ligands, such as CXCL10, are very important for macrophage biology and function, also in the tumor context. Thus, we performed secretome arrays in IFN γ -treated BMDMs and observed partially differentially secreted cytokines, such as CXCL9, which was only secreted upon IFN γ but not upon LPS treatment. Secretion of various other cytokines, such as CXCL10, CCL2 and CCL22 was induced in a stimulus-independent manner. The latter were also, like most others, secreted less in LysM Δ Zeb1 macrophages, corroborating our LPS data and suggesting a stimulus-independent defect in release of these inflammatory mediators upon loss of ZEB1. These data are presented in rev. Fig. 4h and in the main text on page 9.

As to the experimental strategy, we initially observed a number of cytokines from which we chose to focus on CCL2 and CCL22 in the manuscript because we observed reduced secretion in M0 and LPS-stimulated BMDMs, unlike CXCL10, which was not detected in M0 BMDMs. Furthermore, we now included our data on CXCL10 in the downstream analyses, *i.e.*, bead-based determination of concentrations of secreted (rev. Fig. 4g) and intracellular (rev. Fig. 5g; see also point 7, 8 and reviewer #4, point 2) cytokines, as well as CD8 T cell migration. In the rev. manuscript, reduced secretion of CXCL10 upon LPS treatment is now confirmed quantitatively (rev. Fig. 4g). However, supplementation with murine CXCL10 in amounts leveling out the measured difference in secretion between the genotypes did not increase splenic CD8+ T cell migration towards LysM Δ Zeb1 BMDMs *in vitro*, indicating specificity of CCL2 and CCL22 in CD8+ T cell attraction at least in our experimental setup. Since we also performed cytokine receptor surface staining of these enriched CD8+ cells (rev. Fig. S6a; see point 9), revealing mostly coupled expression of CCR2 and CCR4 but not of CXCR3 (approx. 30% mostly single positive), these data imply that the low residual amount of secreted CXCL10 in LysM Δ Zeb1 BMDMs suffices to recruit the CXCR3+ CD8+ T cells, whereas CCL2 and CCL22 becomes functionally limiting. This data is now presented in rev. Fig. 6d. and in the results part of the main text in the respective paragraphs.

7. In Figure 5D, the differences in LysM Δ Zeb1 expression of select intracellular protein trafficking genes appears to be quite small, with the majority less than a Log₂FC of 1. Was differential expression of any of these genes evaluated at the protein level? Demonstrating a clear difference in expression of trafficking protein would help to support the authors claims that these relatively small mRNA expression differences mechanistically responsible for defects in secretion of select protein in LysM Δ Zeb1 macrophages. In addition, the authors should discuss why these broad mechanisms regulating intracellular protein trafficking/secretion would selectively affect a subset of secreted proteins in macrophages and only under certain stimulation conditions ?

Answer:

We thank the reviewer for pointing this out and agree that the differences in the bulk RNA sequencing are comparably small and cannot stand alone convincingly. This led us to substantially revise this mechanistic part of the manuscript in Fig. 5 (rev. Fig. 5). We have generated new BMDM cultures treated with LPS (as it is the main inflammatory stimulus throughout the manuscript) or left them untreated (M0): Using these cultures, we performed ultrastructural analysis by transmission electron microscopy (TEM; in collaboration with the anatomist Dr. Philipp Arnold at FAU Erlangen) as well as western blotting for

selected factors - RAB35, RAB6, VAMP3 and VAMP8 - that were differentially expressed upon LPS in the two genotypes upon LPS stimulations in bulk RNAseq. On the protein level, the RABs were generally increased as well as VAMP8, whereas VAMP3, a proposed pacemaker for cytokine release via the constitutive pathway of secretion in macrophages, was strongly reduced in *Zeb1*-deficient versus *Zeb1*-proficient BMDMs. Notably, the response to LPS was partially consistent with the bulk RNAseq data, i.e., VAMP3 appeared to be slightly more upregulated, whereas VAMP 8 was less induced upon LPS in the absence of ZEB1. As the TEM analyses revealed accumulation of multivesicular bodies (fused vesicles) and most strikingly, electron-dense, lysosomal-like vesicles in LPS-treated *LysM^{ΔZeb1}* BMDMs, the new data altogether support our claim that ZEB1 in macrophages affects intracellular trafficking of endosomal vesicles. Moreover, also intracellular cytokine concentrations have now been quantified (please see point 8). All these new data suggest that the constitutive pathway for cytokine secretion involving endosomal trafficking seems to be affected most by ZEB1 loss and indicate that degradative trafficking (lysosomes, autophagosome fusions) may be inhibited by ZEB1. The new findings are now presented in rev. Fig. 5. f, h-j, S5d, and are explained in the results part of the rev. manuscript on pages 10, 11 as well as in the discussion on pages 16, 17.

*8. In Figure 5F, it appears that almost all proteins on the array are increased in the *LysMΔZeb1* lysates relative to *LysMCtrl* lysates. This raises concerns about artifacts related to small differences in protein loading between groups and non-linearity associated with this method of protein detection. The authors should consider an orthogonal method for intracellular protein detection (e.g. intracellular FACS staining) to confirm these results .*

Answer:

We cordially thank the reviewer for bringing up this critical concern (see also reviewer #4 points 3, 5). We now measured intracellular concentrations of selected cytokines accurately from newly generated BMDM protein lysates using the bead-based Legendplex assay (Biolegend) as in Fig. 4g. This accurate and very sensitive technique revealed strongly reduced, rather than accumulated, cytokine levels in the *LysM^{ΔZeb1}* BMDMs. Considerably affected were TNF α , CCL2, CCL22 and CXCL10 which are mostly secreted via the endosomal, constitutive pathway in macrophages, whereas other, such as IL1 β , can be secreted differently (Murray and Stow, Front Immunol. 2014; doi: 10.3389/fimmu.2014.00538). This is consistent with TEM analyses showing more vesicle fusions and foremost lysosome-like vesicles, as well as the with the new western blotting data showing trafficking factors important for endosome-to-autophagosome and autophagosome-to-lysosomal fusions being more abundant in *LysM^{ΔZeb1}* BMDMs (see point 7). Although detailed mechanisms remain elusive, it altogether seems that loss of ZEB1 unleashes degradative vesicle trafficking and impairs the constitutive pathway of cytokine secretion in macrophages. This very important new data are presented in rev. Fig 5g, rev. S5d, as well as in the results (pages 10, 11) and discussion (page 16, 17) parts of the rev. manuscript.

*9. In Figure 6D, did the authors evaluate the combination of both anti-CCL2 and anti-CCL22? If both chemokines are responsible for driving CD8 T cell migration, an additive effect should be observed . For the current data, it is difficult to understand why anti-CCL2 and anti-CCL22 can both bring the level of T cell migration driven by *LysM^{Ctrl}* macrophages down to the same level as observed with *LysMΔZeb1* macrophages. Wouldn't one expect that blockade of either chemokines alone would only partially affect migration and that the combination of the two blocking antibodies would be required for full suppression of migration?*

Answer:

This is an astute remark that was also pointed out by reviewer #4 (point 4). This prompted us to determine the surface expression of the respective cognate cytokine receptors CCR2 (CD192), CCR4 (CD194) and CXCR3 (CD183) on the splenic CD8⁺ T cells used in the migration assay. This analysis showed that a subset of these CD8⁺ T cells show expression of CCR2 and CCR4 which was mostly coupled, i.e., these cells were double-positive. Subsets of these cells showed expression of CXCR3, which were mostly single-positive. This explains the lack of an additive effect upon CCL2 and CCL22

supplementation and depletion in this assay. The new data is presented in rev. Fig. S6a and in the main text of the rev. manuscript on page 12. Please also see point 6 for a potential implication of this finding.

10. In Figure 7A, the authors should state what timepoint post-tumor implantation this analysis was performed. Given these differences in CD8 T cell infiltration and cleaved caspase, one might expect a difference in tumor growth over time between LysMCtrl and LysMΔZeb1 groups. Was this observed (see comment 3 above)? If not, the authors should address this discrepancy.

Answer:

We have now stated the time points post injection in the respective figure legend of rev. Fig. 7a and analyzed the longitudinal growth curves for the CMT-93 model, as addressed in point 3. Indeed, we do see an effect, that is, tumor regression of CMT-93 tumors (rev. Fig. 3a). Further explanation and discussion on this issue is provided in point 3.

11. In Figure 7C, the authors report CD8 T cells/tumor colony in the lung and show a non-significant result. In Figure 7D, the authors report CD8 T cells/field of view and show a significant result. Why were different quantification approaches used? Given that differences in tumor colony formation in the lung were observed in the KPC model (Figure 3), quantification by field of view doesn't seem appropriate. Do the authors see differences in tumor colony-associated CD8 T cell density between LysMCtrl and LysMΔZeb1 groups when normalized to tumor colony number and tumor area ?

Answer:

We thank the reviewer for this comment and agree that analyses should be harmonized, wherever possible and sensible. Moreover, we apologize for the lack of accuracy and clarity in axis labeling and figure description. To clarify this, in the lung colonization models, the fields of view (FOV) are essentially “tumor cell-associated” and rarely do not contain tumor cells. They contain moderately sized to big macro-colonies (parts of it) and also smaller tumorous foci assembled in microcolonies, which are all relevant but a further categorization or normalization would only increase complexity of the data and its linked variance. In our opinion, this would not be helpful. To harmonize the datasets, we have now re-analyzed the CD8 IHC in Fig. 7c (rev. Fig. 7c) which was previously analyzed as CD8+ cells per colony and thus did not provide any normalization as to per area or alike. Using tumorous FOVs, as explained above, it now contains this internal normalization. This procedure also increased the datapoints in the graph (due to higher magnifications) and resulted in a significant effect. The new data is presented in rev. Fig. 7c and in the text on page 13. Furthermore, this procedure is now described in the methods on pages 49 and 50, stating: “(...) For analysis, fields of view (FOV) containing tumor cells, micro-, moderately sized and/or macro-colonies, were acquired using a Leica DM5500B microscope and automated quantification of these was performed using ImageJ. Briefly, images were formatted to RGB, deconvoluted for RGB and background subtracted for thresholding of positive cells. (...)”.

Reviewer #4:

In the present manuscript, the authors investigated the role of ZEB1 in macrophages by using conditional macrophage-specific ZEB1 KO (cKO) mice. The deletion of ZEB1 in macrophages did not affect organ development or homeostasis. However, tumor formation and colony formation in the lung were increased in cKO mice, suggesting the role for macrophage ZEB1 in anti-tumor effects. The secretion of CCL2 and CCL22 by cKO BM-derived macrophages (BMDM) was decreased compared to control macrophages. When mRNA levels of two chemokines were examined, there were only small differences between cKO and control macrophages, but the expression of genes related to protein trafficking, such as Rab family members and Vamp3 was affected, suggesting its role at a posttranscriptional level. As expected, protein synthesis increased upon LPS treatment but the full translational capacity was only slightly reduced in cKO BMDMs. Intriguingly, cKO BMDMs contained more intracellular cytokines upon LPS treatment than control BMDMs. Particularly, decreased CCL2 and CCL22 secretion levels were more evident than lower mRNA expression levels, and increased intracellular accumulation was remarkable. Finally, the authors found that CCL2 and CCL22 play a role in the recruitment of CD8+ T into tumors. From these result, the authors conclude that ZEB1 in macrophages promotes CD8+ cell infiltration into tumors and metastatic colonies in the lung via secretion of CCL2 and CCL22, preceding tumor cell death to counteract tumor outgrowth.

General comments:

Although some studies demonstrated a direct role of CCL2 in T cell recruitment, CCL2 is a chemoattractant for blood monocytes and its chemotactic activity for T cells is controversial. cKO macrophages secreted a lower level of CCL2, but they still secreted a significant amount of CCL2. Therefore, it is surprising that cKO BMDCs were incapable of recruiting T cells. There is not enough evidence to conclude the role for ZEB1 in the intracellular trafficking of CCL2 and CCL22 in macrophages. There are several sentences which are difficult to understand and editing is needed.

Answer:

We thank the reviewer for the critical assessment of our manuscript.

We fully agree that the chemotactic activity of CCL2 for T cells is controversial, which we also discuss in our manuscript (now on page 18). Although a significant amount of CCL2 was still secreted, the effect was evident, as the reviewer agrees. In the revised manuscript, we show that CD8 T cell migration towards BMDMs is sensitive to modulation of CCL2 and CCL22 but not to CXCL10 when leveling out the newly and accurately quantified concentration differences in secreted CXCL10 between LysM^{Ctrl} and LysM^{ΔZeb1} BMDMs (rev. Fig. 4g and rev. Fig. 6d). We also demonstrate that surface expression of CCR2 and CCR4 is mostly coupled in the splenic CD8+ T cells used in our assays, whereas CXCR3 is not, corroborating the unexpected functional redundancy of CCL2 and CCL22 for CD8 T cell attraction in this setting (rev. Fig. S6a). Moreover, our new finding that ZEB1 in macrophages controls KPC growth via the CCL2-CCR2 as well as CCL22-CCR4 axes that became particularly apparent upon adoptive transfer of CD8+ T cells in an organotypic precision-cut lung slice culture model (rev. Fig. 7, f-h), strongly supports the importance of CCL2 and CCL22 (rev. Fig. S7 f-h).

During the revision, we performed transmission electron microscopy and western blotting of selected trafficking factors to refine and substantiate our conclusion that ZEB1 regulates intracellular trafficking of cytokines including CCL2 and CCL22 in macrophages (rev. Fig. 5). These ultrastructural and molecular analyses jointly confirm a role of ZEB1 in vesicular trafficking in macrophages and indicate that loss of ZEB1 impairs vesicle release and seems to inhibit degradative vesicle fusions, thereby affecting the secretion of cytokines that are mostly secreted by the endosomal constitutive pathway.

We are confident that after editing and with all the additional experimental evidence, as pointed out in more detail below, our revised manuscript is mature enough to convince the reviewer.

Specific comments:

1. *Figure 1f and h: CD68+ macrophages on the Y axis. Are they numbers or percentages per field? Is there any functional differences between ZEB1 high and low macrophages ?*

Answer:

We thank the reviewer for identifying this inaccurate labeling which also was identified by reviewer #1 (point 1). It is number of CD68+ cells per field of view on the left and percentage of ZEB1+ cells among the CD68+ cells per field of view on the right. The figure legend and the axis labels have been modified accordingly in the rev. Fig. 1.

With regards to ZEB1-high and ZEB1-low macrophages, we agree that this is interesting and potentially relevant. However, the biology and the mechanisms of cellular plasticity and heterogeneity are very complex and, unlike it is the case for tumor cells, knowledge about ZEB1 as such in macrophages in the tumor context is still very limited. Thus, we chose to undertake a clean genetic approach employing conditional deletion of *Zeb1* to gain insights into its role in TAMs, discovering an influence on anti-tumor immunity. Exploring the relevance of ZEB1-high and ZEB1-low macrophages in a proper way, as we did recently for tumor cells (Schuhwerk et al., Cell Rep. 2022; doi:10.1016/j.celrep.2022.111819) is experimentally challenging and biologically too complex to align with the scope of this present study. Nevertheless, we now included this relevant idea in the discussion on page 17, stating: "(...) Particularly in light of the intracellular heterogeneity in ZEB1 expression even within the same cell subtype, it will be relevant to explore differential functions of the potentially selectively targetable ZEB1-high and ZEB1-low cells therein, similar to our recent study in tumor cells (Ref 32). (...)"

2. *Figure 4f: CXCR3 ligands are well accepted chemokines for CD8+ T cell recruitment. After LPS stimulation, the expression of CXCL10 was lower in cKO cells but there was no follow up study. In addition, the expression/secretion of other CXCR3 ligands, such as CXCL9, was not studied. It should be noted that significant concentrations of CCL2 and CCL22 were still detected in the supernatants of cKO cells .*

Answer:

We thank the reviewer for this suggestion (see also reviewer #3, point 6). We fully agree that IFN γ signaling and IFN γ -induced CXCR3 ligands, such as CXCL10 and CXCL9 are very important for macrophage biology and function. As also discussed above, it should be noted that CXCL9 was not secreted in response to LPS (rather than not being studied) but only in response to IFN γ , as we analyzed during this revision: We performed secretome arrays in IFN γ -treated BMDMs and observed partially differentially secreted cytokines, such as CXCL9, which was only secreted upon IFN γ but not upon LPS treatment. Secretion of various other cytokines, such as CXCL10, CCL2 and CCL22 was induced in a stimulus-independent manner. *Zeb1*-deficient macrophages secreted less of most cytokines upon IFN γ treatment as well, corroborating our LPS data and suggesting a stimulus-independent defect in release of these inflammatory mediators upon loss of ZEB1. These data are presented in rev. Fig. 4h and in the main text on page 9.

From a number of cytokines initially, which we chose to focus on CCL2 and CCL22 in the manuscript because we observed reduced secretion in M0 and LPS-stimulated BMDMs, unlike CXCL10, which was not detected in M0 BMDMs. CXCL9 was not followed up now because it was barely secreted upon LPS stimulation. Addressing the specific concern of this reviewer, we now included data on CXCL10 in the downstream analyses, *i.e.*, bead-based determination of concentrations of secreted (rev. Fig. 4g) and intracellular (rev. Fig. 5g, see also point 7, 8 and reviewer #4, point 2) cytokines, as well as CD8 T cell migration. In the rev. manuscript, reduced secretion of CXCL10 upon LPS treatment is now confirmed quantitatively (rev. Fig. 4g). However, supplementation with murine CXCL10 in amounts leveling out the measured difference in secretion between the genotypes did not increase splenic CD8+ T cell migration towards LysM^{AZeb1} BMDMs *in vitro*, indicating specificity of CCL2 and CCL22 in CD8+ T cell attraction in our experimental setup. Since we also performed cytokine receptor surface staining of these enriched CD8+ cells (see also point 4 and reviewer #3, point 9), revealing mostly coupled expression of CCR2

and CCR4 but not of CXCR3 (approx. 30% mostly single positive), these data imply that the residual amount of secreted CXCL10 in LysM^{ΔZeb1} BMDMs suffices to recruit CXCR3+ CD8+ T cells, whereas CCL2 and CCL22 becomes functionally limiting. This data is now presented in rev. Fig. 6d. and in the results part of the main text in the respective paragraphs.

3. Figure 5g: Secreted CCL2 by cKO cells was -1, whereas intracellular CCL2 was more than 10. Does this mean cKO cells produce more CCL2 in total? It would be more informative if real values by ELISA are presented. Arrays are good for a screening purpose but not to quantitate the concentration.

Answer:

We cordially thank the reviewer for bringing up this critical concern which was also brought up similarly by reviewer #3 (point 8). We now measured intracellular concentrations of selected cytokines accurately from newly generated BMDM protein lysates using the bead-based Legendplex assay (Biolegend) as in Fig. 4g. This very sensitive technique, revealed strongly reduced, rather than accumulated, cytokine levels in the LysM^{ΔZeb1} BMDMs. Considerably affected were TNF α , CCL2, CCL22 and CXCL10 which are mostly secreted via the endosomal, constitutive pathway in macrophages, but not others, such as IL1 β , which can be secreted differently (Murray and Stow, Front Immunol. 2014; doi: 10.3389/fimmu.2014.00538). This is consistent with our new transmission electron microscopy (TEM) analyses showing more vesicle fusions and foremost lysosome-like vesicles (rev. Fig. 5f) as well as with the new western blotting data showing trafficking factors important for endosome-autophagosome and autophagosome-to-lysosomal fusions being more abundant in LysM^{ΔZeb1} BMDMs (rev. Fig. 5h, i; see reviewer #3, point 7). Altogether, it seems that loss of ZEB1 unleashes degradative vesicle trafficking and impairs the constitutive pathway of cytokine secretion in macrophages. This very important new data are presented in rev. Fig. 5g, rev. S5d, as well as in the results (pages 10, 11) and discussion (page 16, 17) parts of the rev. manuscript.

4. Figure 6d: Actual numbers of migrated T cells should be presented. CCL2 and CCL22 use different receptors. It is unclear why addition of either chemokine or antibody almost completely abolished the differences .

Answer:

The lack of additive effects in our assay is an astute remark (see also reviewer #3, point 9). This prompted us to determine the surface expression of the respective cognate cytokine receptors CCR2 (CD192), CCR4 (CD194) and CXCR3 (CD183) on the splenic CD8+ T cells used in the migration assay. This analysis showed that a subset of these CD8+ T cells show expression of CCR2 and CCR4, which was mostly coupled, *i.e.*, these cells were double-positive. Subsets of the CD8+ T cells showed expression of CXCR3, which were however mostly single-positive. This explains the lack of an additive effect upon CCL2 and CCL22 supplementation and depletion in this assay. The new data is presented in rev. Fig. S6a and in the main text of the rev. manuscript on page 12. Please also see point 2 (and reviewer #3, points 6, 9) for a potential implication of this finding. With regards to data presentation, we noted that all experiments and littermate comparisons for the genotypes showed the same trends, but absolute cell numbers varied considerably, as expected for *ex vivo* handling of primary cells. This is why we normalized all values to the respective mean of the LysM^{Ctrl} PBS condition (for cytokine supplementation) or the IgG condition (for cytokine depletions). This enables pooling of repeats by focusing on the pattern of the data (*i.e.*, a biological response) whilst avoiding technical, not biological, variance (*i.e.*, a varying absolute scale).

5. The level of intracellular cytokine was presented as the amount/100 ug protein. What were the absolute amounts per cell or per a same number of cells ?

Answer:

We thank the reviewer for this comment. However, this data has been removed from the manuscript and been replaced by the quantitative and very sensitive bead-based assay presented in rev. Fig. 5g. Please also see point 3 for detailed response in this regard. It should be noted here that also for the new experiments, the same number of BMDMs were seeded, cultured overnight for attachment and then treated for 24 hours before sampling. Further detachment of extremely adherent BMDMs (especially upon LPS treatment) for cell counting at the timepoint of sampling (after this short period) has not been undertaken. Differences in proliferation or cell death in culture between the genotypes were not observed, even over longer periods (see rev. Fig. S4d, e). Along that line, the amount of harvested protein was similar between samples, indicating that cell numbers and protein content neither changed considerably nor correlated between genotypes and samples.

6. The biological relevance of *in vitro* T cell migration assay is questionable. The results were obtained after 24 hour culture. Why was this time point chosen? T cell migration can be evaluated with much shorter culture time. Random migration could be increased after a long culture .

Answer:

We agree that *in vivo* systems are superior to *in vitro* assay systems where random migration might be included upon prolonged time points. In our setting, we could not retrieve migrated T cells at much shorter time points of co-culture with BMDMs. A considerable number of migrated T cells was found at 24 hours, though, which is why we chose to collect the migrated T cells after 24 hours. We are aware that this harvest may contain a certain fraction of randomly migrated T cells, as indicated by the T cell-only points (grey bars) in Fig. 6d and rev. Fig. 6d. One astonishing observation from our point of view is that the LysM^{ΔZeb1} BMDMs fail to induce CD8+ T cell migration when compared to the T cell-only condition. To improve visualization of the mean differences observed in the experimental conditions, two colored lines indicating the mean of the control conditions upon LysM^{Ctrl} (blue) and LysM^{ΔZeb1} (red) co-cultures have been added to the graphs. Nevertheless, we agree that conclusions drawn from *in vitro* assays should be confirmed *in situ* or using other methods. In this regard, we now show reduced CD8+ cell abundance *in situ* (rev. Fig. 7a, c, d), correlating with CCL2 abundance in KPC-colonized lungs which we analyzed by IHC during the revision (rev Fig.7d; see also point 10). Moreover, efficient control of tumor growth (KPC) by adoptively transferred splenic CD8+ T cells (ACT) in an organotypic precision-cut lung slice culture model required *Zeb1* in macrophages and was sensitive to CCR2/CCR4 antagonists (rev. Fig. 7f-h) Collectively, it seems that CCL2 and CCL22 indeed support recruitment of CD8+ T cells in the tumor context, at least partially via the CCL2-CCR2 and CCL22-CCR4 axes.

7. Line 75: “by safeguarding intracellular protein transport”. What does “safeguarding” actually mean? The authors did not test whether ZEB1 actually regulates intracellular protein transport.

We have removed this inaccurate terminology throughout the manuscript. Notably, the abstract was substantially re-phrased.

8. Figure 7a: What is the histological score on the Y-axis?

Answer:

We apologize for the lack of clarity in axis labeling. It should say “IHC score”, as we also employed recently in a study on cancer-associated fibroblast in colorectal cancer (Menche and Schuhwerk et al., EMBO Reports 2024, doi: 10.1038/s44319-024-00186-7). This is now explained in the methods, as follows: “IHC was performed as previously described whereby IHC scores were estimated from the cellularity of (DAB-)staining positive cells, with none (0), rare (1), few (2), several (3) and many/abundant (4) (Ref 33 = Menche and Schuhwerk et al., 2024).”

9. *Figure 7d: Why were KPC cells used in this experiment? Despite the infiltration of CD8+ T cells was less in cKO mice, the size of metastatic foci appears smaller than those in control mice.*

Answer:

We thank the reviewer for this comment. The KPC cells were used for this kinetics experiment because in this model, a more robust and clearer difference in colonization was observed over time in longitudinal BLI analyses (rev. Fig. 3e) than in the MC-38 model. The original representative image of the $LysM^{\Delta Zeb1}$ was misleading (although it contained 2 colonies) and has been exchanged. Notably, at this time point (10 days post-injection), the colonies are not yet bigger in the $LysM^{\Delta Zeb1}$ mice than in $LysM^{Ctrl}$ mice.

10. *Figure 7d: It would be helpful to examine the kinetics of CCL2 and CCL22 expression in tumors to support the authors' conclusion.*

We are grateful for this suggestion. We now examined CCL2 abundance by IHC, which is reliable and fully setup in our and our collaborator's laboratory (Menche and Schuhwerk et al., EMBO Reports 2024, doi: 10.1038/s44319-024-00186-7). Given the similarities in the ZEB1-dependent CD8+ cell abundance and tumor cell death between the models, we focused on the analysis of CCL2 abundance in KPC-colonized lungs, as also described above, to assess its kinetics and compare it to the available CD8+ cell influx and tumor cell death. This analysis showed that CCL2 abundance increased over time, peaking at 10 days post injection. In accordance with our findings, $LysM^{\Delta Zeb1}$ KPC-colonized lungs contained far less CCL2 (-positive cells) than $LysM^{Ctrl}$ lungs and almost completely failed to induce it over time. This matches the reduced CD8+ cell abundance in situ (rev Fig. 7d) which precedes tumor cell killing in the $LysM^{Ctrl}$ lungs, altogether strengthening our hypothesis that ZEB1 in macrophages supports CD8+ cell influx and tumoricidal activity, at least partially via CCL2. This is also in line with the CCR2/4 and *Zeb1*-dependent effect of ACT in KPC-colonized organotypic lung cultures (see point 6).

11. *Figure 7e: The difference was statistically significant on only Day 21 and this result does not appear to support the conclusion.*

Answer:

We agree that statistical significance is only reached at day 21. However, a clear trend is visible at day 14 that does not reach significance due to variance. In our opinion, this result does support the conclusion that CD8+ cell influx precedes tumor cell killing and is not contradictory.

12. *Throughout the study, primary tumors were not examined. What about tumor size? When were tumors harvested? What about the expression of CCL2 and CCL22 in tumors? What about the infiltration of CD8+ T cells ?*

Answer:

We thank the reviewer for this comment. As primary tumors, we have analyzed subcutaneous (s.c.) tumors as surrogate for autochthonous tumors, whereas lung colonization was analyzed upon tail vein injection without prior formation of a primary tumor. For s.c. tumors, we have now included tumor sizes and volumes of CMT-93 tumors (rev. Fig. 3c) and still present CD8+ cell abundance and tumor cell death by IHC (rev. Fig. 7a). Moreover, during the revision, we have analyzed the s.c. growth of CMT-93 tumors longitudinally, revealing a strong effect of ZEB1 in macrophages on tumor regression after initial growth which is abolished in *Zeb1*-deleted mice (rev. Fig. 3a, b). This suggests an involvement of adaptive-anti tumor immune response and eventually manifests in a significant increase in tumor volume and mass in $LysM^{\Delta Zeb1}$ mice as compared to $LysM^{Ctrl}$ mice. MC-38 cells transplanted subcutaneously showed very rapid and aggressive growth kinetics. Following ethical guidelines, this resulted in premature dropout of a substantial fraction of mice, thereby decreased statistical power and was therefore not included in the manuscript. Please also see reviewer #3 (point 3) for further detailed elaboration on this topic including a figure showing the s.c. growth curves upon injection of 0.75×10^6 cells. While engraftment was not affected in $LysM^{\Delta Zeb1}$ mice, as stated in the main text, it appeared that a fraction of the tumors in $LysM^{\Delta Zeb1}$ mice grew bigger before the mice needed to be sacrificed. This prompted us to conduct the

limiting dilution assay (Fig. 3b and rev. Fig. 3d), in which for ethical reasons the animals were sacrificed early, *i.e.*, once a tumorous mass was visible and palpable over a few days. This confirmed a tumor-suppressive effect of ZEB1 in macrophages.

In rev. Fig. 7a, s.c. tumors are taken for IHC at 55-80 days-post injection (dpi) for CMT-93 and at 13-19 dpi for MC-38, which is now also indicated in the respective figure legend.

Given the similarities in the ZEB1-dependent CD8⁺ cell abundance and tumor cell death between the models, we focused on the analysis of CCL2 abundance in KPC-colonized lungs, as also described above, to assess its kinetics and compare it to the available CD8⁺ cell influx and tumor cell death. This analysis showed that CCL2 abundance increased over time, peaking at 10 days post injection. In accordance with our findings, LysM^{ΔZeb1} KPC-colonized lungs contained far less CCL2 (-positive cells) than LysM^{Ctrl} lungs and almost completely failed to induce it over time. This matches the reduced CD8⁺ cell abundance *in situ* (rev Fig.7d) which precedes tumor cell killing in the LysM^{Ctrl} lungs, altogether strengthening our hypothesis that ZEB1 in macrophages supports CD8⁺ cell influx and tumoricidal activity, at least partially via CCL2. This is also in line with the *Zeb1*-dependent control of KPC growth on organotypic precision-cut lung slice cultures via the CCL2-CCR2 and CCL22-CCR4 axes revealed upon adoptive transfer of CD8⁺ T cells (rev. Fig. 7f-h; also see point 6).

13. BMDCs culture: BM cells were seeded into a petri dish in BMDC medium. On day 1, nonadherent cells were cultured to prepared BMDCs. This is different from the method often used by others. It is unclear why nonadherent cells on day 1 were used to prepare BMDCs .

Answer:

We appreciate the reviewer's concern and fully agree that for most laboratory techniques there is different protocols and procedures. However, there was perhaps a confusion in the methodology. Bone-marrow-derived macrophages (BMDMs), unlike BMDCs, which is what the reviewer seems to refer to, can be isolated, cultivated and differentiated exactly as we have done and described it in the methods. To our knowledge, this is a well-established standard method for the isolation, differentiation and culture of murine BMDMs and we have now included a recent reference in this methods section of the revised manuscript (Auger et al., Nature 2024, doi: 10.1038/s41586-024-07282-7).

March 26th 2025

Subject: Revised manuscript : "Macrophages foster anti-tumor immunity by ZEB1-dependent cytotoxic T cell chemoattraction"

Manuscript Number: COMMSBIO-24-3324-T

Dear Reviewers,

We are very thankful for the assessment of our revised manuscript. We appreciate the few remaining comments and addressed them adequately with new supporting data, rigorous discussion and text editing, as described in detail in the point-by-point response on the following pages. Briefly, we:

1. assessed the abundance of the therapeutic CD8⁺ T cells in KPC-colonized PCLS, documenting its dependence on ZEB1 in macrophages and on the CCL2-CCR2 / CCL22-CCR4 axes.
2. provide new *in silico* analyses, reinforcing the functional crosstalk of macrophage-derived CCL2 and to a lesser extent, CCL22, with CCR2 and CCR4 expressing, tumor-infiltrating CD8⁺ T cell subsets, in addition to the well-established CXCR3 system.
3. refined the conceptual interpretation of our data by adding two new paragraphs in the discussion part to critically dissect residual inaccuracies in our conclusions, as commented on by the reviewers, and with all conscience, weakened claims on direct recruitment of CD8⁺ T cells into tumors by CCL2 and CCL22 *in vivo*.
4. As per editorial requests, we included supplementary files with the numerical source data for our graphs as well as for uncropped western blot scans, provided the requested clear statement on the numbers of mouse experiments and added individual data points in Fig. S1c.

Overall, we reinforced and critically refined a concept of an additional influence of ZEB1 in macrophages on intratumor CD8⁺ T cell abundance in human cancers via the CCL2-CCR2 / CCL22-CCR4 axes. After two revisions, we are confident that our original and translationally relevant study will raise the awareness of our identified critical yet tangible onco-immunological context within the biomedical research community.

Point-by-point response to the reviewers' comments:

Transparency notes to all Reviewers:

During the preparation of the supplemental file with the numerical source data for all graphs, we realized that in Fig. 4g, the graph for CXCL10 was incorrect and thus replaced in the re-revised manuscript version. The correct graph shows the same pattern and does not change any of our conclusions at all.

In general, the changes in the re-revised manuscript are marked in yellow.

Reviewer #1:

Most of the raised questions have been addressed. The new version of the manuscript is sufficiently improved.

Answer:

We thank the reviewer for the re-evaluation of our manuscript and are very happy that our manuscript is sufficiently improved now in the reviewer's eyes.

Reviewer #2 (unavailable):

Authors' note: According to the handling editor, this reviewer was unfortunately unavailable for a re-evaluation of our re-revised manuscript.

Reviewer #3:

The authors have addressed the majority of the questions I had raised in my original review. The quality and clarity of the revised manuscript have been strengthened and the overall conclusions are better supported by the experimental data. I have a few remaining questions and suggestions:

Answer:

We thank the reviewer for evaluating our revised manuscript and are happy to read that our overall conclusions are better supported by the experimental data. Further points are addressed below.

- In Fig 5, the authors conclude that reduced secretion of CCL2, CCL22, and CxCL10 are the result of increased trafficking of secretory endosomes to a degradative pathway in LysM-Zeb1 KO cell.

1) There is a discrepancy between Fig 5G and Fig 4G with respect to the VEGF control, where reduced intracellular levels of VEGF are observed in LysM-Zeb1 M0 macrophages, but this has no apparent effect on VEGF secretion. Similarly, M0 and LPS macrophages have differing levels of intracellular VEGF that doesn't appear to correlate to secreted levels. My assumption is that VEGF is meant as a control to show some degree of specificity of the Zeb1 effect to select chemokines. As such, that authors should address these discrepancies.

Answer:

This is an astute remark. We fully agree and also noted that VEGF seems to not fit into the hypothesis of a degradative pathway at the first glance. However, as we discussed in the revised manuscript, chemokines and cytokines are secreted via different, highly complex pathways. Moreover, actual utilization of these pathways seems to depend on the cargo, the (stimulatory) context and the cell type. This is an intensively studied field where many fine-tuning mechanisms and co-dependencies remain to be elucidated in detail. In the discussion on page 18, the text now states: "We detected generally much less intracellular VEGF, but similar amounts of secreted VEGF as compared to the other tested cytokines, implying that trafficking and release of VEGF might be more efficient than of the other tested cytokines, particularly in response to LPS. It should be emphasized that cytokines are processed and released via different, intricate pathways which depend on the cargo, the (stimulatory) context and the cell type (refs 45,49,69,70). For VEGF trafficking, a complex crosstalk exists, in which autophagy and angiogenesis, including VEGF production, secretion and activity, showed corresponding regulation in various models (refs 71-73). This may together indicate preferential usage of a distinct secretion pathway for VEGF release in BMDMs, such as 'secretory autophagy', as specifically proposed earlier in the context of age-related macular degeneration (ref 69). Notably, the trafficking of IL-1 β , which utilizes more secretory autophagy (ref 70), was largely unaffected by ZEB1 in BMDMs. Specific effects on the trafficking of VEGF, particularly in the M0 BMDMs, or of IL-1 β , might therefore underlie precise alterations in secretory versus degradative fluxes. These nuances, however, remain to be dissected in future studies."

Although we cannot pinpoint the exact shuttling or fusion process that would be unambiguously causal to the observed effects and their partial specificity, we appreciate the reviewer's important comment. It remains to be acknowledged (to the reviewer) that in light of this, VEGFs and IL-1 β indeed seem to be excellent additional control cargos to study ZEB1-dependent intracellular protein trafficking and autophagic flux in macrophages in detail within mechanistic follow-up projects.

2) Given the importance of Fig 5 to deciphering the mechanism of Zeb1-mediated effects on macrophages, have the authors attempted to pharmacologically block lysosomal degradation or autophagy pathways in macrophages? Showing a restoration of intracellular levels of chemokines would provide important supporting evidence for the model shown in Fig 5J.

Answer:

We thank the reviewer for the interesting comment. Rescue experiments for proving the hypothetical model of degradative vesicle transfer is a relevant suggestion. Although we fully share the reviewer's enthusiasm about this new hypothesis in general, one has to keep in mind that it developed on the basis of new data generated during the revision. As described above, vesicular trafficking in general, and the cargo-dependent choices between degradative and secretory routes underlies sophisticated detailed mechanisms that are still poorly understood. However, we are convinced that this needs to be addressed within a dedicated follow-up study in an adequate mechanistic depth. The dissection of intricate vesicular shuttling and fusion pathways that depend on multiple factors, such as cargo and cell type, entails a bigger experimental effort and requires specific techniques, as well as a lot of time and funds. In this regard, we are currently drafting new grant proposals for which the present manuscript should be the major experimental basis. Hence, we are convinced that this is out of scope for the present manuscript but we would be happy to submit a new mechanistic study in the future.

- In Fig 7F-H, the authors utilize PCLS to further investigate mechanisms of Zeb1 regulation of macrophage function and anti-tumor T cell responses. Adoptive cell therapy is generally used to describe transfer of cells *in vivo*. In this case, my understanding is the CD8+ T cells are added to the PCLS culture. Is this correct? If so, the authors may want to avoid using this terminology to avoid confusion (*in vivo* transfer before harvesting lung tissue or adding T cells *in vitro* to the PCLS culture).

Answer:

We thank the reviewer for spotting a potential source of confusion. Indeed, we added the CD8+ T cells *in vitro*. Accordingly, we now exchanged the terminology to "CD8+ T cell addition" - abbreviated as "CD8+ TCA" - in the main text and in Fig. 7 as well as Fig. S10. Moreover, we removed the term adoptive cell transfer from the keywords.

Reviewer #4:

1. As noted in my previous comments, the major chemokine/chemokine receptor system regulating the trafficking of CD8+ T cells into solid tumors is CXCR3 and its ligands, such as CXCL9, CXCL10 and CXCL11 (reviewed by Kohli et al., *Cancer Gene Therapy*, 29:10-21, 2022). After a revision, it is still unclear whether CCL2 and CCL22 directly regulate the recruitment of CD8-positive T cells in the models used in this study. This important point is not even discussed.

Answer:

We thank the reviewer for bringing up this important point. We already discussed before that while CCL2 and CCL22 have been shown to recruit CD8+ T cells *in vitro* and also *in vivo*, there is an apparent ambiguity in the experimental evidence from literature. We have now newly added a thorough discussion about our data and the potential roles of CCL2 and CCL22 we may infer from it (see below and the re-revised manuscript on pages 19 and 20). Moreover, we have amended our abstract and the text accordingly to avoid potential overinterpretation by the scientific community.

We want to take this opportunity to clarify that neither our data, nor we with any claims, are challenging the well-established and important role of CXCR3 ligands for recruitment of CXCR3+ CD8+ T cells into solid tumors. Importantly, it was and it still is not the aim of this study to mechanistically cross-compare different receptor-ligand systems to evaluate, which one is quantitatively more important. That would be another study in our eyes. Hence, we emphasized throughout the re-revised manuscript that we are merely suggesting potential additional function of CCL2 and CCL22 in regulating CD8+ T cell abundance and, in all conscience, have weakened bold conclusions as well as suggestive wording on direct effects on CCL2 and CCL22 on direct CD8+ T cell recruitment *in vivo*.

Nevertheless, we sought to provide more evidence that CCL2 and CCL22 can additionally contribute to increasing CD8+ T cell abundance in tumors – in addition to CXCR3 ligands. Arguing for an additional effect of macrophage-derived CCL2 and CCL22 on CCR2 and CCR4 in CD8+ T cells, we now show *CXCL10*, *CCL2* and *CCL22* expression in macrophages and monocytes as well as *CXCR3*, *CCR2* and *CCR4* expression in CD8+ T cell subtypes, making use of publicly available single cell RNA sequencing datasets from tumor-draining lymph nodes of MC38 tumor-bearing mice, as well as from human colorectal, pancreatic and breast cancers (Fig. S7, S8a-c). Moreover, we now present a similar positive correlation of CD8+ T cell abundance with expression of *CCR2*, *CCR4* and *CXCR3*, as deconvoluted from publicly available bulk RNA sequencing data from human colorectal and pancreatic cancer patients using *Timer2.0*. (Fig. S8d).

Given the pleiotropic, tumor-promoting and suppressive roles of CCL2 and CCL22, which we have already discussed in the original and in the revised manuscript, we invested experimentally in confirming their relevance for CD8+ T cell responses during the first revision. Thereby, we not only showed in a longitudinal IHC analysis that CCL2 abundance correlates with CD8+ T cell abundance in KPC lung colonies, but also demonstrated in a therapeutically relevant PCLS system, that (KPC) tumor control by CD8+ T cell addition depends on ZEB1 in macrophages and on the CCL2-CCL22/CCR2-CCR4 axes. Following up in this, we now provide evidence that early KPC growth control in LysM^{Ctrl} PCLS appeared to correlate with reduced abundance of therapeutically added CD8+ T cells, as approximated by total Claret intensity – used to label the CD8+ T cells prior to addition (Fig. S10a) – and that this was reduced upon treatment with CCR2 and CCR4 antagonists (Fig. S10b). These data are presented on page 14 of the re-revised manuscript and argue for a positive influence of ZEB1 in macrophages and of the CCL2-CCR2 and CCL22-CCR4 axes on CD8+ T cell abundance in tumorous tissues.

In addition to the previous discussion on this topic in the revised manuscript, we have now clarified on page 20 of the re-revised manuscript that from the obtained *in situ* IHC and PCLS data, we cannot definitely conclude if the increased abundances of CD8 + T cells in PCLS and in IHC analyses are a direct consequence of enhanced recruitment or may underlie alternative, indirect mechanisms. These would include enhanced CD8+ T cell activation, or a more complex crosstalk within the TME. Arguing for a direct regulation of CD8+ T cell recruitment by CCL2 and CCL22, we did not see an effect on CD8+ T cell proliferation *in vitro* that depended on ZEB1 in BMDMs but found that CD8+ T cell migration can be modulated by CCL2 and CCL22 manipulation in a manner dependent on ZEB1 in BMDMs (see also the response to point 2).

We therefore favor the interpretation that CCL2 and CCL22, derived from (ZEB1-positive) BMDMs, most likely do not mainly affect T cell activation and proliferation but their migration, at least *in vitro*, and that this might contribute to CD8+ T cell infiltration *in vivo* - in addition to the CXCR3 pathway.

2. Raw data of T cell migration assay are not presented, making it difficult to evaluate the results. The experiment should be conducted with positive controls. It would be more straightforward if the authors used cell culture supernatants instead of BMDMs (Fig. 6d).

Answer:

We regret to read that the reviewer questions our technique and we thought to have answered this after the first revision. Exploring unprocessed, unnormalized and thus potentially misleading individual data points bears the risk of misinterpretation, given that this is an *ex vivo* system with experiment-to-experiment, mouse-to-mouse and litter-to-litter variation. In general, we want to and have to focus on biological effects, not technical variation if we want to gain biological insight.

Please see below a representative set of unprocessed CD8+ T cell numbers. Therein, marked variations between litters (“_1”; “_2”; “_3”) and/or experiments, as indicated, would probably not allow a clear conclusion. After normalization, however, two similar and reproducible biological effects are revealed. This is i) reduced CD8+ T cell migration towards $LysM^{\Delta Zeb1}$ BMDMs and ii) an elevation of CD8+ T cell migration towards $LysM^{\Delta Zeb1}$ by CCL2 addition.

In this regard, it is important to mention that, all raw data constituting the main and supplemental figures have now been provided as a supplementary excel file.

Nevertheless, we would like to take this chance to re-evaluate our assay, aiming to further increase the reviewer's trust in it:

With regards to technical controls and the reliability of our assay, we measured a significant downregulation of the migration of splenic CD8+ T cells towards *Zeb1*-proficient BMDMs upon depletion of CCL2 or CCCL22. This can be regarded as negative controls, in principle, and already demonstrates that our assay works. Notably, we also assessed the expression of these most relevant cytokine receptors on the splenic CD8+ T cells during the first revision (CCR2, CCR4 and CXCR3; Fig. S6) which helped us to further validate the performance of our assay. In fact, strongly suggesting reliability and sensitivity of our assay, the coupled surface expression of CCR2 and CCR4 on the splenic CD8+ T cells used in this setting seems to be reflected in the absence of an additive effect of combined addition of CCL2 and CCL22 on CD8+ T cell migration towards *Zeb1*-deficient BMDMs.

As the reviewer correctly stated in point 1, the CXCR3 pathway is the most established receptor-ligand system for CD8+ T cell recruitment into solid tumors. It is very important to emphasize again that we

fully agree to this. However, we did not observe increased migration of splenic CD8+ T cells *in vitro* towards BMDMs when adding CXCL10 in physiologically relevant concentrations, which essentially level out the differences in secretion by *Zeb1*-proficient versus *Zeb1*-deficient BMDMs. Of course, this does not mean that the CXCR3 system is not important *in vivo* but indicates that in our *in vitro* CD8 + T cell migration assay, CXCR3-linked migration seems to be largely independent of ZEB1 in BMDMs. We reason that the levels of CXCL10 in *Zeb1*-proficient and *Zeb1*-deficient BMDMs are sufficient to saturate CXCR3 binding in the tested concentration, which may actually imply high efficiency of the CXCR3 system for CD8+ T cell chemotaxis.

Addition of a physiological amount of CXCL10 did not increase CD8+ T cell migration towards BMDMs but we did not test, if adding more CXCL10 would have increased CD8 + T cell migration in our setting. However, as described above, we deem our assay to be reliable and a functional. A detailed side-by-side comparison of different cytokine-receptor axes seems peripheral to our study centered around ZEB1 in macrophages. The aim of our manuscript is to document an influence of ZEB1 in macrophages on anti-tumor immune responses and link this to the abundance of CD8+ T cells. Therein, cytokine trafficking and release is strongly affected by ZEB1 loss and less well-established ligand-receptor systems also seem to contribute to increase CD8+ T cell abundance in a ZEB1-dependent manner, as discussed above and in the manuscript on pages 19 and 20.

We appreciate the reviewer's technical suggestion with regards to using conditioned medium. However, as described above, we are convinced that our assay is reliable and changing the experimental system appears questionable to us. In particular, switching to conditioned medium instead of a transwell co-culture setting, in which we can modulate the migration of CD8+ T cells accordingly by depleting and adding cytokines in a physiologically relevant range (see above), is in our eyes a step backward, not forward and will consequently not generate more meaningful insights.

3. In tumor microenvironments, not only macrophages but also other cell types, including cancer cells, produce and release CCL2. In some mouse cancer models, cancer cells rather than macrophages are the main source of this chemokine. It is important to examine the levels of CCL2 and CCL22 in tumors growing in control and cKO mice.

Answer:

We agree that cytokines can be secreted from various cell types. In this regard, we use a conditional, homozygous deletion of the *Zeb1* alleles by foremost LysM-Cre to conduct our study, thereby limiting the direct influence from other confounding cell types on our results. In PCLS, we provide sound evidence that KPC tumor control by CD8+ T cells depends on ZEB1 in macrophages and on the CCL2-CCL22/CCR2-CCR4 axes. From publicly available datasets, as described above, we can now show that in CRC, a fraction of CXCL10, CCL2 and, albeit less clearly, of CCL22, is derived from the myeloid/monocyte compartment. These specific results are presented in Fig. S7a, and in the text on pages 13 (results part) as well as 20 (discussion part).

In line with this observation, during the first revision, we demonstrated that kinetics of CCL2 levels in KPC-colonized lungs depend on ZEB1, as suggested by the reviewer (Fig. 7d). We regret to read that this important insight is apparently not appreciated by the reviewer. We chose KPC lung colonies for this experiment because time points had been generated and the IHC is reliable and established in our laboratory (Menche and Schuhwerk et al., *EMBO Reports* 2024; doi: 10.1038/s44319-024-00186-7). This is a very important piece of data strengthening our hypothesis remarkably and we thank the reviewer for suggesting this in the first revision. From our point of view, however, repeating CCL2/CCL22 IHC analyses on sections of endpoint tumors will not yield a significant knowledge gain and insight, because it would lack longitudinal information of cytokine abundances, yet still remain at a correlative level. This said, new animal experiments would be required for longitudinal dissection and for specific modulation of the cytokine-receptor axes *in vivo*, entailing significant ethical concerns. We approached this alternatively, not only by utilizing PCLS cultures combined with therapeutic CD8+ T cell addition (also see our response to point 1) but also by *in silico* analyses from publicly available datasets, as also explained above, revealing that in colorectal cancers, CCL2 and CCL22 seem to also derive from the macrophage/monocyte compartment, while subsets of CD8+ T cells express *CCR2* and *CCR4*, albeit to

a lesser extent than *CXCR3*, as expected. Furthermore, we show that expression of *CCR2*, *CCR4* and *CXCR3* all correlate positively with CD8+ T cell abundance in human cancer, as deconvoluted from publicly available bulk RNA sequencing data from human colorectal and pancreatic cancer patients using *Timer2.0*. (Fig. S8d). These data altogether indicate that macrophage-derived CCL2/CCL22 indeed seem to act on CCR2/CCR4 on CD8+ T cell subsets in cancer, in addition to CXCR3 ligands. These new *in silico* analyses are presented in Fig. S7 and Fig. S8, described in the text on page 13 and discussed in the text on page 20.

4. Several new sentences/paragraphs are added but they are not clearly written and difficult to understand.

Answer:

We regret to hear that the reviewer did not easily understand some unspecified, newly added sentences. To the best of our knowledge and in all conscience, we used accurate and specific wording whilst trying to avoid overly complicated sentences. We have also proofread our manuscript again and hope that it is now in good shape for acceptance.

Friedrich-Alexander-Universität
Medizinische Fakultät

Uniklinikum
Erlangen

Harald Schuhwerk · Dept. of Exp. Med. I · Glückstr. 6 · D-91054 Erlangen

May 13th, 2025

Subject: Revised manuscript : "Macrophages foster anti-tumor immunity by ZEB1-dependent cytotoxic T cell chemoattraction"

Manuscript Number: COMMSBIO-24-3324B

Dear Reviewers,

We are very thankful for your continued assessment of our manuscript over the whole peer-review process. The repeated critical evaluation of our data and text, as well as the specific comments you have raised, allowed us to improve our study remarkably.

Sincerely,

Harald Schuhwerk (on behalf of all authors including Thomas Brabletz)